# Remote detectability from entanglement bootstrap I: Kirby's torus trick

**Bowen Shi, Jin-Long Huang and John McGreevy**

Department of Physics, University of California at San Diego, La Jolla, CA 92093, USA

## Abstract

Remote detectability is often taken as a physical assumption in the study of topologically ordered systems, and it is a central axiom of mathematical frameworks of topological quantum field theories. We show under the entanglement bootstrap approach that remote detectability is a necessary property; that is, we derive it as a theorem. Starting from a single wave function on a topologically-trivial region satisfying the entanglement bootstrap axioms, we can construct states on closed manifolds. The crucial technique is to immerse the punctured manifold into the topologically trivial region and then heal the puncture. This is analogous to Kirby's torus trick. We then analyze a special class of such manifolds, which we call *pairing manifolds*. For each pairing manifold, which pairs two classes of excitations, we identify an analog of the topological *S*-matrix. This *pairing matrix* is unitary, which implies remote detectability between two classes of excitations. These matrices are in general not associated with the mapping class group of the manifold. As a by-product, we can count excitation types (e.g., *graph excitations* in 3+1d). The pairing phenomenon occurs in many physical contexts, including systems in different dimensions, with or without gapped boundaries. We provide a variety of examples to illustrate its scope.

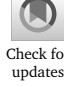

# 1 Introduction

Topological quantum field theory (TQFT) [1–3] is a machine that eats arbitrary manifolds and produces invariants. The entanglement bootstrap [4–6] begins with a single state on a topologically-trivial region, e.g., a ball or a sphere. How can it incorporate the data on nontrivial closed manifolds? In this work, we provide a method to construct quantum states on various manifolds from the reference state on a ball. With this technique at hand, we further give a general proof of the remote detectability for topologically ordered systems.

Remote detectability [7, 8] is the statement that, in topologically ordered systems, each nontrivial topological excitation (which can be a particle or a loop or something else) must be detectable by a remote process involving another (possibly different) class of excitations. It is a broad phenomenon that is believed to occur in many physical setups, including topologically ordered systems in arbitrary dimensions ($d \geq 2$) [7, 9–11] and systems with gapped boundaries [12–14]. Remote detectability is a vast generalization of the braiding non-degeneracy of anyons; see Ref. [15] Appendix E. In these previous approaches, remote detectability was an axiom (or principle). In entanglement bootstrap, we shall derive this property as a theorem.

The main idea in our derivation of remote detectability is as follows. We start with a reference state $\sigma$ on a topologically trivial region, which can be either a ball $\mathbf{B}^n$ or a sphere $\mathbf{S}^n$.[1] We immerse (i.e., locally embed) a closed manifold $\mathcal{M}$ of interest into the ball or sphere upon removing a ball from it, that is, we consider $\mathcal{W} = \mathcal{M} \setminus B^n$ and an immersion $\mathcal{W} \pitchfork \mathbf{B}^n$ or $\mathcal{W} \pitchfork \mathbf{S}^n$. Then with a trick to heal the puncture, we construct a state on the manifold $\mathcal{M}$. We identify a specific class of manifolds that is important for the study of remote detectability; we refer to them as *pairing manifolds*. Each pairing manifold ($\mathcal{M} = X\bar{X} = Y\bar{Y}$[2]) pairs two excitation types, namely the excitation types characterized by the information convex sets of regions $X$ and $Y$ respectively. For each pairing manifold, we identify a "pairing matrix", which is a finite-dimensional unitary matrix analogous to the topological $S$ matrix in the anyon theory. The remote detectability of the excitations follows from the unitarity of the pairing matrix. (See Ref. [16] for an entanglement bootstrap study of the topological $S$ matrix in the anyon theory, which contains a trick we adopt.)

Our method can be thought of as a quantum analog of Kirby's torus trick [17]: it uses immersion and pulls back structures from a topologically trivial region to closed manifolds (e.g., the torus). Then, because the structure on the closed manifold (i.e., the stability in Kirby's case) is better understood, insight into the topologically trivial region (i.e., $\mathbb{R}^n$ in Kirby's case) is gained by the consistency. In our method, we pull back quantum states to punctured manifolds and then heal the puncture.[3] The structures, i.e., universal data, associated with the quantum state on the topologically trivial region understood by our method include the pairing matrix, best defined on the pairing manifolds, as well as various consistency relations derived by making use of the pairing manifolds. Intriguingly, Hastings considered an application of Kirby's torus trick for gapped invertible phases [18], where the local Hamiltonian is pulled back. In comparison, our approach works for gapped invertible phases as well as intrinsic topological orders, and we only make use of the quantum states. One innovation is the use of *building blocks*, which carry the instruction for picking the right state on the punctured manifold. Such states allow a "smooth" healing of the puncture.

Why are we interested in the entanglement bootstrap approach to topological orders and TQFT? The entanglement bootstrap is an independent theoretical framework: it requires an

---

[1]We use boldface letters $\mathbf{B}^n$ and $\mathbf{S}^n$ (instead of the more standard topological notation $B^n$ and $S^n$) to indicate that these regions are those on which we define the reference state.

[2]Here $\bar{X}$ is the complement of $X$ in $\mathcal{M}$, that is $\bar{X} = \mathcal{M} \setminus X$.

[3]The quantum state lives on the 0-cells of a cell decomposition of the ball. As in the original application of the torus trick, the immersion can be used to pull back the information about this cell decomposition. Thus, the pulled-back quantum state lives on a cell decomposition of the immersed manifold.

input reference state that satisfies two axioms on bounded-sized regions (see Eq. (2)). From there, various rules are derived as information-theoretic consistency relations of the quantum theory. In the process of deriving such consistency relations, mathematical objects are identified that capture the universal data of the gapped phase. These data are encoded in the reference state, which is physically the ground state. Below are some additional perspectives:

1. TQFT and its categorical description, while standing as the best-known candidate of the underlying mathematical theory for topological field theory, are not without ambiguities. In mathematics, it is possible to add various adjectives to TQFTs and tensor category theories. Therefore, there has always been the question of which adjective is the right one for a given physical setup. Furthermore, the study of TQFT and topological orders in higher dimensions ($d \geq 3$) (see *e.g.* [7,8,11,19]) is an ongoing research direction.

2. The entanglement bootstrap is a bridge between the program of classifying gapped quantum phases and the classification of quantum states. This is because any data we identify (such as the pairing matrix), becomes a label of the reference state which satisfies an entanglement area law captured by the axioms. In this way, entanglement bootstrap puts labels on quantum states satisfying the axioms and thus classifies them.

3. One expects to discover new connections between the ground states and universal properties. For instance, in this work, we identify excitations that have not been studied, e.g., the graph excitations in 3d topological order. They are excitations occupying a thin handlebody. Handlebodies in 3d are classified by genus, and for each genus, we identify a class of "graph excitations". The remote detectability for this new class of graph excitation is one of the many examples we give. As a byproduct, we provide a counting of the number of graph excitations for an arbitrary genus. As we shall explain, the counting manifests the fact that the coherence in the fusion space in the particle cluster is needed in detecting the graph excitations.

   Recently, some of the tools developed in entanglement bootstrap have been useful guidelines for finding new topological invariants calculable from a ground state wave function. (See [20] for a proposed formula for the chiral central charge, in terms of the modular commutator. See [21] for a proposed formula for the Hall conductance, in the context of which the reference state is symmetric under a $U(1)$ onsite symmetry.) These studies suggest that the ground state wave function on a topologically-trivial region contains data beyond those captured by a tensor category. One may wonder if this line of ideas generalizes into higher dimensions.

4. Lastly, entanglement bootstrap, rooted in earlier related works, suggests a new philosophical relation between the universal data and the ground state(s). It suggests, in a concrete manner, that many (possibly all) data of the gapped phase of matter (or TQFT) can be extracted from a topologically trivial patch of the ground state wave function. In other words, the universal data is encoded in the seed or its "DNA" (reference state on a patch larger than the correlation length) of the "plant" (topologically ordered system). Our approach is to first grow a plant from the seed and then study the morphology of the plant; therefore, we do not need to start with the whole plant. See Fig. 1 for an illustration.

The pairing matrices we identify are analogs of the topological $S$ matrix in the anyon theory. A surprise is that, while each of these pairing matrices is associated with a closed manifold, the matrix is not associated with its mapping class group (MCG), unless the two classes of excitations characterized by $X$ and $Y$ are identical. This distinguishes them from a previous generalization of the topological $S$ matrix in 3d, associated with the 3-tori [10,22]. Some of

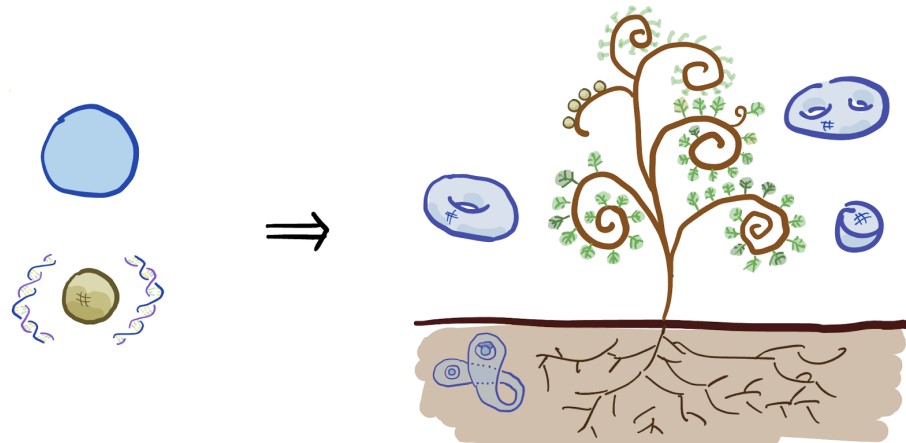

Figure 1: Illustrated is an analog to the growth of a plant from a seed.

the matrices considered in [10, 23, 24] are examples of pairing matrices. Pairing matrices fall into three kinds, depending on whether $X$ and $Y$ encode a nontrivial fusion space; see Table 1 for the examples we consider.

## 1.1 Reader's guide

This section is a reader's guide. It includes a figure that summarizes the key concept developed (or reviewed) in each section and the relations between the sections (see Fig. 2), and a table that summarizes the examples of pairing manifolds (see Table 1). Below are some more details.

### 1.1.1 Content of the sections

In §2, we collect useful tools of entanglement bootstrap, that are developed in previous works. We start by reviewing the axioms in general dimensions. The structure theorems of the information convex sets enable us to talk about superselection sectors and fusion spaces. The merging theorem allows us to glue regions on either an entire entanglement boundary or part of the entanglement boundary. The associativity theorem tells us how the dimensions of the fusion spaces match upon gluing an entire entanglement boundary. We introduce the idea of constrained information convex set, which allows a nice reformulation of the associativity theorem. We recall the definition of quantum dimensions and the properties of the vacuum. Finally, we recall the concept of immersed regions,[4] which will play an important role in this work.

In §3, we explain the duality between excitations and regions. In the bulk, we shall consider excitations (which can be one excitation or a cluster of excitations) that live on a sphere $S^n$. The region dual to the excitations is homeomorphic to the subsystem of a sphere that occupies the complement of the excitations. We also discuss the immersed version of such regions. These regions will be used as "building blocks" for constructing closed manifolds. We further discuss similar dualities in the physical context of gapped boundaries. We also discuss excitation types that are new to our knowledge. For instance, in the 3d bulk, we identify *graph excitations* which are located in "graphs".[5] They are more general than the familiar loop ex-

---

[4]As is explained in Ref. [6], the idea is essentially the same as *topological immersion*: a continuous map from a topological manifold where every point in the source has a neighborhood on which the map restricts to an embedding. We only need immersion maps between manifolds of the same dimension, and in this case, the immersion in question is also a submersion.

[5]An accurate statement is the excitations are supported on thin handlebodies. Handlebodies in 3d are solid genus-$g$ surfaces.



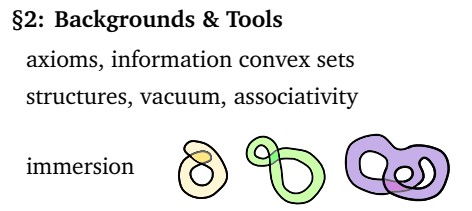

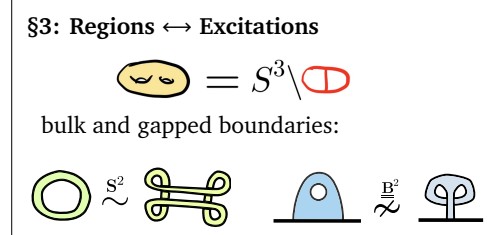

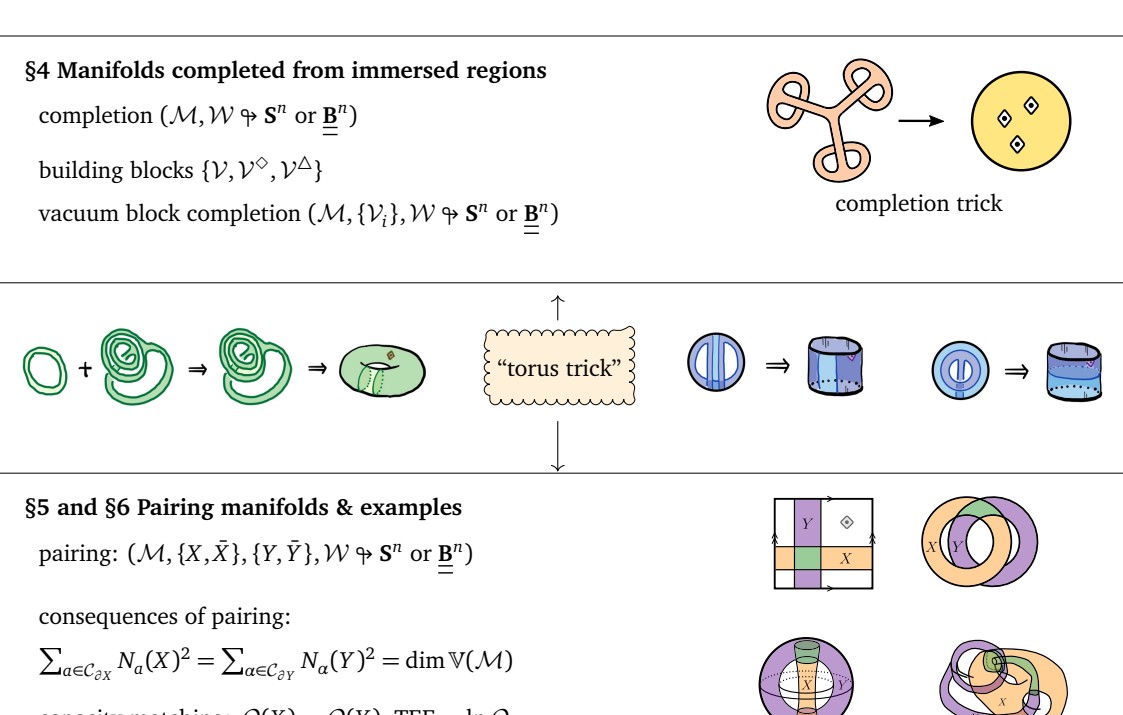

Figure 2: A summary of the content of the main text. The focus is on the concepts developed and the relations between the sections.

citations. The regions that detect them are also handlebodies. These graph excitations are classified into subclasses labeled by the genus.

In §4, we discuss the main conceptual progress of this work: that is the idea of making closed manifolds $\mathcal{M}$ from immersed regions $\mathcal{W} \looparrowright \mathbf{S}^n$, where $\mathcal{W} = \mathcal{M} \setminus B^n$. The concepts and techniques to achieve this goal include completion (Definition 4.1), completion trick (Lemma 4.4), building blocks (Definition 4.5), and vacuum block completion (Definition 4.9). Intuitively, if a closed manifold $\mathcal{M}$ allows a completion, then there exist quantum states on $\mathcal{M}$, which are assembled from the reduced density matrices of the reference state on small patches. The completion trick is a general trick to achieve that. Vacuum block completion is a special type of completion; it carries an instruction to build the closed manifold from a set of building blocks. Each manifold so constructed has a canonical state with respect to the choice of building blocks. This canonical state is obtained by assembling the "vacuum" states of these building blocks. Importantly, the canonical state satisfies the entanglement bootstrap axioms on all balls contained in $\mathcal{M}$. We provide many examples to illustrate these concepts and techniques.

In §5, we introduce the concept of *pairing manifold* (Definition 5.3). It is a special class of manifolds that allows *a pair* of vacuum block completions, as

$$\mathcal{M} = X\bar{X} = Y\bar{Y}, \tag{1}$$

in addition to a couple of extra conditions. The most important condition is that $X$ and $Y$ cut each other into balls and thus hide information from each other completely. From the definition, we prove a set of properties of pairing manifolds. For instance, the Hilbert space dimension on $\mathcal{M}$ is determined by the information convex set of either $X$ or $Y$. In short, this is a manifestation that a pairing manifold is a machine that pairs two classes of excitations.[6] One class is that detected by the region $X$, and another is that detected by the region $Y$.

In §6, we provide many examples of pairing manifolds in various spatial dimensions and systems with or without gapped boundaries. See Table 1 for the examples in 2, 3, and 4 spatial dimensions. As a by-product of this analysis, we count the total number of graph excitations for any given genus. The result is expressed in terms of the fusion multiplicity of point particles. We show that some manifolds can be a pairing manifold in multiple ways.

In §7, we introduce the pairing matrix ($S$), a generalization of the well-known topological $S$-matrix. This is a different generalization that has been considered in [22], and the pairing matrices are not necessarily associated with the mapping class group. A pairing matrix generates a unitary transformation between two bases specified by the two cuts, and it is a unitary matrix. The pairing matrices fall into three types, depending on whether a fusion space needs to be involved in the remote detection process; see Table 1 for some examples. A pairing matrix of the first type can be thought of as the braiding matrix between two classes of excitations; these excitations detect each other without making use of fusion spaces, and the number of excitations in each class must be identical. A pairing matrix of the second type provides an example of remote detection involving nontrivial fusion spaces. The physical picture to keep in mind is that a cluster of coherently-created excitations detects the excitations in another class. A pairing matrix of the third type generically requires the fusion spaces of both excitation clusters (one detected by $X$ and another detected by $Y$) to participate in the remote detection process.

§8 discusses open questions. In Appendix A, we summarize the notations and provide a glossary. In Appendix B, we prove a set of consistency conditions on quantum dimensions and fusion rules, generalizing those found in [6]. This is used in Appendix C, which gives an exposition of graph excitations, and exemplifies them using quantum double models. Appendix D

---

[6]When either $X$ or $Y$ is not sectorizable, it is useful to think of these as two clusters of excitations.

illustrates some of the consequences of the fact that $(S^2 \times S^1)\#(S^2 \times S^1)$ is a pairing manifold in two ways, in the family of 3d quantum double models.

We have placed a $^\star$ next to sections and items that refer to gapped boundaries. A reader uninterested in gapped boundaries may skip these items without losing the logical flow of the paper.

### 1.1.2  Why Kirby's torus trick?

Finally, we explain the somewhat mysterious statement that our method is an analog of Kirby's torus trick. The first question is *what is the torus trick?* The general idea of using such an immersion to pull back structure from $\mathbb{R}^n$ to another topological manifold is sometimes called the *Kirby's torus trick* [17]. In Kirby's work, this idea is used to pull back smooth or piecewise-linear structures. In [18] this idea was used to pull back local Hamiltonians for invertible phases. Hastings [18] and Freedman [25,26] also used this idea to pull back Quantum Cellular Automata.

We pull back the information about the universal property of the gapped phases encoded in the quantum (ground) states, and therefore, it can be thought of as a quantum analog of Kirby's torus trick [17]. We use immersion to pull back structures from a topologically trivial region to closed manifolds, e.g., the torus. More specifically, we first construct quantum states on immersed punctured manifolds and then heal the puncture. The structures associated with the topologically trivial region understood this way include the pairing matrix, best defined on the pairing manifolds, as well as various consistency relations derived by making use of the pairing manifolds.

The second question is: *which sections of this work contain the analog of the torus trick?* An answer is indicated in Fig. 2. It says that the trick is most relevant to §4 and §5. The reason is as follows. Section 4 solves the problem of how to pull back quantum states from a ball (or a sphere) to a closed manifold. Section 5 identifies a class of closed, connected manifolds that can provide insights into some mathematical structures, e.g., consistency relations and the braiding nondegeneracy. (The discussion of pairing matrices in Section 7 is a continuation of this analysis.)

That a canonical state on a closed manifold can be constructed with the additional instruction carried by the building blocks is a new feature. This enables us to construct a state on the punctured manifold $\mathcal{W}$, for which a "smooth" healing of puncture is possible even for intrinsic topological orders (i.e., non-invertible phases). This provides one way to overcome a difficulty about intrinsic topological orders, emphasized by Hastings [18] (we also note that we are dealing with quantum states rather than Hamiltonians).

## 2  Central pillars of entanglement bootstrap

In this section, we summarize the essential working tools of the entanglement bootstrap [4,5], as extended to three spatial dimensions in [6]. These tools are available in a general space dimension, as the previous works give a clear clue to such generalizations. As the proofs were given in previous works, we shall focus on the statements and physical explanations. In §2.2, we give an in-depth discussion of the topology of immersed regions. In §2.3, we introduce the concept of constrained fusion spaces, which will be useful later.

In appendix A, we review the notations and terminology that we use often. In particular, because we shall also discuss physical systems with gapped boundaries, we need to distinguish between gapped boundaries and *entanglement boundaries*: an entanglement boundary is a component of the boundary of a region that is not part of the gapped boundaries.

Table 1: This table summarizes some examples of the pairing manifolds we study. The examples of $X$ or $Y$ that are not sectorizable are this color, while those that are sectorizable are this color. In the cartoons, gapped boundaries are depicted in thick black; entanglement boundaries are in blue here. In the two rightmost columns, the support of the excitation is in red; in grey is a possible support of the flexible operator that creates the excitations. The icing on the bundt cake is decorative.

| context | pairing manifold, $\mathcal{M}$ | $X$ | $Y$ | excitation detected by $X$ | excitation detected by $Y$ |
|---|---|---|---|---|---|
| 2d bulk | $T^2$ | annulus | annulus | particle pair | particle pair |
| 2d bulk | genus-$g > 1$ Riemann surface | $g$-hole disk | $g$-hole disk | $g + 1$-particle cluster | $g + 1$-particle cluster |
| 2d gapped boundary | cylinder with two identical gapped boundaries | half-annulus | boundary annulus | boundary particle pair | bulk anyon condensed on boundary |
| 2d gapped interface | striped sphere | n-shape | interface annulus | interface particle pair | bulk anyon condensed on interface |
| 2d gapped interface | striped torus | x-striped annulus | y-striped annulus | interface particle pair | $P - Q$ anyon pair |
| 3d bulk | $S^2 \times S^1$ | solid torus | sphere shell | loop | particle pair |
| 3d bulk | $\underbrace{(S^2 \times S^1) \# \cdots \# (S^2 \times S^1)}_{g \text{ times, } g > 1}$ | genus-$g$ handlebody | ball minus $g$ balls $\equiv \mathcal{B}_g$ | genus-$g$ graph excitation | $g + 1$-particle cluster |
| 3d bulk | $(S^2 \times S^1) \# (S^2 \times S^1)$ | ball minus torus | ball minus torus | loop-particle cluster | loop-particle cluster |
| 3d gapped boundary | sphere shell with two gapped boundaries | solid cylinder | boundary sphere shell | boundary string excitation | bulk anyon condensed on boundary |
| 3d gapped boundary | solid torus with gapped boundary | bundt cake | half sphere shell | string excitation ending on the boundary | boundary particle pair |
| 3d gapped boundary | genus-$g$ handlebody with gapped boundary | genus-$g$ bundt cake | $\mathcal{B}_g$ cut in half | $g$-arch bridge excitation | $g + 1$ boundary particles |
| 4d bulk | $S^3 \times S^1$ | 3-sphere shell, $B^1 \times S^3$ | $B^3 \times S^1$ | particle | 2-brane |
| 4d bulk | $S^2 \times S^2$ | $S^2 \times B^2$ | $B^2 \times S^2$ | loop | loop |

## 2.1 Axioms and existing tools

The entanglement bootstrap axioms can be stated in an arbitrary dimension. In the bulk, we always have two axioms. For concreteness, we start by stating the axioms for the entanglement bootstrap in three dimensions (Eq. (2)). We assume given a *reference state* $\sigma$ supported on a large ball $\mathbf{B}^3$. (Note the distinction between $\mathbf{B}^n$ and an arbitrary ball $B^n$. We shall denote *the* large ball with the reference state as $\mathbf{B}^n$ for $n$-dimensional entanglement bootstrap.) We assume that the reference state $\sigma$ lives on a many-body quantum system whose Hilbert space is the tensor product of local onsite Hilbert spaces $\mathcal{H}_{\text{total}} = \otimes_i \mathcal{H}_i$. This assumption means that we restrict our attention to systems made of bosons.[7] We further assume that each local Hilbert space $\mathcal{H}_i$ is finite-dimensional. This is for the purpose of avoiding possible exotic cases that arise only in infinite dimensions.

For the purposes of the present paper, we also assume that the sites $i$ hosting the local Hilbert spaces are the 0-cells of a cell decomposition of the large ball. This means that we have a set of boundary maps specifying which sites are in the boundary of each link (1-cell), and which links are in the boundary of each face (2-cell), and so on in higher dimensions, in such a way that these maps are nilpotent. This data allows us to speak meaningfully about the topology of regions without taking any continuum limit.

We assume that the following two axioms hold on ball-shaped subsystems contained in $\mathbf{B}^3$, for the reference state $\sigma$:

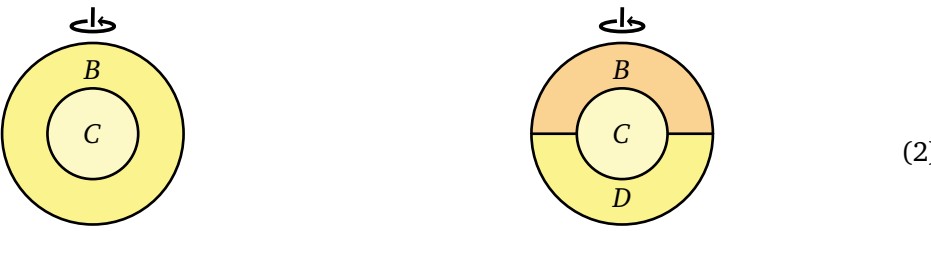

$$\textbf{A0}: \ (S_{BC} + S_C - S_B)_\sigma = 0, \qquad \textbf{A1}: \ (S_{BC} + S_{CD} - S_B - S_D)_\sigma = 0, \tag{2}$$

where $(S_X)_\sigma = -\text{Tr}(\sigma_X \ln \sigma_X)$ is the von Neumann entropy of the state $\sigma$ reduced to the region $X$. Each partition of the balls shown in Eq. (2) is topologically equivalent to the volume of revolution of the indicated 2d region.

We shall denote the entropy combinations appearing in the axioms as

$$\Delta(B, C) \equiv S_{BC} + S_C - S_B, \quad \Delta(B, C, D) \equiv S_{BC} + S_{CD} - S_B - S_D. \tag{3}$$

We shall refer to the two axioms as **A0** and **A1**. As in the 2d case [4], the strong subadditivity (SSA) [27] implies that if the axioms hold for balls of a certain length scale,[8] the same conditions must hold for all larger balls. In other words, there is no problem zooming out to a larger length scale. Because of this, we shall assume that we have a fine enough lattice, and we only consider large enough regions consisting of enough (but finite) lattice sites and having sufficient distance separation between each other. This continuum limit allows us to borrow

---

[7]Fermionic systems have a $\mathbb{Z}_2$-graded Hilbert space instead.

[8]We remark that the balls are topological balls. They do not need to be round, as they are a collection of sites whose topology is associated with a cell complex. We only require that the region is topologically equivalent to the one shown in Eq. (2). (Physically speaking, the sites in $B$ and $D$ may be thought of as coarse-grained sites so that $B$ and $D$ are thicker than the correlation length.) We introduced the volume of revolution in figures only for the visualization of the topology, and rotational symmetry is not required.

topology concepts such as ball, annulus, and sphere.[9,10]

The axioms **A0** and **A1** can be defined in an arbitrary space dimension $n$. For axiom **A0**, $B$ will be the sphere shell, and for axiom **A1**, $BD$ will be the sphere shell, where $B$ and $D$ are hemisphere shells. We shall study the generalizations of the axioms to systems with gapped boundaries as well, see §3.2 for the axioms in that setup.

The axioms **A0** and **A1**, are closely related to two well-known quantities, respectively, the mutual information $I(A : C) \equiv S_A + S_C - S_{AC}$ and the conditional mutual information $I(A : C|B) \equiv S_{AB} + S_{BC} - S_B - S_{ABC}$. SSA refers to the statement that $I(A : C|B) \geq 0$ for any tripartite mixed state. If a tripartite state $\rho_{ABC}$ satisfies $I(A : C|B)_\rho = 0$, i.e., if it saturates the strong subadditivity, we say it is a quantum Markov state (with respect to this partition).

One important object that can characterize various nontrivial structures is the information convex set, denoted as $\Sigma(\Omega)$. It depends on the region $\Omega$ and on the reference state $\sigma$. We suppress the dependence of $\sigma$ in the notation $\Sigma(\Omega)$ since the reference state is fixed at the beginning. There are equivalent ways to define $\Sigma(\Omega)$. One intuitive definition is: $\Sigma(\Omega)$ *is the set of density matrices on $\Omega$ which can be smoothly extended to any larger regions $\Omega'$ ($\Omega'$ regular homotopic to $\Omega$ and $\Omega' \supset \Omega$), where the state on $\Omega'$ is locally indistinguishable with the reference state.*[11] As the name suggests, $\Sigma(\Omega)$ is a convex set of density matrices. The region $\Omega$ can be immersed, as we shall explain.

From the axioms **A0** and **A1**, properties of information convex sets can be proved as theorems. Results appearing in previous literature [4–6] include:

- **Merging technique:** It includes the *merging lemma* [28] and the *merging theorem* [4]. When we use the word merging as a physical process, we always refer to the process described by the merging lemma [28]. That is, it is possible to construct a unique quantum Markov state $\tau_{ABCD}$ (with $I(A : D|BC)_\tau = 0$) from two quantum Markov states $\rho_{ABC}$ (with $I(A : C|B)_\rho = 0$) and $\lambda_{BCD}$ (with $I(B : D|C)_\lambda = 0$), as long as $\rho_{BC} = \lambda_{BC}$. The resulting "merged" state $\tau_{ABCD}$ is identical with $\rho_{ABC}$ and $\lambda_{BCD}$ on marginals $ABC$ and $BCD$.

  In entanglement bootstrap, merging theorem [4, 5] says that whenever two quantum Markov states in two information convex sets can merge,[12] the resulting state must be an element of a third information convex set.

- **Immersed region:** The concept of immersed region [6, 16] is a natural generalization of subsystems. An immersed region is locally embedded and has the same dimension as the physical system. For each immersed region, one can consider the information convex set. Since immersed regions will be crucial in this work, we shall introduce them further in §2.2.

---

[9]The mathematical theory for going from the lattice to the continuum topology is nontrivial, but will not be needed for our purposes. It is unknown if an existing branch of mathematics can formulate this idea with full rigor. This is a meaningful topic for future studies.

[10]In realistic settings, and in particular for gapped chiral phases, these axioms may not be exact. Known models of gapped chiral phases all have finite (nonzero) correlation length, at least when the local Hilbert space is finite dimensional. Nonetheless, we believe that they are satisfied in a renormalization group sense for a large class of physical systems. That is, the violation of the axioms decay towards zero (at a fast enough speed) as we coarse grain the lattice further. Justification of this conjecture is an interesting future problem.

[11]A state on an immersed region is locally indistinguishable with the reference state if the reduced density matrix on any (small) embedded ball is identical to that of the reference state. This is equivalent to two other definitions that appeared in [4] by the isomorphism theorem.

[12]The regions in question need to be above a minimum (finite) thickness for the merging theorem to apply. See Theorem II.3 of [5] for a review of the statement of the merging theorem. (Concrete choices of regions can be explicitly constructed in a given coarse-grained lattice. See, e.g., (9.12) and (9.16) in Chapter 9 of [29].) This requirement is satisfied in all our applications.

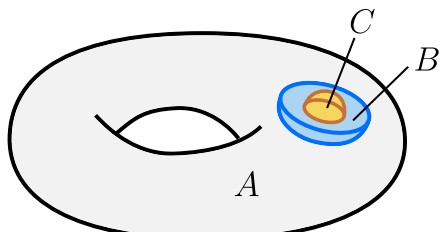

Figure 3: An illustration of an elementary step. Here the ball $C$ and added to $AB$ in a topologically trivial manner. $A$ can be large and has an arbitrary topology. $BC$ is part of a small ball $BCD$ on which the axioms are imposed. $A$ is contained in the complement of $BCD$. $AB \to ABC$ is an elementary step of extension, whereas $ABC \to AB$ is an elementary step of restriction. Axiom **A1**, on $BCD$ (with $D$ not shown such that $BD$ separate $C$ from the "outside") implies that $I(A : C|B)$ must vanish for any element of information convex set $\Sigma(ABC)$.

- **The isomorphism theorem:** If two immersed regions are connected by a *path*, then the information convex sets are isomorphic. In other words,

$$\Omega_0 \sim \Omega_1 \quad \Rightarrow \quad \Sigma(\Omega^0) \cong \Sigma(\Omega^1). \tag{4}$$

  Here, a path between $\Omega_0$ and $\Omega_1$ is a finite sequence of immersed regions $\{\Omega^t\}$, $t \in \{0, 1/N, 2/N, \cdots, 1\}$ such that adjacent regions in the sequence are related by adding (removing) a small ball in a topologically trivial manner. This relation is an elementary step; see Fig. 3 for an illustration. In the remaining of the paper, when two regions are connected by a path, we say one region can be "smoothly" deformed into another. The isomorphism $\cong$ refers to the isomorphism of two information convex sets as convex sets, as well as the preservation of distance measures and the entropy difference. (We do not require the entropy to be preserved, only the entropy difference.) The isomorphism $\cong$ between the two information convex sets can depend on the path. Nonetheless, it can be shown that only the topological class of the path matters.

- **Structure theorems:** The geometry of information convex sets must be of a certain form. While the isomorphism theorem says that we only need to consider topological classes of regions, the structure theorems describe powerful statements on the topological dependence:

  1. *Simplex theorem:* If an immersed region $S$ is *sectorizable*,[13] the information convex set is a simplex:

$$\Sigma(S) = \left\{ \sum_{I \in \mathcal{C}_S} p_I \rho_S^I \,\middle|\, \sum_I p_I = 1, p_I \geq 0 \right\}. \tag{5}$$

     Here $\mathcal{C}_S$ is the (finite) set of labels, where each label represents a superselection sector. The set of density matrices $\{\rho_S^I\}$ are the extreme points, and they are mutually orthogonal.

  2. *Hilbert space theorem:* For any immersed region $\Omega$, the information convex set $\Sigma(\Omega)$ is a convex hull of mutually orthogonal convex subsets $\Sigma_I(\Omega)$, where $I \in \mathcal{C}_{\partial\Omega}$. (Here $\partial\Omega$ is the thickened entanglement boundary[14] of $\Omega$, and it is always sectorizable.)

---

[13]A region $X$ is said to be sectorizable if it contains two disjoint pieces $X'$ and $X''$ such that each can be deformed to $X$ via a sequence of extensions.

[14]A thickened entanglement boundary is always of the form $m \times \mathbb{I}$, where $m$ is a manifold and $\mathbb{I}$ is an interval. In this work, we always consider thickened entanglement boundaries that are thick enough so that the interval $\mathbb{I}$, though being a lattice analog, can be partitioned into smaller intervals.

Each subset is isomorphic to the state space (i.e., the set of density matrices) of a finite-dimensional Hilbert space $\mathbb{V}_I(\Omega)$. We denote this by $\Sigma_I(\Omega) \cong \mathcal{S}(\mathbb{V}_I(\Omega))$, where $\mathcal{S}(\mathbb{V})$ denotes the state space of $\mathbb{V}$. The dimension of the Hilbert spaces, denoted as $\{N_I(\Omega) \equiv \dim \mathbb{V}_I(\Omega)\}$, are the fusion multiplicities. We refer to the finite dimensional Hilbert spaces $\mathbb{V}_I(\Omega)$ as fusion spaces. (The origin of this name is the case where $\Omega$ is the 2-hole disk, and these numbers are associated with the fusion of two anyons into a third.)

We comment that the Simplex Theorem can be understood as a special instance of the Hilbert space theorem, whether the fusion multiplicities are either 0 or 1.

- **Associativity theorem:** If an immersed region $\Omega$ can be cut into halves by a hypersurface that does not touch the entanglement boundary of $\Omega$, then the fusion space dimensions associated with $\Omega$ are completely characterized by the multiplicities of its subsets ($\Omega_L$ and $\Omega_R$) and the way the hypersurface connects them:

$$\dim \mathbb{V}^{a_R}_{a_L}(\Omega) = \sum_{i \in \mathcal{C}_S} \dim \mathbb{V}^i_{a_L}(\Omega_L) \cdot \dim \mathbb{V}^{a_R}_i(\Omega_R), \tag{6}$$

where $S$ is the thickened hypersurface, and $a_L \in \mathcal{C}_{A_L}$ and $a_R \in \mathcal{C}_{A_R}$. Here $A_L = \Omega_L \cap \partial\Omega$ and $A_R = \Omega_R \cap \partial\Omega$. It is guaranteed that $S$ is sectorizable and $\mathcal{C}_S$ denotes the set of superselection sectors on $S$. (Alternatively, one can write: $N^{a_R}_{a_L}(\Omega) = \sum_{i \in \mathcal{C}_S} N^i_{a_L}(\Omega_L) N^{a_R}_i(\Omega_R)$.) This theorem [6] is proved by considering the merging of whole boundary components and making use of the Hilbert space theorem.

- **Vacuum and sphere completion:** Let $\Omega \subset \mathbf{B}^n$ be a subsystem of the ball. The vacuum state on $\Omega$ is defined as $\sigma_\Omega$, i.e., the reduced density matrix of the reference state. It is shown that the vacuum state is an isolated extreme point of $\Sigma(\Omega)$ [6]. If $S$ is a sectorizable region embedded in $\mathbf{B}^n$, then $\rho^1_S \equiv \sigma_S$ corresponds to a very special label in $\mathcal{C}_S$, i.e., the *vacuum sector*, denoted as $1 \in \mathcal{C}_S$.

Moreover, a reference state on a sphere $\mathbf{S}^n$ always has a vacuum, that is the unique pure state in $\Sigma(\mathbf{S}^n)$ [4]. A vacuum state on a sphere can be obtained from one on the ball by the "sphere completion" [6]. We shall generalize the completion trick in later sections.

To what extent can the definition of vacuum generalize to immersed regions that are not embedded? This largely remains an open problem, but we report some progress in §4.1.

- **Quantum dimension:** For a sectorizable region $S$ embedded in $\mathbf{B}^n$, the quantum dimension of a sector $I$ can be defined as

$$d_I = \exp\left(\frac{S(\rho^I_S) - S(\sigma_S)}{2}\right), \quad \forall I \in \mathcal{C}_S. \tag{7}$$

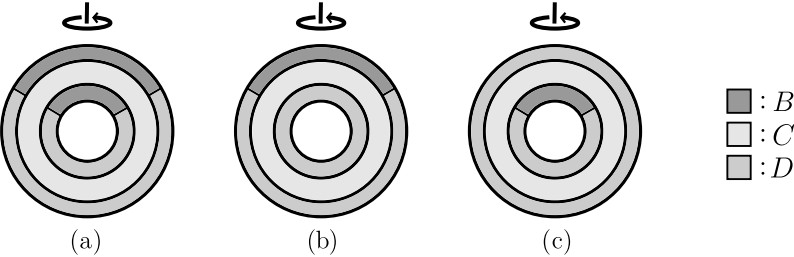

Figure 4: Partitions of sphere shells relevant to the quantum dimension of point excitations.

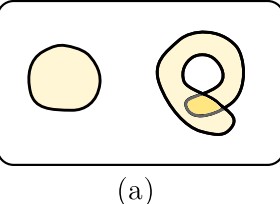
(a)
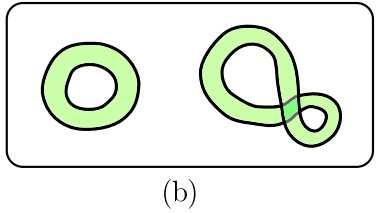
(b)
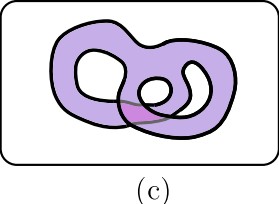
(c)

Figure 5: Illustration of the three kinds of immersed regions.

We shall need another definition of quantum dimension for point-like excitations, which are characterized by the information convex set of a sphere shell. Let the sphere shell be $S^{n-1} \times \mathbb{I} = BCD$, where $\mathbb{I} = [0,1]$ is an interval. $S^{n-1} = NS$, where $N$ is the northern hemisphere and $S$ is the southern hemisphere. Let $C = S^{n-1} \times [1/3, 2/3]$, $B = (N \times \mathbb{I}) \setminus C$ and $D = (S \times \mathbb{I}) \setminus C$. This partition for the case $n = 3$ is shown in Fig. 4(a). Then, the quantum dimension for point excitation $a$ is

$$d_a = \exp\left( \frac{\Delta(B,C,D)_{\rho^a}}{4} \right), \quad a \in \mathcal{C}_{S^{n-1} \times \mathbb{I}}, \quad \text{for Fig. 4(a).} \tag{8}$$

If instead, $C = S^{n-1} \times [1/3, 2/3]$, $B = N \times [2/3, 1]$ (or $B = (N \times [0, 1/3]$ ) and $D$ is the rest, the quantum dimension is[15]

$$d_a = \exp\left( \frac{\Delta(B,C,D)_{\rho^a}}{2} \right), \quad a \in \mathcal{C}_{S^{n-1} \times \mathbb{I}}, \quad \text{for Fig. 4(b) and (c).} \tag{9}$$

If we know $\Delta(B,C,D)_{\rho^a}$ for any one of the three partitions, we have a definition of quantum dimension for point particles. This alternative definition has an advantage. We only need the state $\rho^a$ on the sphere shell, and there is no need to compare it with the vacuum. This makes Eq. (8) and Eq. (9) work for immersed sphere shells. Moreover, there are generalizations of them for many other excitation types [6]. Moreover, from Eq. (8) and Eq. (9) it is manifest that $d_a \geq 1$. This is because $\Delta(B,C,D)$ is nonnegative.

## 2.2 Immersed regions

The concept of *immersed region* is a natural generalization of subsystems. A subsystem is an embedded region that has the same dimension as the physical system. An immersed region is a region that is *locally* embedded and has the same dimension as the physical system. By this definition, embedding is a special case of immersion.

Immersed regions come in three kinds (see Fig. 5):

1. Regions of the first kind can always be smoothly deformed to embedded regions. An example is a ball in any dimension; see Fig. 5(a).

2. Regions of the second kind have inequivalent[16] immersions into the physical system. One is an embedding, and at least one other is a nontrivial immersion. One such example is the annulus in 2d; the figure eight at right in Fig. 5(b) cannot be deformed by regular homotopy to the embedded annulus at left because their boundaries have different winding numbers.

---

[15]The technically nontrivial part of the claim in Eq. (9) is that the partitions in Fig. 4(b) and (c) are related by turning the sphere shell inside-out. The proof is written by one of us; see Appendix A of [30]. (Note however, in 3d, this is intuitively related to the fact that any sphere shell immersed in a 3-dimensional ball can be turned inside out.)

[16]We say two immersions are inequivalent if no path can connect the pair of regions. In the continuous limit, this is the statement that the two immersions are not regular homotopy equivalent.

3. Regions of the third kind cannot be embedded in a ball. One example is a torus with a ball removed; see Fig. 5(c).

In particular, because of the existence of the third kind, immersed regions are strictly more diverse than embedded regions.

Below, we give a few examples of immersed regions, focusing on the topology that cannot be obtained from regions embedded within a ball. Let $\mathcal{W} \equiv \mathcal{M} \backslash B^n$. Here $\mathcal{M}$ is a $n$-dimensional closed manifold and $B^n$ is a solid ball. In other words, $\mathcal{W}$ is a closed manifold with a ball removed. Here are some choices of such $\mathcal{W}$, which can be immersed in a ball $\mathcal{W} \looparrowright B^n$, for $n = 2, 3$:

1. In 2d, any closed connected orientable surface (classified by genus-$g$) with a ball removed is a choice of $\mathcal{W}$. It is not hard to visualize these immersions [31].

2. In 3d, it is shown recently that every closed connected orientable 3-manifold can be immersed in a ball upon removing a ball [32]. In particular, the immersions of the following manifolds[17] are known explicitly.

   (a) $(S^2 \times S^1) \backslash B^3$.

   (b) $\underbrace{(S^2 \times S^1) \# \cdots \# (S^2 \times S^1)}_{g \text{ times}} \backslash B^3 \equiv \#g(S^2 \times S^1) \backslash B^3$. Here $\#$ means connected sum.[18]

   (c) $T^3 \backslash B^3$, where $T^3$ is the three-dimensional torus. This case is nontrivial to visualize. In the math literature, the existence proof of such immersion appeared first [33], and was generalized to arbitrary dimensions later on [34, 35]. Several explicit constructions followed, see [36, 37] for example.

We shall describe other examples of immersion in §3; some of these examples are related to the physical setup of gapped boundaries. Furthermore, in §4, we shall find ways to obtain reference states on various closed manifolds with the combination of two powerful techniques: immersion and the completion trick, which we shall discuss. For later convenience, we shall always consider immersed regions that leave enough space so that we can thicken them while keeping them immersed.

## 2.3 Constrained fusion space

We have discussed information convex set $\Sigma(\Omega)$ and the subsets $\Sigma_I(\Omega)$ in which the sectors ($I$) on the thickened entanglement boundaries are specified. One motivation for studying these sets is to characterize how quantum information is distributed in subsystems of a quantum state.

For this purpose, it is sometimes useful to specify a constraint that is not on the thickened entanglement boundary. This motivates us to define information convex sets with constraints, and their constrained fusion spaces (or constrained Hilbert space).

**Definition 2.1** (Constrained information convex sets and constrained fusion spaces)**.** Let $\Omega = AE$ and choose $\rho_A^\kappa \in \text{ext}(\Sigma(A))$, the set of extreme points of $\Sigma(A)$. We define the constrained information convex set as

$$\Sigma_{[\kappa_A]}(\Omega) \equiv \{\rho_\Omega \in \Sigma(\Omega) | \text{Tr}_E \rho_\Omega = \rho_A^\kappa\}, \quad \text{and} \quad \Sigma_{I[\kappa_A]}(\Omega) \equiv \{\rho_\Omega \in \Sigma_I(\Omega) | \text{Tr}_E \rho_\Omega = \rho_A^\kappa\}. \quad (10)$$

---

[17]We shall extensively use the standard notation for manifold topology, e.g., $S^n$ refers to $n$-sphere, $T^n$ refers to $n$-dimensional torus; see Appendix A for other notations we frequently use in this work.

[18]A connected sum of two $n$-dimensional manifolds is also a $n$-dimensional manifold. It is formed by first deleting a ball in the interior of each manifold and then gluing together the resulting boundary spheres.

According to Lemma 2.2 below, $\Sigma_{I[\kappa_A]}(\Omega) \cong \mathcal{S}(\mathbb{V}_{I[\kappa_A]})$, for some fusion space $\mathbb{V}_{I[\kappa_A]}(\Omega)$. We shall refer to $\mathbb{V}_{I[\kappa_A]}(\Omega)$ as a constrained fusion space.

**Remark.** We further allow the usage of $[\rho_A^\kappa]$ as an alternative for $[\kappa_A]$, where $\rho_A^\kappa$ is the extreme point of $\Sigma(A)$ in question. Constrained fusion spaces are particularly convenient for the study of closed manifolds. When $\Omega = \mathcal{M}$ is a closed manifold the index $I$ is dropped, we have constrained fusion spaces $\mathbb{V}_{[\kappa_A]}(\mathcal{M})$.

**Lemma 2.2.** *The various constrained sets defined in Definition 2.1 satisfy:*

1. *$\Sigma_{[\kappa_A]}(\Omega)$ and $\Sigma_{I[\kappa_A]}(\Omega)$ are compact convex sets.*

2. *$\Sigma_{I[\kappa_A]}(\Omega) \cong \mathcal{S}(\mathbb{V}_{I[\kappa_A]}(\Omega))$, where $\mathbb{V}_{I[\kappa_A]}(\Omega)$ is a finite-dimensional Hilbert space, and it is a subspace of $\mathbb{V}_I(\Omega)$.*

3. *If $A$ is sectorizable, we can relabel $\kappa$ as $J \in \mathcal{C}_A$. In this case,*

$$\mathbb{V}_I(\Omega) = \oplus_J \mathbb{V}_{I[J_A]}(\Omega), \tag{11}$$

   *is a direct sum.[19]*

*Proof.* The proof of statement 1 is as follows. It is evident that $\Sigma_{[\kappa_A]}(\Omega)$ and $\Sigma_{I[\kappa_A]}(\Omega)$ are convex sets. The compactness follows from that of $\Sigma(\Omega)$. In more detail, the convex set $\Sigma(\Omega)$ is compact, and the subsets are defined by a set of constraints that are linear equalities.

To prove statement 2, we can shrink $A$ a little bit, without changing its topology. Let $\widetilde{A} = A \setminus \partial A$ and $\widetilde{E} = E \cup \partial A$ (recall that $E \equiv \Omega \setminus A$). Then $\widetilde{A}$ and $\widetilde{E}$ share an entire boundary. The extreme point label $\kappa$ on $A$ induces a label $\Phi(\kappa)$ of the extreme points of $\Sigma(\partial A)$. This follows from the structure theorem for the information convex set. Note that $\partial A$ is a sectorizable region. While $\kappa$ may be a continuous label, $\Phi(\kappa)$ must be a discrete finite set. The state is a Markov state on $\widetilde{A}, \partial A, E$ whose marginal on $A$ is $\rho_A^\kappa$. Thus, any element of $\Sigma_{I[\kappa_A]}(\Omega)$ is obtained by merging some element of $\Sigma_{I\Phi(\kappa)}(\widetilde{E})$ with *the same* state $\rho_A^\kappa$. Thus, $\Sigma_{I[\kappa_A]}(\Omega) \cong \mathcal{S}(\mathbb{V}_{I\Phi(\kappa)}(\widetilde{E}))$. Thus, statement 2 is true, and furthermore $\dim \mathbb{V}_{I[\kappa_A]}(\Omega) = \dim \mathbb{V}_{I\Phi(\kappa)}(\widetilde{E})$.

Statement 3 is a corollary of statement 2. To see this, we observe that density matrices in $\Sigma_I(\Omega)$, which reduces to the extreme points of $\Sigma(A)$ associated with $J, J' \in \mathcal{C}_A$ must be orthogonal, for $J \neq J'$. This follows from the monotonicity of fidelity and that $F(\rho_A^J, \rho_A^{J'}) = 0$ for $J \neq J'$. Second, if we take all different choices of $J$, the right-hand side of Eq. (11) gives the Hilbert space dimension that matches that for the left-hand side. $\qquad \square$

With the language of constrained Hilbert space, we can usefully refine the Associativity Theorem as:

**Theorem 2.3** (Associativity theorem, new form)**.** *Let $S$ be the thickened hypersurface that appeared in the setup of the Associativity Theorem (see around Eq. (6)), then*

$$\dim \mathbb{V}_{a_L[i_S]}^{a_R}(\Omega) = \dim \mathbb{V}_{a_L}^i(\Omega_L) \cdot \dim \mathbb{V}_i^{a_R}(\Omega_R), \quad \forall i \in \mathcal{C}_S. \tag{12}$$

# 3 Regions dual to excitations

In this section, we discuss various regions that can detect either an excitation or a cluster of excitations. As reviewed in §2.1, immersed regions can be classified into sectorizable regions and non-sectorizable regions. A sectorizable region has an information convex set isomorphic

---

[19] A Hilbert space is a direct sum when the subspaces in the sum are mutually orthogonal.

to a simplex; the extreme points correspond to the superselection sectors. For non-sectorizable regions, information about fusion spaces can be detected; in the case of a nontrivial fusion space, the information is quantum.[20]

Sometimes, we say a region can detect the superselection sector of some excitation or the fusion space associated with a cluster of excitations. When does this happen?[21] The purpose of this section is to explain this. We further provide many examples, the physical setups of which differ in the dimensions and the presence (absence) of a gapped boundary.

The basic intuition is as follows. Imagine a ground state $|\psi\rangle$ of a topologically ordered system, on a sphere. Consider an excited state $|\varphi\rangle$ with a few excitations on the sphere. The state $|\varphi\rangle$ on a region $\Omega$ separated from the excitation(s) by a few correlation lengths must be locally indistinguishable from the ground state. Roughly speaking, the region $\Omega$ is the complement of the excitations. If the excitations are nontrivial, that is, if the excitations cannot be created by local operators, the region $\Omega$ must be able to detect them.

In entanglement bootstrap, this intuition is guaranteed. The reference state plays the role of the ground state. The reduced density matrix of $|\varphi\rangle$ on $\Omega$ must be an element of the information convex set $\Sigma(\Omega)$. From the "sphere completion lemma" of Ref. [6], we can always construct a reference state on the sphere ($S^n$), where the axioms **A0** and **A1** are satisfied everywhere. (We shall not review the sphere completion technique here. Instead, we review it after proving a more powerful version Lemma 4.4; see Fig. 21.) In the context of gapped boundaries, as we discuss in §3.2, the analog of sphere completion is the fact that we can always define a reference state on a ball with an entire gapped boundary, which we denote as $\underline{\underline{B}}^n$. String operators and membrane operators creating the excitations can also be constructed in entanglement bootstrap, and the properties of these operators can be useful in proving things; see [16]. In this work, we will not rely on string or membrane operators to prove any statement, but as they provide complementary intuition, we sometimes draw them in figures. They will be discussed more explicitly in [38].

All regions considered in this section *can* be embedded in a ball $B^n$ (or $S^n$, or $\underline{\underline{B}}^n$ in the presence of a gapped boundary). However, we also consider the immersed version of these regions. They will be useful later, as building blocks (see §4.1). We start with the 2d bulk and 3d bulk. Then we discuss the boundary generalizations.

## 3.1 2d bulk and 3d bulk regions

The setup and axioms of the 2d bulk and 3d bulk have been discussed in §2.1. The intuition to keep in mind is that the combination of the two axioms (especially axiom **A1**) makes it possible to deform the regions smoothly: if two regions can be connected by a path, i.e., if two regions are related by a regular homotopy, then the information convex sets associated with the pair of regions are isomorphic. Thus, we shall only be interested in the topological class of immersed regions.

### 3.1.1 2d bulk regions

The 2d bulk is the physical context of anyons and topological orders, and it is the context in which 2+1d TQFT [2, 3, 39] is expected to apply. As studied in detail in [4], there is a finite

---

[20]When we say the information is quantum, we mean that the information cannot be copied. This happens when we have a fusion space of at least 2 dimensions.

[21]The question becomes nontrivial when the region is not embedded in a ball or a sphere. The thickened Klein bottle in 3d is one example where we don't know what are the excitations it characterizes, or whether these excitations can be identified as excitations living in a sphere.

set of basic regions: the disk, the annulus, and the pair of pants.

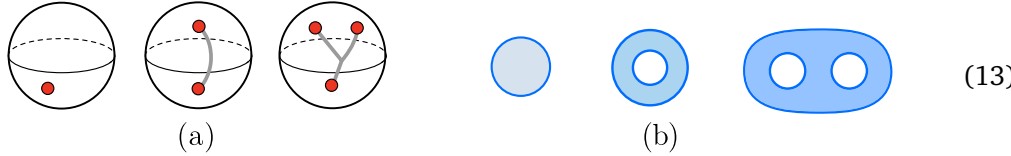 (13)

(a)       (b)

These regions are closely related to anyons and the string operators creating them; see Eq. (13). First, the regions are homeomorphic to the complement of the anyon excitations on $S^2$. The string operators pass through the regions and cut them into disks. Note that a single excitation on the sphere always carries a trivial superselection sector.

In the topology literature, it is well-known that any compact orientable surface can be constructed by gluing these basic topology types along closed curves. This is known as the pants decomposition [40]. This intuition is known in the framework of topological quantum field theory (TQFT), see e.g. [3] and [39]. In §4.1, we will construct reference states on more interesting manifolds by building them from pieces of a state on a topologically trivial region. To achieve that, we shall make use of immersed versions of these regions. To gently prepare the reader for this discussion, we illustrate immersed versions of regions in Fig. 6.

Even for the simple class of embedded regions, e.g., a sphere minus $k$-balls, nontrivial immersed versions exist. If we only allow deformations within $\mathbf{B}^2$, the immersed regions described in Fig. 6 are not regular homotopic to embedded regions. This is denoted as $\Omega \overset{\mathbf{B}^2}{\nsim} \Omega'$. Nonetheless, they *are* regular homotopic to embedded regions on the background manifold $\mathbf{S}^2$, (denoted as $\Omega \overset{\mathbf{S}^2}{\sim} \Omega'$). In other words, the deformation is more flexible on $\mathbf{S}^2$ compared with $\mathbf{B}^2$. This is one of the reasons why the notion of manifold completion, detailed in §4, is useful.

### 3.1.2    3d bulk regions

The 3d bulk provides more diverse topology types. One simple class of regions is genus-$g$ handlebodies; see Fig. 7. Note that it is already an infinite series. While there are other interesting topologies, we remark that the seemingly simple list of handlebodies is surprisingly fundamental: every closed compact orientable 3-manifold can be divided into two handlebodies; this is known as Heegaard splitting [41]; see Fig. 8 for an illustration. (The Heegaard genus, which is the smallest number of handles of a Heegaard splitting, is additive under connected sums of 3-manifolds. Therefore, all genus-$g$ handlebodies are needed to construct an arbitrary closed compact orientable 3-manifold in this approach.)

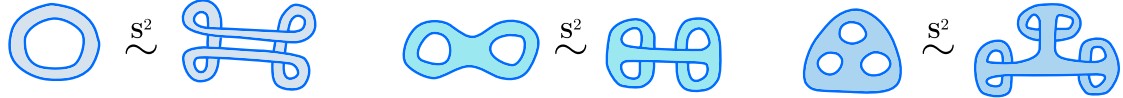

Figure 6: Immersed versions of embedded regions. For every example shown here, there exists a smooth deformation that connects the two configurations. The deformation is done on $\mathbf{S}^2$. Note that the deformation is more flexible on $\mathbf{S}^2$ compared with $\mathbf{B}^2$.

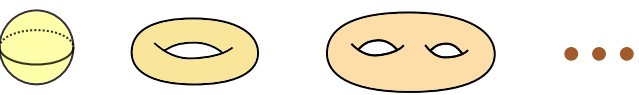

Figure 7: Genus-$g$ handlebodies, where $g = 0, 1, 2, \dots$

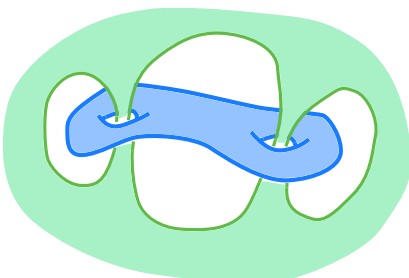

Figure 8: A Heegaard splitting of a 3-sphere, inspired by [41].

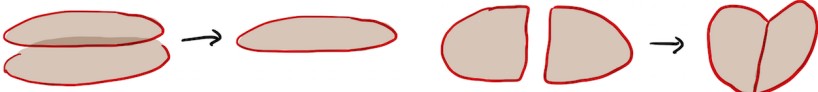

Figure 9: Two ways to fuse flux loops: (left) along the whole loop or (right) along a segment. The latter produces a genus-two graph excitation.

Interestingly, all these handlebodies are sectorizable. The excitations detected by these regions are supported on thin handlebodies of the same genus. Thin handlebodies look like graphs, and for this reason, we call these excitations *graph excitations*. For genus $g = 1$, the graph excitations are detected by the solid torus; they are familiar in the literature, and known as pure flux loops; see, e.g., [6–8, 11].

How about graph excitations with a higher genus? Are they intrinsically new? Shouldn't they be labeled by two flux loops as is suggested by the fusion process of loops, illustrated in Fig. 9? As it turns out, the answer is 'no' in general. For instance, the $G = S_3$ quantum double model has 3 flux sectors and $11 > 3 \times 3$ graph excitations[22] for genus $g = 2$. Here $S_3$ is the permutation group of three elements. This is illustrated in the example of the 3-dimensional $S_3$ quantum double model below.

Here are the various superselection sector labels for 3d topological orders, which we shall use in Example 3.1 and later. We use the same notation as Ref. [6]. $\mathcal{C}_{\text{point}} = \{1, a, b, \cdots\}$ refers to the superselection sectors of point particles detected by the sphere shell. $\mathcal{C}_{\text{flux}} = \{1, \mu, \nu\}$ is the set of pure flux loops, i.e., graph excitations supported on $g = 1$ graphs. Below, we shall use $\mathcal{C}_g$ to denote the superselection sectors for the graph excitations on an unknotted genus $g$ graph; they are detected by a genus $g$ handlebody.

**Example 3.1** (3d $S_3$ quantum double). The finite group $S_3$ is the smallest non-abelian group: $S_3 = \{1, r, r^2, s, sr, sr^2 | r^3 = s^2 = 1, sr = r^2 s\}$. The quantum double with $S_3$ group has the following superselection sectors:

1. $\mathcal{C}_{\text{point}}$ contains 3 labels, with $\{d_a\} = \{1, 1, 2\}$.

2. $\mathcal{C}_{\text{flux}}$ contains 3 labels, with $\{d_\mu\} = \{1, \sqrt{2}, \sqrt{3}\}$.

3. $\mathcal{C}_g$ contains $6^{g-1} + 3^{g-1} + 2^{g-1}$ labels. In particular, $\mathcal{C}_{g=2}$ contains 11 labels.

The reader familiar with the 2d $S_3$ quantum double model may be disturbed by the square roots in the quantum dimensions of the fluxes. One way to see this choice is sensible is to check the matching rule $\sum_a d_a^2 = \sum_\mu d_\mu^2$.

---

[22]Among the 11 types, 6 are *genuine* graph excitations, meaning that they cannot be reduced to a single loop. This number is greater than the number $(9 - 5 = 4)$ predicted by the naive approach.

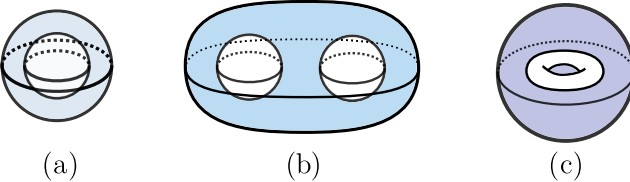

Figure 10: (a) A sphere shell. (b) A ball with two balls removed. The two removed balls are smaller, disjoint, and are within the interior of the large ball. (c) A ball with a solid (unknotted) torus removed.

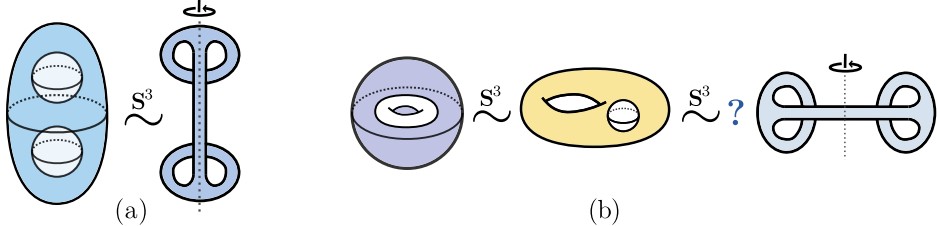

Figure 11: Examples of embedded regions in $\mathbf{S}^3$ and immersed regions obtained by a "smooth" deformation starting from embedded ones on $\mathbf{S}^3$. Whether the rightmost region can be obtained by the indicated deformation is unclear to us, and this is indicated by the question mark.

**Proposition 3.2** (Graph sectors for 3d quantum double)**.** *For 3d quantum double with finite group G, the set ($\mathcal{C}_g$) of superselection sectors of graph excitations characterized by the information convex set of genus-g handlebody has*

$$|\mathcal{C}_g| = \frac{1}{|G|} \sum_{h \in G} |E(h)|^g \,, \tag{14}$$

*elements, where $E(h)$ is the centralizer group of $h$ : $E(h) \equiv \{k \in G | kh = hk\}$. $|G|$ denotes the order of finite group G. In particular when $G = S_3, |\mathcal{C}_g| = 6^{g-1} + 3^{g-1} + 2^{g-1}$.*

The proof of Proposition 3.2 is given in Appendix C.

A few other basic topologies are shown in Fig. 10. The first one detects a particle,[23] the second one detects a three-particle cluster and the third one detects a particle-loop cluster. Only the sphere shell is sectorizable. These regions are studied in [6].

The sphere shell and $B^3$ with two balls removed, shown in Fig. 10(a) and (b) are direct analogs of the 2d region we discussed in §3.1.1. In fact, every region in Fig. 10 is the volume of revolution of a 2d region along an axis, and therefore, the dimensional reduction consideration in [6] applies. Clearly, this list is incomplete. For instance, there are regions with knotted or linked torus entanglement boundaries; some of them are studied in [6].

As with the 2d bulk, we can make immersed versions of these basic regions. Intuitively speaking, immersion can let the region "flip" along an entanglement boundary; see Fig. 11.

Connected components of thickened entanglement boundaries provide another basic class of 3d region. The topology of any such region is of the form $m \times \mathbb{I}$, where $m$ is a genus-$g$ surface and $\mathbb{I}$ is an interval. In other words, such regions are thickened genus-$g$ surfaces. Therefore, it is always possible to embed them in $\mathbf{S}^3$. Interestingly, when $g \geq 1$, it is also possible to immerse such a region in $\mathbf{S}^3$ nontrivially, in a way not regular homotopic to any embedding; see, e.g., Corollary 1.3 of Ref. [42].

---

[23]On a 3-sphere, the complement of a sphere shell is two disjoint balls. For this reason, one can say the sphere shell detects a particle-antiparticle pair.

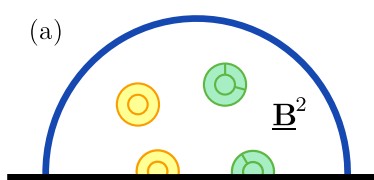
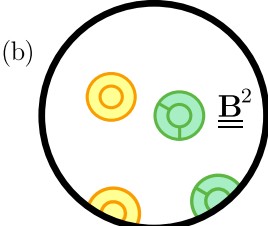

Figure 12: The setup of 2d entanglement bootstrap with a gapped boundary. One may start from a reference state on either: (a) a "half disk" adjacent to the boundary ($\underline{\mathbf{B}}^2$), or (b) a disk with an entire boundary ($\underline{\underline{\mathbf{B}}}^2$). Axioms are imposed on bounded radius disks both within the bulk and on the gapped boundary. The gapped boundary is represented by a thick black line.

## 3.2  *Systems with gapped boundaries in 2d and 3d

We shall first describe the setup and the axioms of entanglement bootstrap for the gapped boundary problems [5]. They are direct analogs of the axioms in other previously studied contexts [4,6]. After that, we describe the basic choices of regions and the excitations or fusion spaces they characterize. Although we focus here on gapped boundaries, the same technology applies to the more general case of gapped domain walls between topological phases [5].

### 3.2.1  *Axioms for gapped boundaries in 2d and 3d

**2d setup:** The entanglement bootstrap setup of the 2d gapped boundary problem is a reference state ($\sigma$) on a half disk ($\underline{\mathbf{B}}^2$) adjacent to the gapped boundary. The total Hilbert space is the tensor product of finite dimensional Hilbert spaces on lattice sites. The number of such lattice sites is a finite number but is large enough. These lattice sites make a sensible discretization of a topological manifold. One can, for example, obtain such a lattice by coarse-graining a realistic many-body system made of qudits on a manifold.[24] Each site has a finite-dimensional Hilbert space. The region with the gapped boundary is depicted in Fig. 12, where the thick black line is the gapped boundary. We assume that nontrivial Hilbert space only exists on one side of the boundary, and the other side is empty. Two boundary axioms are assumed in addition to the bulk axioms. These axioms are of similar forms, in the sense that we always partition a small region into two or three pieces, labeled by $BC$ or $BCD$:

- If we partition the region into two pieces, we call the interior $C$ and call the thickened entanglement boundary $B$, then we require $\Delta(B, C)_\sigma = 0$ on the reference state, for the indicated region (yellow disk or half-disk in Fig. 12).

- If we partition the region in three pieces, we call the interior $C$ and call the thickened entanglement boundary $BD$, where the topology of $B$ and $D$ are indicated in Fig. 12 (green regions). We require that $\Delta(B, C, D)_\sigma = 0$.

An equally simple setup is a reference state on a disk with an *entire* boundary, with analogous axioms imposed; see Fig. 12(b) for an illustration. The justification is a direct analog of the sphere completion lemma (Lemma 3.1 of Ref. [6]). See *e.g.* [43,44] for some solvable models of gapped boundaries in 2+1d and see [45] for explicit computation of information convex sets in a related context. We note that there are systems that admit only gapless boundaries, see [13,46]; we expect that the boundary axioms are violated for these systems.

We clarify the distinction between terminologies. The *gapped boundary* should not be confused with the *entanglement boundary*. The entanglement boundary is a boundary that has

---

[24]The same considerations as in footnote 10 apply here.

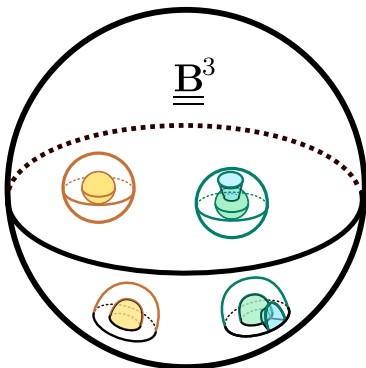

Figure 13: The setup of 3d entanglement bootstrap with a gapped boundary. We impose bulk and boundary versions of axioms **A0** and **A1** on bounded radius balls. Here the reference state is given on a 3-dimensional ball with an entire gapped boundary, denoted as $\underline{\underline{\mathbf{B}}}^3$, where the reference state is pure. (As in 2d, another reasonable starting point is a reference state on a half ball $\underline{\mathbf{B}}^3$ adjacent to a gapped boundary. The "completion trick" shows that these two starting points are equally simple.)

nontrivial physical degrees of freedom lying on both sides, and for the quantum states we are interested in, these physical degrees of freedom have entanglement across the entanglement boundary. Another perspective is that it is possible to pass the entanglement boundary with the extensions allowed by the (generalized) isomorphism theorem. When we write $\partial\Omega$ for an immersed region $\Omega$, we always mean the thickened entanglement boundary. (In the remainder of the paper, we sometimes refer to an entanglement boundary as a boundary for short, but we always say gapped boundary.)

**3d setup:** Below we describe the entanglement bootstrap setup for the gapped boundary of a 3d gapped system. It is very similar to the 2d setup, as can be seen in Fig. 13. We refer to the half-ball adjacent to the boundary as $\underline{\mathbf{B}}^3$. An alternative starting point is the (pure) vacuum state on a ball with an entire boundary $\underline{\underline{\mathbf{B}}}^3$.

**Remark.** A topologically ordered system in 2d and 3d usually has multiple gapped boundary types. In 3d, models of gapped boundaries have been studied by several authors (see e.g. [47, 48] and references therein), where we expect our axioms to apply. A complete classification of gapped boundaries of 3d topologically ordered systems is an open problem. These gapped boundary types cannot be converted to each other by finite depth circuit, and they should be thought of as different "boundary phases" separated by a phase transition. These different boundary types have different universal data and a "defect" on the boundary is necessary to separate two boundary phases. (Here the defect is codimension 1 to the gapped boundary.) Axiom **A1** is expected to be violated on such defects. Each reference state we consider in Fig. 12 or Fig. 13 is associated with a particular boundary type.

### 3.2.2 ⋆2d: regions adjacent to a gapped boundary

Figure 14 shows a few basic topology types of regions, for a 2d system with a gapped boundary. We draw them either on $\underline{\mathbf{B}}^2$ or $\underline{\underline{\mathbf{B}}}^2$, whichever is more convenient.

For some of the topology types, one can make immersed versions of the regions. Two examples are shown in Fig. 15. In particular, in the second example, the "clock" region cannot be smoothly deformed into the "mushroom" region within $\underline{\mathbf{B}}^2$. This is a new feature, and because of this, it is not obvious whether the two regions have isomorphic information convex sets. Nevertheless, it is possible to show that the information convex sets of the "clock" and the "mushroom" are isomorphic.

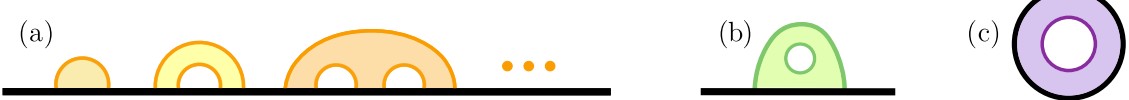

Figure 14: A few basic topologies of regions adjacent to a gapped boundary.

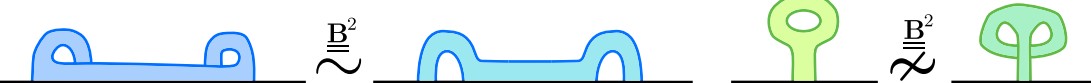

Figure 15: Immersed versions of embedded regions. (a) The immersed region is regular homotopic to the embedded one on $\underline{\underline{\mathbf{B}}}^2$. (b) The immersed region is not regular homotopic to the embedded one on $\underline{\underline{\mathbf{B}}}^2$, although the two regions are homeomorphic.

We also note that the gapped boundary of a 2d system only contributes one type of thickened entanglement boundary. That is the half annulus, i.e., the second region in Fig. 14(a). On $\underline{\underline{\mathbf{B}}}^2$, any immersion of the half annulus is regular homotopic to the embedded one.

### 3.2.3 ⋆3d: regions adjacent to a gapped boundary

In the context of 3d systems with a gapped boundary, we discuss two classes of basic topologies. One class is the boundary analog of genus-$g$ handlebodies; see Fig. 16. There are three subclasses. Each region is sectorizable. Another class can be thought of as the boundary analog of ball minus $k$-ball; see Fig. 17.

Connected components of thickened entanglement boundaries adjacent to the gapped boundary provide another class of basic regions. They are of the form genus-$g$ handlebody shell cut by circles. We note that there can be inequivalent ways to "fill in" the shell and obtain connected sectorizable regions. In fact, the orange regions and the blue regions in Fig. 16 are related in this way. To see this, we observe that the orange regions and blue regions in Fig. 16 complement each other in $\underline{\underline{\mathbf{B}}}^3$.

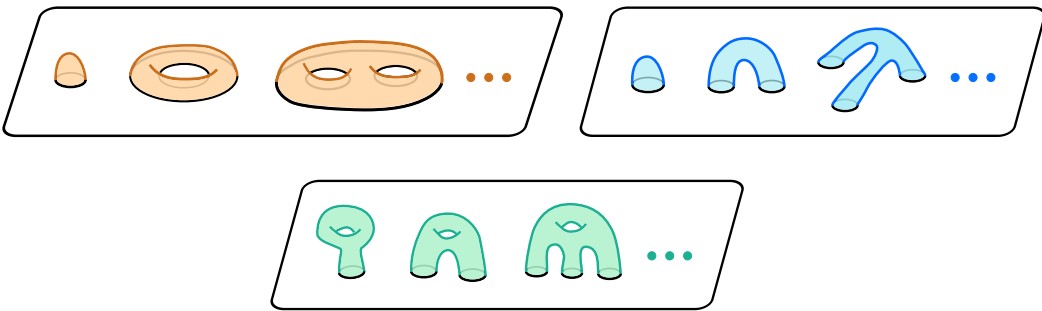

Figure 16: Boundary analogs of genus-$g$ handlebodies. There are three subclasses.

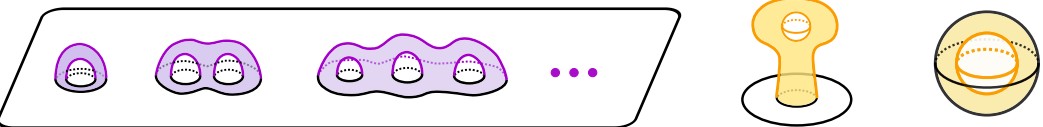

Figure 17: Boundary analogs of ball minus $k$ balls.

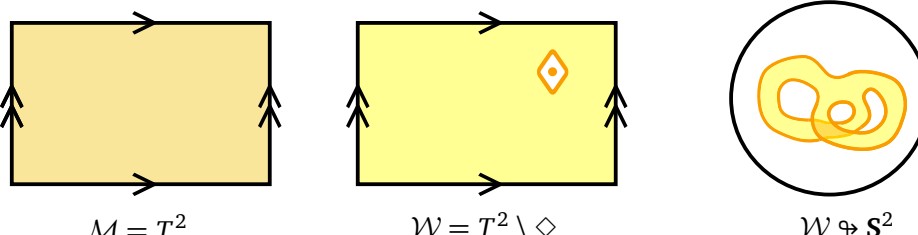

$\mathcal{M} = T^2$ $\qquad\qquad$ $\mathcal{W} = T^2 \setminus \diamond$ $\qquad\qquad$ $\mathcal{W} \pitchfork \mathbf{S}^2$

Figure 18: The *completion* $(T^2, T^2 \setminus B^2 \pitchfork \mathbf{S}^2)$ as an illustration of Definition 4.1. When $\mathcal{M}$ is a torus $T^2$, we remove a small ball $B^2$ (which we refer to as a *completion point*, represented in the figure as a "diamond" $\diamond$) to obtain $\mathcal{W}$. The immersion $\mathcal{W} \pitchfork \mathbf{S}^2$ is explicitly shown in the figure. In many of the later figures, we omit the drawing of the black circle indicating $\mathbf{S}^n$ whenever this is convenient and not confusing.

## 4 Closed manifolds completed from immersed regions

In this section, we shall be interested in finding reference states on closed manifolds. If a gapped many-body system is described by topological quantum field theory (TQFT) [2,3,39], it is natural to put the system on various space manifolds (and spacetime manifolds). If an entanglement bootstrap reference state secretly obeys the TQFT description, as is suggested by all the progress up to now, one should expect that a reference state can be put on various orientable space manifolds. The justification of this for general space dimensions remains an important open problem. Below, we solve the 2d case and make concrete progress on the 3d case.

Our progress rests on the idea of completing immersed regions to compact manifolds. As before, we use $\mathcal{A} \pitchfork \mathcal{B}$ to denote $\mathcal{A}$ immersed in $\mathcal{B}$. When it is necessary to specify the immersion map, we use $\mathcal{A} \stackrel{\varphi}{\pitchfork} \mathcal{B}$. We start with a general definition.

**Definition 4.1** (Completion in $\mathcal{N}$). Let $\mathcal{N}$ be a connected manifold, possibly with entanglement boundaries, equipped with a reference state $\sigma_{\mathcal{N}}$ that satisfies the entanglement bootstrap axioms. We say a closed manifold $\mathcal{M}$ allows a *completion in* $\mathcal{N}$, $(\mathcal{M}, \mathcal{W} \stackrel{\varphi}{\pitchfork} \mathcal{N})$, if

1. $\mathcal{W} \subset \mathcal{M}$ is obtained by removing a finite number of separated balls from $\mathcal{M}$.

2. $\mathcal{W} \stackrel{\varphi}{\pitchfork} \mathcal{N}$ is an immersion for which $\Sigma(\mathcal{W})$ is non-empty.

If $\mathcal{N} = \mathbf{B}^n$ or $\mathbf{S}^n$, we call $(\mathcal{M}, \mathcal{W} \stackrel{\varphi}{\pitchfork} \mathcal{N})$ a *completion* for short.

The reason we are interested in completion is that it provides a state on manifold $\mathcal{M}$ which possesses a few nice properties (Proposition 4.2). The state satisfies axioms **A0** and **A1** except possibly at a number of "completion points". Moreover, on $B^n \subset \mathcal{W}$ the state is locally indistinguishable from the reference state $\sigma$. The reason we are interested in the specific choice $\mathcal{N} = \mathbf{B}^n$ is that it is the "cheapest" choice: a ball is a subsystem of any manifold. While we require $\mathcal{N}$ to be connected, $\mathcal{M}$ does not have to be connected. An equally simple choice is a sphere $\mathbf{S}^n$; it is equally simple because of the sphere completion lemma, which we shall review; see Fig. 21. (We shall also consider versions of completion in the context related to gapped boundaries, where $\mathcal{M}$ is a compact[25] manifold with boundaries and the simplest choices of $\mathcal{N}$ are $\underline{\mathbf{B}}^n$ and $\underline{\underline{\mathbf{B}}}^n$.)

---

[25]In entanglement bootstrap, we always consider regions consisting of a finite number of sites. We shall refer to a manifold without entanglement boundaries as compact, and if entanglement boundaries and gapped boundaries are both absent, we say the manifold is closed.

**Proposition 4.2.** *Let $(\mathcal{M}, \mathcal{W} \overset{\varphi}{\pitchfork} \mathcal{N})$ be a completion in $\mathcal{N}$ of the closed manifold $\mathcal{M}$, where the thickened boundary $\partial\mathcal{W}$ has $k$ connected components. Then, there exists a choice of superselection sector labels $\{a_i\}_{i=1}^{k}$ and a state $|\phi^{\{a_i\}}\rangle$ on $\mathcal{M}$ such that*

$$\mathrm{Tr}_{\mathcal{M}\backslash\mathcal{W}} |\phi^{\{a_i\}}\rangle\langle\phi^{\{a_i\}}| \in \mathrm{ext}(\Sigma_{a_1\cdots a_k}(\mathcal{W})). \tag{15}$$

*Furthermore, **A0** and **A1** hold everywhere on $|\phi^{\{a_i\}}\rangle$ expect for possible violations of **A1** at the completion points: for a ball centered at the ith completion point, $\Delta(B,C,D)_{|\phi^{\{a_i\}}\rangle} = 2\ln d_{a_i}$. Here, $d_{a_i}$ is the quantum dimension of $a_i$. The information convex set $\Sigma_{a_1\cdots a_k}(\mathcal{W})$ is determined by the immersion $\mathcal{W} \overset{\varphi}{\pitchfork} \mathcal{N}$ as well as the reference state $\sigma_{\mathcal{N}}$.*

*Proof.* The proof follows from the completion trick (Lemma 4.4). $\qquad\square$

Intuitively speaking, Proposition 4.2 says that, if a manifold $\mathcal{M}$ allows a completion in $\mathcal{N}$, we can obtain a state on $\mathcal{M}$ that satisfies the entanglement bootstrap axioms, up to possible violations at a finite number of isolated completion points. These violations are well-controlled and are attributed to non-Abelian superselection sectors of point particles[26] at those completion points. Furthermore, the state is locally "vacuum-like" since it is locally indistinguishable from the reference state on $\mathcal{N}$ for all small balls contained in $\mathcal{W}$.

**Proposition 4.3.** *If $(\mathcal{M}, \mathcal{W} \overset{\varphi}{\pitchfork} \mathcal{N})$ is a completion in $\mathcal{N}$ of $\mathcal{M}$, and $\mathcal{N}$ is orientable, then $\mathcal{M}$ is orientable.*

*Proof.* We provide proof by contradiction. Suppose that the manifold $\mathcal{M}$ is nonorientable. Then there exists a small ball in $\mathcal{W}$, which can be transported around and back to the original place and gets its orientation flipped; the whole process happens within $\mathcal{W}$. This happens no matter how small the ball is. However, because $\mathcal{W} \pitchfork \mathcal{N}$, we can always choose the ball small enough so that it remains embedded in $\mathcal{N}$ during the whole process. Such a process cannot flip the orientation of the small ball because $\mathcal{N}$ is orientable. This leads to a contradiction and accomplishes the proof. $\qquad\square$

**Remark.** Here is the more general observation in differential geometry terms. An immersion $\mathcal{W} \pitchfork \mathcal{N}$ implies that all nontrivial characteristic classes of the tangent bundle $T\mathcal{W}$ must agree with those of $T\mathcal{N}$. This includes the first Stieffel-Whitney class $w_1$ whose vanishing is required for orientability. Therefore, $\mathcal{M}$ is orientable if $\mathcal{N}$ is orientable, and $\mathcal{M}$ is spin if $\mathcal{N}$ is spin (see *e.g.* [49] Corollary 6.2.4).

Next, we describe the completion trick (Lemma 4.4). It is a direct generalization of the idea of sphere completion introduced in [6]; see Lemma 3.1 therein. The completion trick allows us to collapse an arbitrary subset of spherical entanglement boundaries, each of which becomes a completion point. See Fig. 19 for an illustration.

**Lemma 4.4** (Completion trick)**.** *Let $\widetilde{\mathcal{W}}$ be an $n$-dimensional manifold with $l$ boundary components. $\mathcal{W} \subset \widetilde{\mathcal{W}}$ is obtained by deleting $k$ internal balls (see Fig. 19 for an illustration). $\mathcal{W}$ is an immersed region, which carries a non-empty information convex set $\Sigma(\mathcal{W})$. Let $\partial\mathcal{W} = (X_1 \cup \cdots \cup X_k) \cup (Y_1 \cup \cdots \cup Y_l)$, where $X_i$ and $Y_j$ each is a connected component. $X_i$s are spherical boundaries (i.e., $X_i = S^{n-1} \times \mathbb{I}$) obtained by deleting balls from $\widetilde{\mathcal{W}}$. For each $\rho_{\mathcal{W}} \in \Sigma_{a_1\cdots a_k b_1\cdots b_l}(\mathcal{W})$, there is a state $\widetilde{\rho}_{\widetilde{\mathcal{W}}_+}$ (where $\widetilde{\mathcal{W}}_+$ is a thickening of $\widetilde{\mathcal{W}}$) such that*

---

[26]For topologically nontrivial choices of $\mathcal{N}$, such non-Abelian superselection sectors may also come from topological defects. By "topological defects" in this context, we meant extrinsic defects, such as the point defect in the toric code on which the duality wall ends. More generally, these defects exhibit interesting phenomena such as permuting anyons, [43]. See Example 4.27 in §4.3 for more discussion.

1. $\mathrm{Tr}_{\widetilde{\mathcal{W}}_+ \backslash \mathcal{W}}\, \widetilde{\rho}_{\widetilde{\mathcal{W}}_+} = \rho_{\mathcal{W}}$.

2. If $l = 0$, and $\rho_{\mathcal{W}} \in \mathrm{ext}(\Sigma_{a_1 \cdots a_k}(\mathcal{W}))$, then $\widetilde{\mathcal{W}}$ has no entanglement boundary, and $\widetilde{\rho}_{\widetilde{\mathcal{W}}}$ is pure.

3. Axioms **A0** and **A1** hold on $\widetilde{\mathcal{W}}_+$ except for possible violation of **A1** on the "completion points". On a ball centered at the ith completion point,

$$\Delta(B, C, D)_{\widetilde{\rho}} = 2 \ln d_{a_i}, \quad a_i \in \mathcal{C}_{X_i}, \quad \text{for the partition in Fig. 20(b).} \tag{16}$$

In particular, **A1** holds on the i-th completion point if $a_i$ is Abelian.

Note that we do not need to assume that the spherical entanglement boundaries are embedded. This trick will be useful in "filling the holes" for various immersed spherical entanglement boundaries. The key idea, as is illustrated in Fig. 19, is (1) for an extreme point, each connected component of the entanglement boundary can be purified separately, and (2) each spherical boundary, together with its associated purifying region, can be collapsed to a completion point.

**Remark.** For most applications, it is sufficient to thicken the subset of (spherical) entanglement boundaries that we wish to purify. Nonetheless, we choose to thicken all entanglement boundaries, both orange and green ones in Fig. 19. (For this reason, the final region we obtain is $\widetilde{\mathcal{W}}_+$, which contains $\widetilde{\mathcal{W}}$.) This is convenient in some applications: the extra layer can sometimes be useful when considering information convex sets.

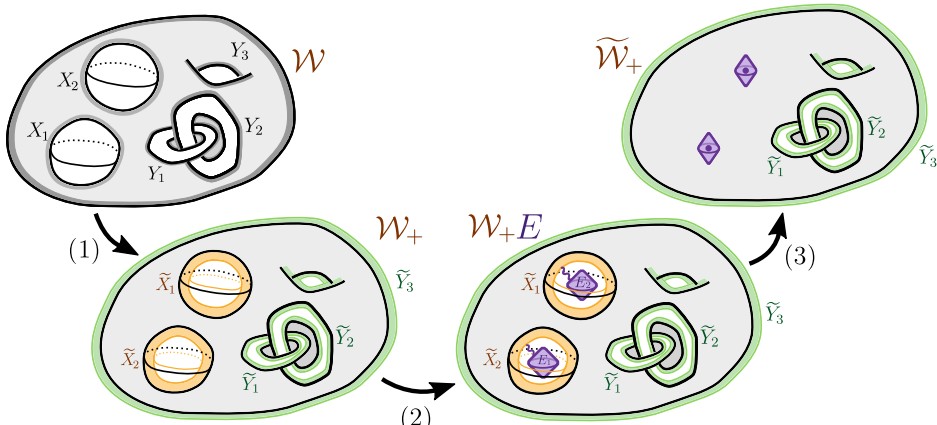

Figure 19: Illustration of the completion trick (Lemma 4.4). For illustration purposes, we consider a region $\mathcal{W}$ (gray) in the 3d bulk, which has 2 spherical entanglement boundaries and 3 torus entanglement boundaries. We wish to collapse both spherical entanglement boundaries, and therefore this corresponds to the case with $k = 2$ and $l = 3$ in Lemma 4.4. The dark gray area in the leftmost figure is $\cup_{i=1}^{k} X_i \cup_{j=1}^{l} Y_j$, the thickened entanglement boundary of $\mathcal{W}$. (1) Expand $\mathcal{W}$ passing its entanglement boundary and obtain its thickening $\mathcal{W}_+$. $\partial \mathcal{W}_+ = \mathcal{W}_+ \backslash \mathcal{W} = \cup_{i=1}^{k} \widetilde{X}_i \cup_{j=1}^{l} \widetilde{Y}_j$. The orange sphere shells are $\cup_{i=1}^{k} \widetilde{X}_i$. The green torus shells are $\cup_{j=1}^{l} \widetilde{Y}_j$. (2) Purify each spherical entanglement boundary of $\mathcal{W}_+$ separately, where each purple box represents a purifying system $E_i$. $E = \cup_{i=1}^{k} E_i$. (3) Collapsing each spherical entanglement boundary (labeled by $i \in \{1, \ldots, k\}$) to a single site. In terms of quantum states, this is done by identifying each purifying region $E_i$ and the thickened spherical entanglement boundary $\widetilde{X}_i$ (which we choose to collapse) as a site. Each site so obtained is a *completion point*. The resulting region is $\widetilde{\mathcal{W}}_+$.

**Remark.** Implicit in our statement of the completion trick is the following fact. Part of the data associated with the reference state is a cell decomposition of the manifold on which the state lives. In the completion trick, we are using the immersion of $\mathcal{W}$ to pull back information from $\mathcal{N}$ to $\mathcal{W}$. As in the original applications of the torus trick [17], the immersion can be used to pull back the structure of the cell decomposition from $\mathcal{N}$ to $\mathcal{W}$. In the completion step, we extend this cell complex by gluing in balls.

*Proof of Lemma 4.4, items 1 and 2.* First, we expand $\mathcal{W}$ passing every entanglement boundary to obtain $\mathcal{W}_+$, so that $\partial\mathcal{W}_+ = \mathcal{W}_+ \setminus \mathcal{W}$. See Fig. 19 for an illustration. We further write $\mathcal{W}_+ \setminus \mathcal{W} = (\widetilde{X}_1 \cup \cdots \cup \widetilde{X}_k) \cup (\widetilde{Y}_1 \cup \cdots \cup \widetilde{Y}_l)$, where the labels of the connected components match that of $\partial\mathcal{W}$. By the Hilbert space theorem (in particular, Proposition D.4 of Ref. [5]), any state $\rho_{\mathcal{W}_+} \in \Sigma_{a_1 \cdots a_k b_1 \cdots b_l}(\mathcal{W}_+)$ allows a simple factorization along the thickened boundaries:

$$\rho_{\mathcal{W}_+} = \left(\bigotimes_{i=1}^{k} \rho^{a_i}_{A_i B_i^L}\right) \otimes \left(\bigotimes_{j=1}^{l} \rho^{b_j}_{A_{k+j} B_{k+j}^L}\right) \otimes \lambda_{(\cup_{i=1}^{k+l} B_i^R) \cup \widehat{C}}, \tag{17}$$

where $\widehat{C}$ is $\mathcal{W}_+ \setminus \cup_i A_i B_i$. Here, for $i \in \{1, \cdots, k\}$, $A_i B_i \subset \widetilde{X}_i X_i$, $A_i \supset \widetilde{X}_i$, and $B_i$ can deform to $X_i$ by extensions; similarly, for $i \in \{k+1, \cdots, k+l\}$, $A_{k+i} B_{k+i} \subset \widetilde{Y}_i Y_i$, $A_{k+i} \supset \widetilde{Y}_i$ and $B_{k+i}$ can deform to $Y_i$ by extensions. $\mathcal{H}_{B_i} = (\mathcal{H}_{B_i^L} \otimes \mathcal{H}_{B_i^R}) \oplus \cdots$. Note that $B_i^L$ and $B_i^R$ do not represent subsystems in general. The density matrices $\{\rho^{a_i}_{A_i B_i^L}\}$ are supported on $\mathcal{H}_{A_i} \otimes \mathcal{H}_{B_i^L}$, $\{\rho^{b_i}_{A_{k+i} B_{k+i}^L}\}$ are supported on $\mathcal{H}_{A_{k+i}} \otimes \mathcal{H}_{B_{k+i}^L}$, and the state $\lambda_{(\cup_{i=1}^{k+l} B_i^R) \cup \widehat{C}}$ is supported on $(\otimes_{i=1}^{k+l} \mathcal{H}_{B_i^R}) \otimes \mathcal{H}_{\widehat{C}}$. $\{a_i\}_{i=1}^{k}$ and $\{b_j\}_{j=1}^{l}$ are labels of superselection sectors, and the states on $\rho^{a_i}_{A_i B_i^L}$ and $\rho^{b_j}_{A_{k+j} B_{k+j}^L}$ only depend on the choice of them; other information of $\rho_{\mathcal{W}_+} \in \Sigma_{a_1 \cdots a_k b_1 \cdots b_l}(\mathcal{W}_+)$ are in $\lambda_{(\cup_{i=1}^{k+l} B_i^R) \cup \widehat{C}}$. To summarize the intuition behind Eq. (17), we say a "fuzzy cut" is contained in $B_i$ for any $i$.

Next, we introduce a purifying system supported on $E = \cup_{i=1}^{k} E_i$. Each $E_i$ carries a Hilbert space $\mathcal{H}_{E_i}$, whose dimension is finite but large enough. For $i \in \{1, \cdots, k\}$ we purify $\rho^{a_i}_{A_i B_i^L}$ with $E_i$ and let the resulting state be $|\phi^{a_i}_{A_i B_i^L E_i}\rangle$. We topologically collapse $\widetilde{X}_i E_i$ to a single site (i.e., a completion point). We call the region obtained by collapsing all the $k$ sphere shells as $\widetilde{\mathcal{W}}_+$, which is equipped with a topology associated with topologically collapsing the entanglement boundaries into points (i.e., "healing the punctures"). The state we obtain can be written as

$$\widetilde{\rho}_{\widetilde{\mathcal{W}}_+} = \left(\bigotimes_{i=1}^{k} |\phi^{a_i}_{A_i B_i^L E_i}\rangle\langle\phi^{a_i}_{A_i B_i^L E_i}|\right) \otimes \left(\bigotimes_{j=1}^{l} \rho^{b_j}_{A_{k+j} B_{k+j}^L}\right) \otimes \lambda_{(\cup_{i=1}^{k+l} B_i^R) \cup \widehat{C}}. \tag{18}$$

From this expression, we can immediately verify statement 1. Because $\lambda_{(\cup_{i=1}^{k+l} B_i^R) \cup \widehat{C}}$ is pure for extreme points (Eq. D.9 of Ref. [5]), statement 2 holds. This completes the verification of statements 1 and 2 of Lemma 4.4. □

*Proof of Lemma 4.4, item 3.* For axiom **A0** on $i$-th completion point in statement 2, consider the partition of a ball centered around the $i$-th completion point as $BC$, shown in Fig. 20(a). We can choose $BC$ so that $X_i \subset C$. Here the state considered is $\widetilde{\rho}$ in Eq. (18). To prove $\Delta(B, C)_{\widetilde{\rho}} = 0$, we can first prove $\Delta(X_i, \widetilde{X}_i E_i)_{\widetilde{\rho}} = 0$. Then $\Delta(B, C)_{\widetilde{\rho}} = 0$ follows from the extension of the axioms [4]. The structure of the state $\widetilde{\rho}$ (and the relation between $A_i B_i C_i$ and $X_i \widetilde{X}_i$) is shown in (19); the horizontal direction in the figure is the distance to the hole labeled $i$. The wiggly yellow line indicates the "fuzzy cut" that separates $B_i^L$ and $B_i^R$.

$$\begin{array}{|c|c|c|c|} \hline E_i & A_i & B_i^L \, \rangle \, B_i^R & C \\ \hline & \widetilde{X}_i & X_i & \\ \hline \end{array} \tag{19}$$

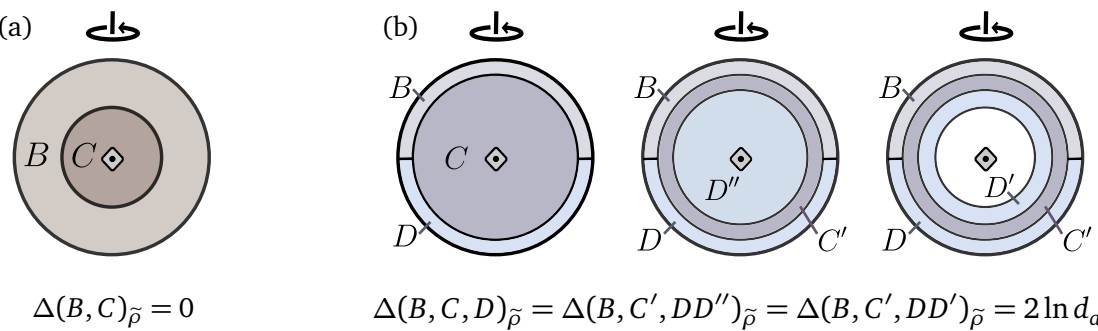

(a) $\Delta(B,C)_{\widetilde{\rho}} = 0$

(b) $\Delta(B,C,D)_{\widetilde{\rho}} = \Delta(B,C',DD'')_{\widetilde{\rho}} = \Delta(B,C',DD')_{\widetilde{\rho}} = 2\ln d_{a_i}$

Figure 20: About axiom **A0** and the potential violation of axiom **A1** centered at a completion point. Note that $\widetilde{X}_i E_i$ is within the completion point.

The reduced density matrices on $X_i \widetilde{X}_i E_i$ for $\widetilde{\rho}_{\widetilde{\mathcal{W}}_+}$ can be directly computed from Eq. (18) as $\widetilde{\rho}_{X_i \widetilde{X}_i E_i} = |\phi^{a_i}_{A_i B_i^L E_i}\rangle\langle\phi^{a_i}_{A_i B_i^L E_i}| \otimes \lambda_R$, where $A_i, B_i$ were defined below Eq. (17), and $\lambda_R$ is $\lambda$ reduced to the Hilbert space $\mathcal{H}_{B_i^R} \otimes \mathcal{H}_{X_i \widetilde{X}_i \backslash A_i B_i}$. Using the tensor product structure of $\widetilde{\rho}_{X_i \widetilde{X}_i E_i}$ and $\widetilde{\rho}_{X_i}$, it is straightforward to see that $\Delta(X_i, \widetilde{X}_i E_i)_{\widetilde{\rho}} = 0$.

Next we show that axiom **A1** is violated by an amount $2\ln d_{a_i}$ on the $i$-th completion point, where $d_{a_i}$ is the quantum dimension of the point excitation $a_i \in \mathcal{C}_{X_i}$. (Note that the sphere shell may not be embedded, and therefore, we use the definition of the quantum dimension in Eq. (9).) The proof idea is shown in Fig. 20(b) and is similar to the use of the Decoupling Lemma in Appendix D of [6]. The first step is to show

$$\Delta(B,C,D)_{\widetilde{\rho}} = \Delta(B,C',DD'')_{\widetilde{\rho}}, \tag{20}$$

for the partitions illustrated in Fig. 20(b). This is true because $\Delta(C',D'')_{\widetilde{\rho}} = 0$ and $\Delta(BC',D'')_{\widetilde{\rho}} = 0$, which are extended axiom **A0** at the $i$-th completion point, as verified in the earlier part of the proof. The second step is to prove

$$\Delta(B,C',DD'')_{\widetilde{\rho}} = \Delta(B,C',DD')_{\widetilde{\rho}}. \tag{21}$$

This follows from $\Delta(D',D'' \backslash D')_{\widetilde{\rho}} = 0$ and $\Delta(C'DD',D'' \backslash D')_{\widetilde{\rho}} = 0$, where we applied axiom **A0** twice again.

From the second definition of the quantum dimension of a point particle Eq. (9), we see that the correction is $\Delta(B,C',DD')_{\widetilde{\rho}} = 2\ln d_{a_i}$, and it is nonzero only when the point particle is non-Abelian. This completes the proof. $\qquad\square$

Immediate corollaries of the completion trick (Lemma 4.4) are: (1) It is possible to complete to a sphere $\mathbf{S}^n$ with a reference state on a ball $\mathbf{B}^n$. This is discussed in Ref. [6]. (2) It is possible to obtain a reference state on $\underline{\mathbf{B}}^n$ from a reference state on $\underline{\underline{\mathbf{B}}}^n$. In both cases, the reference state on the compact manifolds ($\overline{\mathbf{S}}^n$ or $\underline{\underline{\mathbf{B}}}^n$) satisfies the axioms everywhere, including the ball containing the completion point. (This also means that the Hilbert space dimensions on compact manifolds $\mathbf{S}^n$ and $\underline{\underline{\mathbf{B}}}^n$ are one dimensional, i.e., $\dim\mathbb{V}(\mathbf{S}^n) = \dim\mathbb{V}(\underline{\underline{\mathbf{B}}}^n) = 1$.) We illustrate the two cases in Fig. 21.

For a given closed manifold $\mathcal{M}$, is it possible to determine if allows a completion $(\mathcal{M}, \mathcal{W} \overset{\varphi}{\hookrightarrow} \mathbf{S}^n)$? There are a couple of challenges to answering this question generally. The first challenge is to understand what closed manifold $\mathcal{M}$ can immerse in a sphere upon removing a ball. In space dimensions $d \leq 4$, some interesting topology results have been obtained recently [32]. The second challenge is that even if $\mathcal{M}$ can immerse in a sphere, we still need to know if the information convex set $\Sigma(\mathcal{W})$ is non-empty. In the presence of topological order, where there are nontrivial superselection sectors, we need to understand if the density

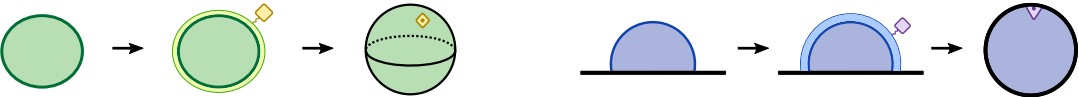

Figure 21: Sphere completion and its analog for systems with a gapped boundary. (Explicitly shown here are the 2d cases.) They are immediate corollaries of the completion trick. Completion points in the bulk are illustrated as a 'diamond' $\diamond$ and completion points adjacent to a gapped boundary are illustrated as a 'triangle' $\triangle$. This convention will be adopted throughout the paper.

matrices on pieces of the immersed regions can be consistently merged. In addition, we wish to know if there is always a special *abelian* superselection sector allowed on the punctures, so that they may be healed. The general answer to this question will be explored elsewhere. Nonetheless, the vacuum block completion, developed in §4.1, gives a constructive answer to many interesting cases, including all orientable manifolds in two dimensions.

Let us point out that for invertible phases (i.e., systems with only trivial superselection sectors), the second challenge mentioned in the previous paragraph disappears, and the task is simpler. Intriguingly, for invertible phases, Kitaev has developed a different approach for putting the ground state on a class of closed manifold; see [50] at around 1 hour and 35 minutes, where the class of manifolds is referred to as normally framed manifolds. We note that this is the same condition[27] that allows a punctured manifold to be immersed in a sphere, at least up to dimension four [32]. It is unclear to us, however, if our approach is related to Kitaev's approach.

## 4.1 Vacuum block completion

In this section, we consider a specific type of completion, which we shall refer to as *vacuum block completion*. It comes with two important features. First, the immersed region $\mathcal{W} \subset \mathcal{M}$ decomposes as $\mathcal{W} = \cup_i \mathcal{V}_i^{\diamond}$, where $\{\mathcal{V}_i^{\diamond}\}$ is a set of *building blocks*, and the "gluing" between any two building blocks is on whole entanglement boundaries. Second, on each building block $\mathcal{V}_i^{\diamond}$, a "vacuum" can be specified, even though $\mathcal{V}_i^{\diamond}$ may not be embedded in the space on which the reference state lives.

In the rest of the paper, we will mainly be interested in the immersion of $\mathcal{W}$ in $\mathbf{S^n}$ in the study of bulk phases and the immersion $\mathcal{W}$ in $\underline{\mathbf{B}}^n$ in cases related to the gapped boundaries. The reason, as explained in Fig. 21, is that it is always possible to make a reference state on these compact manifolds starting from a reference state on a ball ($\mathbf{B}^n$ or $\underline{\mathbf{B}}^n$). Nevertheless, we define building blocks in such a way that they have the freedom to be immersed in balls.

### 4.1.1 Building blocks for vacuum block completion

We first give a precise definition of building blocks.

**Definition 4.5** (Building blocks, bulk)**.** We say a connected region $\mathcal{V}^{\diamond} \pitchfork \mathbf{S}^n$ (or $\mathbf{B}^n$) is a building block if it satisfies:

1. $\partial \mathcal{V}^{\diamond} = (\cup_{i=1}^k E_i) \cup F$, where $E_i, F$ are connected components of $\partial \mathcal{V}^{\diamond}$. $E_i$s are embedded, $F$ is either empty or an immersed sphere shell ($S^{n-1} \times \mathbb{I}$); in the former case, we may denote $\mathcal{V}^{\diamond}$ as $\mathcal{V}$.

---

[27]Normally framed implies that the tangent bundle can be made trivial by adding trivial bundles. Thus the characteristic classes must vanish.

2. $\Sigma_{[1_E]}(\mathcal{V}^\diamond) = \{\hat{\sigma}_{\mathcal{V}^\diamond}\}$, has a unique element, where $E = \cup_{i=1}^k E_i$. Furthermore, $\hat{\sigma}_{\mathcal{V}^\diamond}$ reduces to an Abelian sector on $F$. (We denote this Abelian sector[28] as $\hat{1} \in \mathcal{C}_F$.)

**Remark.** Note that, in writing $\Sigma_{[1_E]}(\mathcal{V}^\diamond)$, we are using the notation of constrained information convex set (Definition 2.1). In other words, we want the convex subset of $\Sigma(\mathcal{V}^\diamond)$, whose elements reduce to the vacuum on the embedded region $E = \cup_i E_i$. We shall often drop the distinction between the abstract region $\mathcal{V}^\diamond$ with its immersion in the physical system, which is equipped with a Hilbert space and needs the immersion map to specify.

The generalization of building blocks to gapped boundaries is straightforward. The difference is that we immerse the region in $\underline{\mathbf{B}}^n$ or $\underline{\underline{\mathbf{B}}}^n$ (instead of $\mathbf{S}^n$ or $\mathbf{B}^n$) and allow a broader sense of immersed "spherical" entanglement boundary. These entanglement boundaries may either lie in the bulk or adjacent to the gapped boundary.

**Definition 4.6** (*Building blocks, gapped boundary). We say a connected region $\mathcal{V}^{\diamond/\triangle} \looparrowright \underline{\mathbf{B}}^n$ (or $\underline{\underline{\mathbf{B}}}^n$) is a building block if it satisfies:

1. $\partial\mathcal{V}^{\diamond/\triangle} = (\cup_{i=1}^k E_i) \cup F$, where $E_i, F$ are connected components of $\partial\mathcal{V}^{\diamond/\triangle}$. $E_i$s are embedded. $F$ may be empty, an immersed bulk sphere shell in the bulk or an immersed half-sphere shell[29] attaches to the boundary. In these three cases, we may alternatively denote $\mathcal{V}^{\diamond/\triangle}$ as $\mathcal{V}$, $\mathcal{V}^\diamond$, or $\mathcal{V}^\triangle$.

2. $\Sigma_{[1_E]}(\mathcal{V}^{\diamond/\triangle}) = \{\hat{\sigma}_{\mathcal{V}^{\diamond/\triangle}}\}$, has a unique element, where $E = \cup_{i=1}^k E_i$. Furthermore, $\hat{\sigma}_{\mathcal{V}^{\diamond/\triangle}}$ reduces to an Abelian sector ($\hat{1} \in \mathcal{C}_F$) on $F$.

At the moment, it should not be clear how hard it is to find such building blocks. While the first requirement can be verified by looking at the topology, the requirement on the constrained information convex set $\Sigma_{[1_E]}(\mathcal{V}^{\diamond/\triangle}) = \{\hat{\sigma}_{\mathcal{V}^{\diamond/\triangle}}\}$ may not be easy to check. Nonetheless, it turns out that a large class of immersed regions can be shown to be building blocks. Here are some examples.

**Example 4.7** (Building blocks, bulk). A list of examples of building blocks $\mathcal{V}^\diamond$ satisfying Definition 4.5:

1. Any dimension: any connected embedded region.

2. 2d bulk: $S^2$ with $(k+1)$ balls removed. Here, $k$ of the spherical boundaries are embedded, and the remaining one is not. See Eq. (22) for some examples:

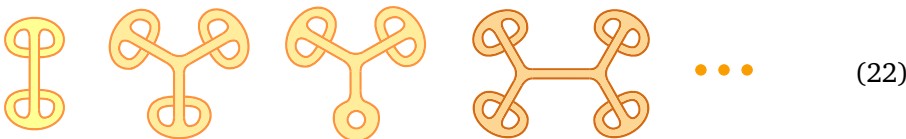

$$\tag{22}$$

To see why these regions are building blocks, we let the thickening of the immersed spherical boundary be $F$. Then the first condition in Definition 4.5 is verified. Some thought goes into the verification of the second condition. We provide two methods.

---

[28]Note that $\hat{1} \in \mathcal{C}_F$ is a special Abelian sector and it is an analog of the vacuum. We use $\hat{1}$ instead of $1$ because if $F$ is not embedded, there will be no reference state (on $F$) to compare with. We do not know if $\hat{1}$ depends on $\mathcal{V}^\diamond$ or not for a given $F$. Below (in Example 6.5) we will give an example where different vacuum block completions of the same manifold produce the same abelian state on $F$.

[29]In $n$ space dimensions, half-sphere shell $B^{n-1} \times \mathbb{I}$ attaches to the gapped boundary at $S^{n-2} \times \mathbb{I}$.

- We convert the immersed region to an *embedded $k$-hole disk* by a smooth deformation, the same as the strategy illustrated in Fig. 6. Here, we want a deformation that keeps the $k$ embedded boundaries embedded (in $\mathbf{S}^2$) in the whole process. Therefore, the sectors must still be the vacuum when they reach the final configuration. The remaining boundary must also carry the vacuum, according to the structure theorem of the information convex set of embedded $k$-hole disks and the fusion rules for anyons.

- Alternatively, we use the completion trick (Lemma 4.4). The strategy is outlined in Eq. (23) below.

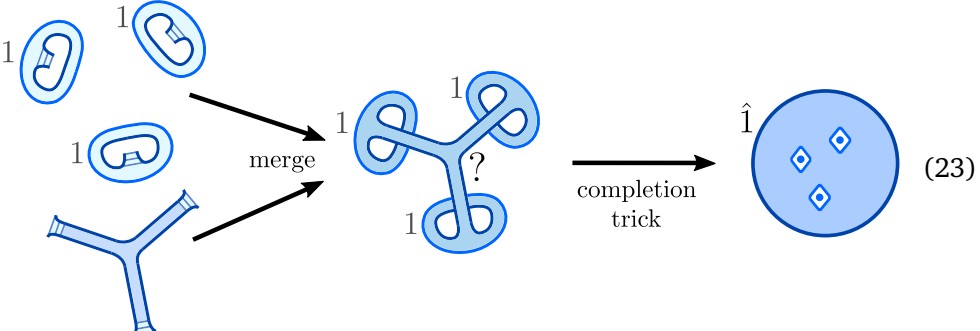

$$\text{(23)}$$

First, $\Sigma_{[1_E]}(\mathcal{V}^{\diamond})$ is non-empty. This is because we can always obtain the immersed region in question by merging a state on an embedded disk to the vacuum of the $k$ disjoint embedded annuli ($E_i$). This gives a construction of an element of $\Sigma_{[1_E]}(\mathcal{V}^{\diamond})$. With the completion trick, we collapse the $k$ embedded spherical boundaries and get a disk $B^2$ with $k$ completion points. (Shown in Eq. (23) is the case $k = 3$.) Because the sector is the vacuum at all these completion points, the axioms are satisfied on them. Finally, the information convex set on a disk has a unique element, where the boundary carries an Abelian sector. We identify this Abelian sector as $\hat{1}$, and this completes the argument.

3. 3d bulk: $S^3$ with $k+1$ balls removed. Here, $k$ of the spherical boundaries are embedded, and the remaining one is not. This is true for the same reason as case 2.

4. 3d bulk: some choices of regions with torus entanglement boundaries, e.g.:

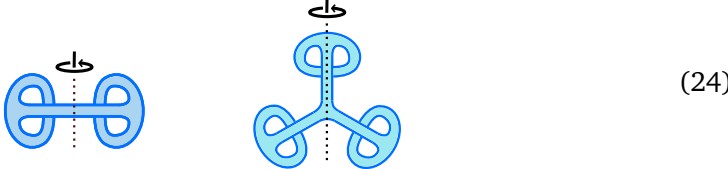

$$\text{(24)}$$

The nontrivial feature of these regions is the existence of a torus entanglement boundary, which cannot be collapsed to a point. We also do not know if it is possible to deform the regions smoothly into an embedded region and keep the torus boundary embedded in the whole process. Alternative justification is needed. One justification comes from the dimensional reduction method developed in Appendix E of [6]. This method maps the problem to a problem in a 2d system with a gapped boundary. Then, according to Example 4.8 below, these 3d regions must be valid building blocks.

**Example 4.8** (*Building blocks, gapped boundary). Here are a few examples of building blocks adjacent to the gapped boundary. In this example, all embeddings are in $\underline{\mathbf{B}}^n$, and in the figures, we only draw part of the gapped boundary of $\underline{\mathbf{B}}^n$.

1. Any dimension: connected embedded regions adjacent to the gapped boundary.

2. 2d with a gapped boundary: a few non-embedded examples as shown:

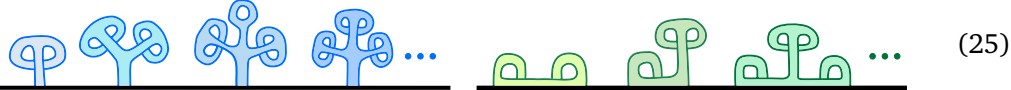 (25)

These are building blocks, as can be checked with the "collapsing" argument explained around Eq. (23). The crucial observation is that, after we collapse all the embedded entanglement boundaries, the region so obtained (with the completion points included) is a half-disk adjacent to the gapped boundary. Thus, the remaining entanglement boundary must be in an Abelian sector.

3. 3d with a gapped boundary: a few non-embedded examples as shown:

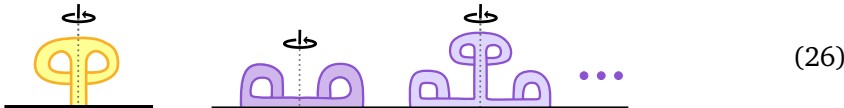 (26)

It is evident that the topology of the regions satisfies the definition of building blocks. We also need to verify the property related to the quantum state. The first example can be shown to be a building block by the "collapsing" argument explained around Eq. (23). However, the same trick does not apply to the remaining two examples due to the existence of an embedded entanglement boundary that is not spherical. Nevertheless, these are building blocks; we shall establish this fact only after we understand pairing manifolds. Explicitly, the first purple region is precisely the $\bar{X}^{\triangle}$ of Example 6.7 in §6.2. The second purple region is related to the first one by the collapsing of the spherical entanglement boundary in the bulk.

### 4.1.2 Vacuum block completion: definition and properties

We give the precise definition of vacuum block completion and then discuss its properties. Examples will be presented in the next few sections.

**Definition 4.9** (Vacuum block completion, bulk)**.** We say a manifold $\mathcal{M}$ allows a vacuum block completion $(\mathcal{M}, \{\mathcal{V}_i\}, \mathcal{W} \overset{\varphi}{\looparrowright} \mathbf{S}^n)$ if $\mathcal{M} = \cup_i \mathcal{V}_i$, glued along entire boundaries, $\mathcal{W} \subset \mathcal{M}$, and

1. $\mathcal{V}_i \cap (\mathcal{M} \setminus \mathcal{W})$ is either an internal ball of $\mathcal{V}_i$ or empty.

2. $\mathcal{V}_i^{\diamondsuit} \equiv \mathcal{V}_i \cap \mathcal{W}$, equipped with the immersion $\varphi$ restricted onto it, is a building block (by Definition 4.5), where $\partial \mathcal{V}_i^{\diamondsuit} = E \cup F$ with $E = \partial \mathcal{V}_i$ embedded in $\mathbf{S}^n$ and $F = \mathcal{V}_i \cap \partial \mathcal{W}$.

**Remark.** We will also use a boundary version of vacuum block completion. We omit the precise definition since it is obtained by replacing $\mathbf{S}^n$ with $\underline{\underline{\mathbf{B}}}^n$ and adopting the boundary version of building blocks (Definition 4.6).

**Proposition 4.10.** *Any vacuum block completion $(\mathcal{M}, \{\mathcal{V}_i\}, \mathcal{W} \overset{\varphi}{\looparrowright} \mathbf{S}^n)$ gives a completion of $\mathcal{M}$. Furthermore, there is a special element $\hat{\sigma}_{\mathcal{W}}$ of $\Sigma(\mathcal{W})$. It is the unique state in $\Sigma(\mathcal{W})$ that reduces to $\hat{\sigma}_{\mathcal{V}_i^{\diamondsuit}}$ for any building block $\mathcal{V}_i^{\diamondsuit}$. We call $\hat{\sigma}_{\mathcal{W}}$ the canonical state on $\mathcal{W}$.*

*Proof.* First, since $\mathcal{V}_i^{\diamondsuit}$ is a building block, it has a state $\hat{\sigma}_{\mathcal{V}_i^{\diamondsuit}}$ in its information convex set. We can merge these states, after expanding the regions $(\mathcal{V}_i^{\diamondsuit})$ slightly on their shared embedded hypersurfaces (i.e., embedded entanglement boundaries). This merging is possible because

these states all carry the vacuum sector on these shared embedded hypersurfaces, and therefore, they also have the quantum Markov state structure required in the merging theorem. The merged state on $\mathcal{W}$ is an extreme point $\hat{\sigma}_{\mathcal{W}} \in \Sigma_{\hat{1}}(\mathcal{W})$. This follows from Lemma 2.20 of [6] (used to prove the Associativity Theorem). Here $\hat{1}$ refers to the fact that the state carries an Abelian sector $\hat{1}$ on every (spherical) entanglement boundary of $\mathcal{W}$. □

**Definition 4.11** (Canonical state on $\mathcal{M}$)**.** For a given vacuum block completion $(\mathcal{M}, \{\mathcal{V}_i\}, \mathcal{W} \overset{\varphi}{\hookrightarrow} \mathbf{S}^n)$, call the completion of the canonical state $\hat{\sigma}_{\mathcal{W}}$ of $\mathcal{W}$ (defined in Proposition 4.10) on $\mathcal{M}$ (by the completion trick, Lemma 4.4) as the canonical state on $\mathcal{M}$. We denote this pure canonical state as $|1_{\{\mathcal{V}_i\}}\rangle$.

The immediate consequences of this definition are (1) $|1_{\{\mathcal{V}_i\}}\rangle$ reduces to $\hat{\sigma}_{\mathcal{V}_i^\diamond}$ for any building block $\mathcal{V}_i^\diamond$, and (2) axioms **A0** and **A1** holds everywhere on $\mathcal{M}$ for $|1_{\{\mathcal{V}_i\}}\rangle$.

Note that the state $|1_{\{\mathcal{V}_i\}}\rangle$ is not unique. Nevertheless, it is unique up to tensoring in extra product degrees of freedom and applying local unitaries $\{U_i\}$ on the completion points. Taking these operations as equivalence relations, we say $|1_{\{\mathcal{V}_i\}}\rangle$ is canonical.

## 4.2 Examples of vacuum block completion

### 4.2.1 Vacuum block completion in the 2d bulk

**Proposition 4.12.** *Every 2d orientable manifold allows a vacuum block completion.*

We construct each case as an example; they comprise a constructive proof of the Proposition. We leave the generalization of this statement to nonorientable manifolds to the interested reader. The trick is to consider the completion in a non-orientable $\mathcal{N}$.

**Example 4.13** (Sphere $S^2$)**.** We already have a reference state on $S^2$, and therefore not much needs to be done. For pedagogical reasons, we discuss two methods. The first method is to pick one building block $\mathcal{V} = \mathbf{S}^2$. There is no completion point, and $\mathcal{W} = \mathcal{M} = \mathbf{S}^2$. The second method is to pick two building blocks $\{\mathcal{V}_1, \mathcal{V}_2\}$, both of which are balls obtained by cutting the sphere in halves. In this construction, again, we do not need any completion points. As the reader should expect, the state completed on $\mathcal{M}$ is precisely the reference state on $\mathbf{S}^2$, for both constructions.

**Example 4.14** (Torus $T^2$)**.** For the torus $\mathcal{M} = T^2$, we need two building blocks $\{\mathcal{V}_1, \mathcal{V}_2^\diamond\}$ and one completion point. $\mathcal{V}_1$ is an embedded annulus. $\mathcal{V}_2^\diamond$ is a 2-hole disk, immersed in such a way that two of the three entanglement boundaries are embedded. Here is an illustration:

$$\begin{array}{ccccccc} & & & & & & \\ \underset{\mathcal{V}_1}{\bigcirc} & + & \underset{\mathcal{V}_2^\diamond}{\reflectbox{\text{(shape)}}} & \Rightarrow & \underset{\mathcal{W}}{\text{(shape)}} & \Rightarrow & \underset{\mathcal{M} = T^2}{\text{(torus)}} \end{array} \tag{27}$$

The essence of vacuum block completion is that we can choose the vacuum state on $\mathcal{V}_1$ and merge it with a unique state on $\mathcal{V}_2^\diamond$, so that the state has a special Abelian sector ($\hat{1}$) on $\partial\mathcal{W}$. Then we heal the puncture to obtain a state on $T^2$ that satisfies the axioms everywhere.

**Example 4.15** (genus-$g$ surface $\#g\, T^2$)**.** The genus-$g$ surface is the connected sum of $g$ tori, and therefore we denote it as $\#g\, T^2$. The vacuum block completion for

$$\mathcal{M} = \#g\, T^2 = \underbrace{T^2 \# \cdots \# T^2}_{g \text{ times}}, \tag{28}$$

can be done in multiple ways. Consider the way shown in Fig. 22. This construction needs two building blocks $\{\mathcal{V}_1, \mathcal{V}_2^\diamond\}$, and there is one completion point. $\mathcal{V}_1$ is an embedded $g$-hole disk, and $\mathcal{V}_2^\diamond$ is an $(g+1)$-hole disk with $g$ of its entanglement boundaries embedded.

### 4.2.2   Vacuum block completion in the 3d bulk

In this section, we provide examples of vacuum block completions in the 3d bulk.

**Example 4.16** (3-sphere $S^3$)**.** This can be done in a way analogous to the completion of $S^2$ in Example 4.13. Essentially, nothing needs to be done, given that we have a reference state on the 3-sphere already.

**Example 4.17** ($S^2 \times S^1$)**.** The vacuum block completion of $\mathcal{M} = S^2 \times S^1$ can be constructed as follows.

$$\tag{29}$$

This construction needs two building blocks $\{\mathcal{V}_1, \mathcal{V}_2^\diamond\}$, and there is one completion point. Here $\mathcal{V}_1 = S^2 \times \mathbb{I}$ is an embedded sphere shell, and $\mathcal{V}_2^\diamond$ is a ball minus two balls. $\mathcal{W} = \mathcal{M} \setminus B^3$. (In Eq. (29), $\mathcal{M}$ is illustrated in such a way that a pair of $S^2$ are identified as indicated.) Note that this is a close analog of vacuum block completion of the torus in 2d, considered in Example 4.14.

**Example 4.18** ($\#g(S^2 \times S^1)$)**.** Here $g \geq 2$ is an integer. It is possible to choose two building blocks $\{\mathcal{V}_1, \mathcal{V}_2^\diamond\}$, and there is one completion point, as the illustration below shows.

$$\tag{30}$$

Here, $\mathcal{V}_1$ is embedded and it is homeomorphic to a ball with $g$ balls removed. $\mathcal{V}_2^\diamond$ is a ball with $g+1$ balls removed, immersed in such a way that $g$ of its spherical entanglement boundaries are embedded.

**Remark.** In many of the examples above, the immersed building block can be deformed to an embedded region keeping all (but one) entanglement boundary embedded in the whole deformation process. If we keep track of the deformation, we can further use the isomorphism theorem to compute the Hilbert space dimensions $\dim \mathbb{V}_a(\mathcal{W})$ and $\dim \mathbb{V}(\mathcal{M})$. We omit the details. As we shall explain later, the pairing manifold provides another way to compute the Hilbert space dimensions. Examples will be given in §6.

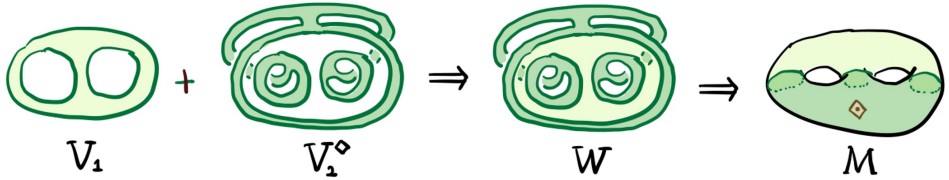

Figure 22: The vacuum block completion for a genus-$g$ surface $\#g\, T^2$. The case $g=2$ is shown explicitly. $\mathcal{V}_1$ is a $g$-hole disk. $\mathcal{V}_2^\diamond$ is an immersed $(g+1)$-hole disk.

### 4.2.3 *Vacuum block completion in 2d with gapped boundaries

Entanglement bootstrap works for gapped boundaries, the setup and axioms are reviewed in §3.2. Below, we provide examples of vacuum block completion in this physical context.

**Example 4.19** (Disk with an entire gapped boundary $\underline{\mathbf{B}}^2$). Let $\mathcal{M} = \underline{\mathbf{B}}^2$. For vacuum block completion, nothing needs to be done here, given that we already have a reference state on $\underline{\mathbf{B}}^2$; see Example 4.13 for an analog. (Recall that the reference state on $\underline{\mathbf{B}}^2$ can come from the boundary analog of the sphere completion; see Fig. 21.)

**Example 4.20** (A finite cylinder with a pair of gapped boundaries). Let $\mathcal{M}$ be the finite cylinder $S^1 \times \mathbb{I}$, with a pair of gapped boundaries. Here $\mathbb{I} = [0,1]$ is a finite interval. (Note that the cylinder with gapped boundaries is not a sectorizable region because we cannot smoothly deform the region back to $\mathcal{M}$ if we trace out the region along the boundary. This is in contrast with the annulus in the bulk.) The vacuum block completion of $\mathcal{M}$ needs two building blocks $\{\mathcal{V}_1, \mathcal{V}_2^{\triangle}\}$, and a completion point, illustrated as follows.

$$
\begin{array}{c}
V_1 \\
V_2^{\triangle}
\end{array} \implies W \implies \mathcal{M}
\tag{31}
$$

Here $\mathcal{V}_1$ is a half-annulus embedded in the disk $\underline{\mathbf{B}}^2$. $\mathcal{V}_2^{\triangle}$ has two embedded entanglement boundaries and its third entanglement boundary is not embedded. From the construction, we see that the two gapped boundaries are of the same type because they are copies of the same boundary of the disk $\underline{\mathbf{B}}^2$.

Another vacuum block completion of the cylinder still uses two building blocks $\{\widetilde{\mathcal{V}}_1, \widetilde{\mathcal{V}}_2^{\triangle}\}$ and has one completion point. This is illustrated as follows.

$$
\begin{array}{c}
\widetilde{V}_1 \\
\widetilde{V}_2^{\triangle}
\end{array} \implies W \implies \mathcal{M}
\tag{32}
$$

In this construction, which is different from the one above, $\widetilde{\mathcal{V}}_1$ is the annulus that covers the entire gapped boundary, and it is embedded in $\underline{\mathbf{B}}^2$. $\widetilde{\mathcal{V}}_2^{\triangle}$ is a "mushroom", which has an embedded entanglement boundary in the bulk and an immersed entanglement boundary that touches the gapped boundary.

**Example 4.21** (A sphere with $k$ disks removed, with $k$ gapped boundaries). Let $\mathcal{M}$ be a sphere with $k$ disks removed, where all the boundaries are gapped boundaries. The vacuum block completion of $\mathcal{M}$ can be done with $k+1$ building blocks, $\{\mathcal{V}_j\}_{j=1}^k \cup \{\mathcal{V}_{k+1}^{\diamond}\}$ and a completion point $\diamond$ within the bulk, as follows:

$$
\{V_j\}_{j=1}^k \quad {}^{\times k} \quad + \quad V_{k+1}^{\diamond} \underset{(k=3)}{\bigcirc} \implies \underset{(k=3)}{\mathcal{M}}
\tag{33}
$$

Here $\{\mathcal{V}_j\}_{j=1}^k$ are "identical copies" of the annulus that covers the entire boundary, embedded in $\underline{\underline{\mathbf{B}}}^2$. $\mathcal{V}_{k+1}^\diamond$ is an immersed region within the bulk, homeomorphic to a $(k+1)$-hole disk, and it is immersed in such a way that $k$ of its $k+1$ entanglement boundaries are embedded.

Since the construction in Eq. (33) is sufficiently nontrivial, we recall the basic intuition hidden in the abstract formulation of vacuum block completion for this example. At an intuitive level, how do we obtain $\mathcal{M}$ by gluing the building blocks? First, we need to deform the $k$ copies of the boundary annulus $\{\mathcal{V}_j\}_{j=1}^k$ such that their entanglement boundaries match the positions of the embedded entanglement boundaries of $\mathcal{V}_{k+1}^\diamond$. Then we glue these entanglement boundaries. The remaining entanglement boundary is in the bulk, and it is then filled using the completion trick, resulting in the completion point in the bulk.

**Remark.** It is also possible to construct vacuum block completions for nonplanar regions with gapped boundaries. These are 2d regions that cannot be embedded in a sphere, which have only gapped boundaries but no entanglement boundaries. One example is illustrated below, and we leave the explicit construction to interested readers.

$$\tag{34}$$

### 4.2.4 *Vacuum block completion in 3d with gapped boundaries

The setup of the following examples is the 3d entanglement bootstrap with gapped boundaries, as stated in §3.2.

**Example 4.22** ($\underline{\underline{\mathbf{B}}}^3$, the ball with an entire gapped boundary)**.** As a region equipped with a reference state, we can simply take one building block $\mathcal{V} = \underline{\underline{\mathbf{B}}}^3$. No completion point is needed here.

**Example 4.23** (Solid torus with an entire gapped boundary)**.** Let $\mathcal{M}$ be a solid torus with an entire gapped boundary. It is possible to have a vacuum block completion of $\mathcal{M}$ with two building blocks $\{\mathcal{V}_1, \mathcal{V}_2^\triangle\}$, and it has a completion point lying adjacent to the boundary, as illustrated below.

$$\tag{35}$$

Here, $\mathcal{V}_1$ is a half-sphere shell adjacent to the gapped boundary. $\mathcal{V}_2^\triangle$ is a half-sphere shell with a $\underline{\underline{\mathbf{B}}}^3$ removed, and it is immersed in $\underline{\underline{\mathbf{B}}}^3$ in such a way that two of its three entanglement boundaries are embedded. In fact, the topologies of these regions are closely related to those that appeared in Example 4.17, the vacuum block completion of $S^2 \times S^1$ in the 3d bulk. To see the relation, we notice that every region (and the completion point) that appeared in this example is some region in Eq. (29) cut in half by the "plane of reflection symmetry".

**Remark.** Following the same idea, we can have a vacuum block completion for the genus-$g$ handlebody, whose gapped boundary is a genus-$g$ surface. We left the details as an exercise for the interested reader. An alternative (and more efficient) way of constructing these manifolds follows from the "sequential completion" technique discussed in §4.3. The idea is to use regions constructed from simpler manifolds to assemble more interesting topologies.

**Example 4.24** (Sphere shell $S^2 \times \mathbb{I}$ with two gapped boundaries)**.** The vacuum block completion of $S^2 \times \mathbb{I}$, a sphere shell in 3d with two spherical gapped boundaries, is shown below.

$$\mathcal{V}_1, \mathcal{V}_2 \quad + \quad \mathcal{V}_3^{\diamond} \quad \Rightarrow \quad \mathcal{M} \tag{36}$$

In this construction, we used three building blocks $\{\mathcal{V}_1, \mathcal{V}_2, \mathcal{V}_3^{\diamond}\}$. $\mathcal{V}_1$ and $\mathcal{V}_2$ are sphere shells that cover an entire gapped boundary. They are embedded in $\underline{\mathbf{B}}^3$, and they are identical copies up to deformations. Therefore, the two gapped boundaries must match in their type. The third building blocks $\mathcal{V}_3^{\diamond}$ is a ball with two balls removed, within the bulk, and it is immersed in such a way that two of its three entanglement boundaries are embedded.

How do we obtain $\mathcal{M}$ topologically? The idea is that we "glue" the entanglement boundary of $\mathcal{V}_1$ ($\mathcal{V}_2$) to an embedded entanglement boundary of $\mathcal{V}_3^{\diamond}$. An important detail is that we need to deform the regions slightly to get the position right! The completion point in the bulk is obtained by collapsing the immersed entanglement boundary of $\mathcal{V}_3^{\diamond}$.

**Remark.** Following the same idea that we used in Example 4.21, we can obtain a vacuum block completion for the connected sum of $k$ solid balls. It has $k$ disjoint spherical gapped boundaries of the same type. (The $k = 2$ case recovers Example 4.24.) We simply need $k$ copies $\mathcal{V}_1$ shown in Eq. (36) and generate the building block in the bulk ($\mathcal{V}_3^{\diamond}$ of Eq. (36)) in an obvious way.

## 4.3 Other examples of completion in $\mathcal{N}$

Here we provide a few examples of completions in $\mathcal{N}$, which are either not vacuum block completion or not obviously so.

**Example 4.25** (Exotic immersion of punctured torus)**.** Let $\mathcal{M} = T^2$ be a torus. Let $\mathcal{W} = T^2 \backslash B^2$ be a torus with a disk removed, i.e., a punctured torus. As we have discussed, $\mathcal{W}$ can be immersed in $\mathbf{S}^2$; see Fig. 18. Moreover, $\mathcal{W}$ cannot be embedded in $\mathbf{S}^2$ since every embedded region is homeomorphic to a $k$-hole disk. Here we consider another immersion $\mathcal{W} \looparrowright \mathbf{S}^2$, as depicted in Fig. 23.

It is an inequivalent immersion compared with the one shown in Fig. 18. This immersion has a non-empty information convex set $\Sigma(\mathcal{W})$, and therefore $(T^2, \mathcal{W} \overset{\varphi}{\looparrowright} \mathbf{S}^2)$ is a completion. However, we do not know if the immersion $\mathcal{W} \looparrowright \mathbf{S}^2$ in Fig. 23 allows a decomposition of $\mathcal{W}$ into building blocks. (The other immersion does, as is discussed in Example 4.14 and Eq. (27).)

**Example 4.26** (Torus with an anyon)**.** Let $\mathcal{M} = T^2$ and $\mathcal{W} = T^2 \backslash B^2$. We take the immersion $\mathcal{W} \looparrowright \mathbf{S}^2$ shown in Fig. 18, which is the immersion that allows a vacuum block completion (27). However, for some topologically ordered systems, there are states $\rho_{\mathcal{W}}^a \in \mathrm{ext}(\Sigma_a(\mathcal{W}))$, where $a$ is a superselection sector of anyon that can exist on the puncture $\partial \mathcal{W}$, which is different from $\hat{1}$, the sector obtained in the vacuum block completion. By the completion trick, we can then obtain a state $|\varphi^a\rangle$ on $\mathcal{M}$.

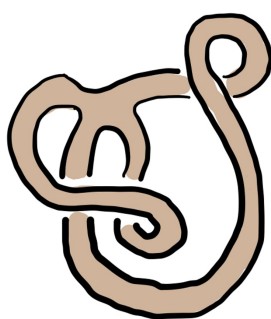

Figure 23: Another immersion of the punctured torus to $\mathbf{S}^2$. We think it is not regular homotopic to the one shown in the rightmost of Fig. 18.

The most interesting case is when $a$ is Abelian (and different from $\hat{1}$). In this case, the state $|\varphi^a\rangle$ on $\mathcal{M}$ satisfies axioms **A0** and **A1** everywhere on $\mathcal{M}$. A simple model where this happens is the Ising anyon model. $\mathcal{C} = \{1, \sigma, \psi\}$, where $\psi$ is Abelian and $\sigma$ is non-Abelian with $d_\sigma = \sqrt{2}$ (which should not be confused with the reference state $\sigma$). If we take $|\varphi^\psi\rangle$ as the reference state on $T^2$, one can verify

$$\dim \mathbb{V}(T^2) = \dim \mathbb{V}_\psi(\mathcal{W}) = 1, \quad |\varphi^\psi\rangle \text{ as the reference state on } T^2. \tag{37}$$

This is different from the answer we obtain from the vacuum block completion, which gives $\dim \mathbb{V}(T^2) = |\mathcal{C}| = 3$, where the reference state on $T^2$ is taken to be the canonical state associated with the vacuum block completion; we postpone the explanation of this fact in Example 6.1 in §6.1.

**Remark.** The Hilbert space dimension in both cases has a nice TQFT interpretation; see, e.g., Appendix E of [15]. In the Ising anyon model, we omitted the subtlety that the model is chiral. Its chiral central charge is $c_- = 1/2$. It is likely that the axioms can only hold approximately, although we expect the errors of the axioms to decay towards zero at large length scales in realistic models with finite correlation length, e.g., [15]. This phenomenon described in Example 4.26 also happens in zero correlation length models; see, e.g., Appendix E of [51] and [52].

**Example 4.27** (When $\mathcal{N}$ is an annulus with a defect line passes through)**.** Let $\mathcal{N}$ be an annulus, and suppose that a defect line[30] passes through it; see the gray annulus in Fig. 24. Let $\mathcal{M} = T^2$ be the torus and $\mathcal{W} = \mathcal{M} \setminus B^2$. with the immersion $\mathcal{W} \looparrowright \mathcal{N}$ in Fig. 24(a) and (b) by $\varphi_1$ and $\varphi_2$ respectively.

When a defect exists, $\mathcal{N}$ cannot be identified with any annulus subsystem of the sphere $\mathbf{S}^2$. (Recall that when we call a sphere $\mathbf{S}^2$ rather than $S^2$, we assume that there is a reference state on it, which satisfies **A0** and **A1** everywhere.) This also prevents the identification of the punctured tori $\mathcal{W}$ in Fig. 24(a) and (b) as regions immersed in $\mathbf{S}^2$. While the punctured tori $\mathcal{W}$ in the figures are divided into two building blocks, this example does not qualify as a vacuum block completion.

We explore the analog and distinction further. Interestingly, each subset of $\mathcal{W}$ (i.e., $\mathcal{V}_1$ or $\mathcal{V}_2^\diamond$) shown in Fig. 24(a) and (b) is still a building block. The reason is that each piece is contained in a disk. (While the 2-hole disk $\mathcal{V}_2^\diamond$ is not immersed in any embedded disk, it is immersed in an immersed disk!) The reason this construction is not a vacuum block completion is that the union $\mathcal{W}$ is not immersed in a disk or sphere. Nonetheless, following the technique behind vacuum block completion, we can learn a few things about this special type of completion in $\mathcal{N}$.

---

[30]Defects are located at the endpoints of the defect line. Anyons are permuted when crossing the defect line.

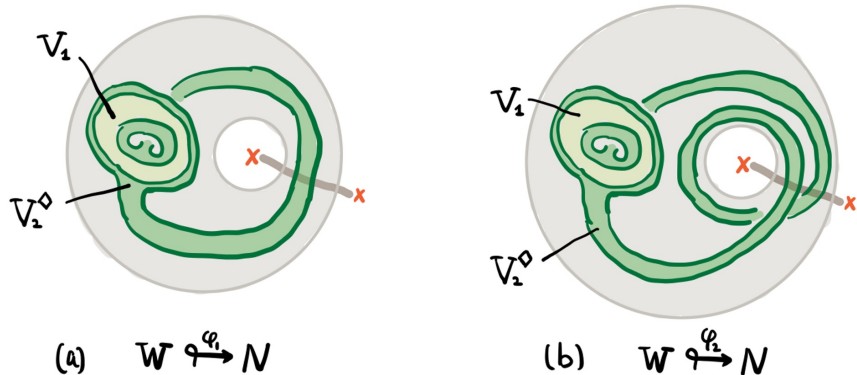

Figure 24: Completion in $\mathcal{N}$ of a torus. Here $\mathcal{N}$ is an annulus with a defect line passes through. $\mathcal{W} = T^2 \setminus B^2$. (a) $\mathcal{W} \overset{\varphi_1}{\looparrowright} \mathcal{N}$, and the decomposition of $\mathcal{W}$ into two building blocks. (b) $\mathcal{W} \overset{\varphi_2}{\looparrowright} \mathcal{N}$, and the decomposition of $\mathcal{W}$ into two building blocks.

1. The vacuum state on the building block can merge. The resulting state has an Abelian sector $\hat{1}$ on $\partial \mathcal{W}$.

2. After healing the puncture, we have a state on $T^2$ which satisfies the axioms everywhere. The Hilbert space dimension $\dim \mathbb{V}(T^2)$ for this reference state can be different from that of models without defects. For instance, if we take the defect to be the "duality wall" of the toric code [53], we should expect $\dim \mathbb{V}(T^2) = 2$, not 4.

The immersions in Fig. 24 generalize into a class of immersions that differ from one another by the "winding number" around the defect annulus; those shown in Fig. 24(a) and (b) correspond to the winding number equal to 1 and 2.

**Example 4.28** (Sequential completion). Here is another way to obtain states on a nontrivial manifold that satisfies the axioms everywhere. The idea is that we first start from a ball $\mathbf{S}^n$ or $\underline{\mathbf{B}}^n$ and construct a state on a relatively simple manifold $\mathcal{M}_1$, by vacuum block completion. Then, we make use of pieces of density matrices on $\mathcal{M}_1$ to construct more interesting manifolds $\mathcal{M}_2$. We can do it this repeatedly and obtain states on $\mathcal{M}_3, \mathcal{M}_4, \ldots$ We call this approach *sequential completion*.

Sequential completion is interesting for a few reasons. First, it gives a sequence of completions of $\mathcal{M}_{i+1}$ in $\mathcal{M}_i$. Second, the states so obtained satisfy the axioms everywhere on closed manifolds $\mathcal{M}_{i+1}$. More interestingly, as long as we avoid the completion points in $\mathcal{M}_i$ in constructing the state in $\mathcal{M}_{i+1}$, the construction can be translated into an immersion of $\mathcal{M}_{i+1}$ with punctures into $\mathbf{S}^n$ or $\underline{\mathbf{B}}^n$. For this reason, sequential completion can produce completions.

Below we explain the details of constructing a state on $\mathcal{M}_{i+1}$ from $\mathcal{M}_i$, using the idea of sequential completion. We are able to do the connected sum in the bulk and adjacent to the gapped boundary.

**Case 1, connected sums in the bulk:** Here we obtain $\#g\mathcal{M}$ from $\mathcal{M}$, where $\mathcal{M}$ is compact, connected, and may or may not have boundaries. (The way to obtain $\#g T^2$ from $T^2$ is illustrated in Fig. 25.) The idea is to remove a ball from $\mathcal{M}$ and call the resulting region $\mathcal{P}$. We take $g$ copies of $\mathcal{P}$ and denote the them separately as $\{\mathcal{P}_i\}_{i=1}^g$. We pick $\mathcal{V}$, a building block immersed in the ball, which is topologically a ball with $g$ balls removed, and it is immersed in such a way that $g$ of its $g+1$ entanglement boundaries are embedded. We merge the reduced density matrix of a state on $\mathcal{M}$ to the regions $\{\mathcal{P}_i\}_{i=1}^g$ and the state $\hat{\sigma}_{\mathcal{V}}$, such that the merging glues whole entanglement boundaries. The end result is a state on $\#g\mathcal{M}$ with a puncture in the bulk. The sector on the spherical entanglement boundary is Abelian (in fact, a special

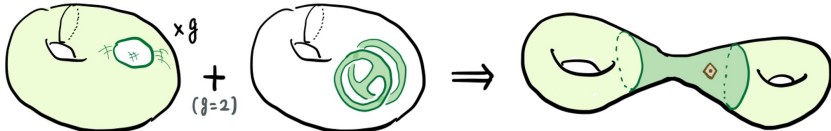

Figure 25: Sequential completion to obtain $\#g\,T^2$ from $T^2$. The case $g = 2$ is illustrated explicitly.

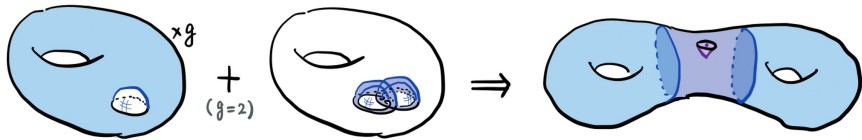

Figure 26: Sequential completion, to obtain a genus-$g$ handlebody with a gapped boundary. The case $g = 2$ is illustrated.

Abelian sector, which we referred to as $\hat{1}$). Therefore, we can heal the puncture as usual and obtain a state on $\#g\mathcal{M}$.

**Case 2, connected sums near the boundary:** We can also obtain $\natural g\mathcal{M}$ from $\mathcal{M}$, where $\mathcal{M}$ is compact, connected, and with a (connected) gapped boundary. (The case of obtaining a genus-$g$ handlebody with a gapped boundary from a solid torus with a gapped boundary is illustrated in Fig. 26.) Similarly, we have a set of regions $\{\mathcal{P}_i\}_{i=1}^{g}$, which are "identical copies" obtained by removing a ball $\underline{B}^3$ from the solid torus. We further need a region $\mathcal{V}$, which is a building block adjacent to the gapped boundary. $\mathcal{V}$ is a ball minus $g$ balls cut in half by the gapped boundary: it has $g+1$ half-sphere entanglement boundaries; it is immersed in $\underline{B}^n \subset \mathcal{M}$ in such a way that $g$ of these entanglement boundaries are embedded. We can therefore attach entire entanglement boundaries of $\mathcal{P}_i$ to $\mathcal{V}$ and obtain $\natural g\mathcal{M} \setminus \underline{B}^n$. Then we heal the puncture to obtain the closed manifold $\natural g\mathcal{M}$.

It is interesting to notice that the partition of all regions in Fig. 26, restricted to the neighborhood of the gapped boundary, look similar to regions in Fig. 25.

# 5 Pairing manifolds

Compact manifolds without entanglement boundaries are interesting in many ways. For instance, ground states on the torus in 2d encode the mutual braiding of anyons [54, 55] (see [16, 56–58] for other thoughts). In higher dimensions, one may hope that certain closed manifolds can provide insight into why one class of excitations should be paired with another class of excitations that braid nontrivially with it; this is termed as *remote detectibility* [7, 9, 11] and assumed for physical reasons. Moreover, as the existence of graph excitations (Fig. 9) shows, the understanding of remote detectability should require *a class of* compact manifolds. An important motivation we have in mind is to justify remote detectability in entanglement bootstrap for general dimensions.

For these purposes, we introduce a class of connected compact manifolds called *pairing manifolds*. Each of these pairing manifolds can be constructed using vacuum block completion, discussed in §4.1. Pairing manifolds satisfy a few extra conditions, from which we derive concrete information-theoretic constraints. We shall find many examples, in which the physical meaning of these constraints will be clear.

The structure of this section is as follows. In §5.1, we motivate the definition of pairing manifolds by asking a few questions. In §5.2, we give the general definition of the pairing

manifold (Definition 5.3). In §5.3, we derive several general consequences from the definition. Furthermore, in §6, we provide examples of pairing manifolds in a few physical setups and describe the explicit properties we derive for them.

## 5.1  Motivating questions and Kirby's torus trick

To motivate the notion of pairing manifolds (which we define in Definition 5.3), we discuss three examples and consider a few questions. The three examples are the torus $T^2$, the finite cylinder with a pair of gapped boundaries, and the manifold $(S^2 \times S^1)\#(S^2 \times S^1)$ in 3d. These examples are illustrated in Fig. 27.

As can be seen from the figure, common features of these examples include:

- There are two ways to partition the manifold into pieces: $\mathcal{M} = X\bar{X} = Y\bar{Y}$. Here $\bar{X} = \mathcal{M} \setminus X$ and $\bar{Y} = \mathcal{M} \setminus Y$ are the complements of $X$ and $Y$ on $\mathcal{M}$.

- The regions, $X$, $Y$, and their complements, cut each other into balls. These balls can be $B^n$ or $\underline{B}^n$ depending on whether we consider the gapped boundary.

Some natural questions arise: (1) Can we count the excitations characterized by $X$ in terms of those characterized by $Y$? (2) Is the "ground state degeneracy" on $\mathcal{M}$, i.e., $\dim \mathbb{V}_{\mathcal{M}}$, determined by the information convex set of either $X$ or $Y$? (3) Can the excitations characterized by $X$ remotely detect those characterized by $Y$? (4) Can we extract an analog of the $S$ matrix that characterizes the remote detectability?

As we shall explain, the pairing manifold provides a general (sufficient) condition for affirmative answers to all of these questions. Therefore, pairing manifolds are closely related to remote detectability in very general settings.

The intuition we develop is that the two ways of cutting $\mathcal{M}$ into pieces characterize two classes of excitations (or two classes of excitation clusters). One class of excitations is associated with the information convex set $\Sigma(X)$, and the other is associated with $\Sigma(Y)$. The fact that each of $X$ and $Y$ cut the other into balls means that they hide (classical and quantum) information from each other completely. Each region is associated with a natural choice of vacuum by immersion into a large ball or a sphere (possibly upon removing a ball). As a consequence of these relations, we will show that $X$ and $Y$ enjoy an uncertainty relation: when the state on $X$ is a minimum entropy state of $\Sigma(X)$ the state on $Y$ must be the maximum-entropy state on $\Sigma(Y)$ and vice versa (Lemma 5.9). $X$ and $Y$ provide two natural bases of states on the closed manifold, and therefore, a braiding matrix can be extracted, which shall be referred to as the pairing matrix (see Section 7). In general, this is not associated with any mapping class group, and therefore, our approach is a different generalization than the $S$ matrix on $T^3$ [22].

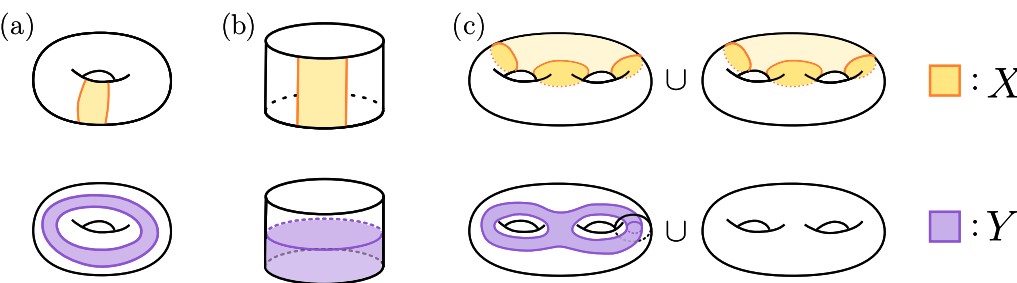

Figure 27: Three motivating examples ($\mathcal{M} = X\bar{X} = Y\bar{Y}$) for pairing manifolds. (a) The torus $T^2$ in 2d. (b) The finite cylinder with a pair of gapped boundaries, in 2d. (c) The manifold $(S^2 \times S^1)\#(S^2 \times S^1)$ in 3d. It is visualized as the gluing of a pair of solid genus-two handlebodies.

Finally, we comment on the use of immersion in our approach and its relation to Kirby's torus trick. Of conceptual interest is the fact that we only needed a reference state on a topologically trivial region, that is, either a ball or its completion to a sphere. By immersing the pairing manifolds (with a ball removed) into the trivial region, we are able to "pull back" several finite-dimensional consistency relations, and a finite-dimensional "pairing matrix". This is, in spirit, a close analog of Kirby's torus trick, which uses immersion of a compact manifold to pull back mathematical structures. (See [18] for a different application of Kirby's torus trick, which applies to the local Hamiltonian of gapped invertible phases.)

## 5.2 Pairing manifolds, the definition

In this section, we give a precise definition of *pairing manifolds*. For later convenience, for vacuum block completion $(\mathcal{M}, \{X, \bar{X}\}, \mathcal{W} \overset{\varphi}{\looparrowright} \mathbf{S}^n)$, we will use a short-hand notation for the canonical state

$$|1_X\rangle \equiv |1_{\{X, \bar{X}\}}\rangle, \tag{38}$$

where $|1_{\{X, \bar{X}\}}\rangle$ is the canonical state in the notation introduced in Definition 4.11. (Everything said here applies to the context with a gapped boundary once we replace $\mathbf{S}^n$ by $\underline{\mathbf{B}}^n$.) The notation on the left-hand side of Eq. (38) is well-motivated, noting that the constrained Hilbert spaces all contain a single element, i.e., $\dim \mathbb{V}_{[1_X]}(\mathcal{M}) = \dim \mathbb{V}_{[1_{\bar{X}}]}(\mathcal{M}) = \dim \mathbb{V}_{[1_{\partial X}]}(\mathcal{M}) = 1$. This simplification happens because $\mathcal{W}$ is decomposed into precisely two building blocks. To prepare the definition, we further introduce two concepts.

**Definition 5.1** (Transverse intersection)**.** Let $A$ and $B$ be two subsystems of a manifold $\mathcal{X}$. We say the intersection between $A$ and $B$ is transverse if

- It is possible to write $\partial A \cup \partial \bar{A}$ as a topological product $m \times \mathbb{I}$, where $\mathbb{I}$ is an interval, such that $B \cap (\partial A \cup \partial \bar{A}) = m_B \times \mathbb{I}$ for a suitable $m_B \subset m$. (Note that both $m$ and $m_B$ may be empty.)

- The same holds if we switch $A$ and $B$.

**Remark.** Our terminology "transverse intersection" comes from the analogous terminology used in the smooth category (see *e.g.* [59, 60]). In that context, a transverse intersection of submanifolds $A, B$ is one where the tangent vectors to $A$ and $B$ at each point of intersection span the tangent space of the ambient manifold $\mathcal{X}$.

Informally speaking, a transverse intersection between $A$ and $B$ is an intersection such that $B$ ($A$) intersects with the entanglement boundary of $A$ ($B$) in a non-singular way. In other words, there is enough room such that a small deformation of the entanglement boundary of $A$ will not change the topology of the intersection. Fig. 28(a), (b), and (c) are examples of transverse intersections between two 2d disk-like regions. The intersection between $A$ and $B$ in Fig. 28(d) is not transverse, and this can be seen from the fact that $B \cap \partial A$ and $B \cap \partial \bar{A}$ are topologically different.

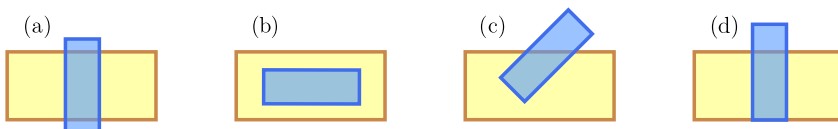

Figure 28: Transverse intersection, examples, and non-examples. $A$ is yellow, and $B$ is blue. For illustration purposes, they are chosen to be disk-like regions in the 2d bulk. (a), (b) and (c) are examples, whereas (d) is a non-example.

**Definition 5.2** (Natural partition)**.** Let $\Omega$ be an immersed region. We say the ordered triple $\{B, C, D\}$ is a natural partition of $\Omega$ for an element $\rho_\Omega \in \Sigma(\Omega)$, if

1. $C = \Omega \setminus \partial\Omega$ is the interior, and $BD = \partial\Omega$ is the thickened boundary.

2. $B$ and $D$ are of the form $m_B \times \mathbb{I}$ and $m_D \times \mathbb{I}$, where $\mathbb{I}$ is the interval appears in $\partial\Omega = m \times \mathbb{I}$ and $m = m_B \cup m_D$. Note that either $B$ or $D$ can be empty.

3. $\Delta(B, C, D)_\rho = 0$.

**Definition 5.3** (Pairing manifold, bulk)**.** A connected compact manifold $\mathcal{M}$ is a pairing manifold if it allows a pairing $(\mathcal{M}, \{X, \bar{X}\}, \{Y, \bar{Y}\}, \mathcal{W} \overset{\varphi}{\pitchfork} \mathbf{S}^n)$. Here, a pairing is the requirement that the following properties are satisfied:

1. $(\mathcal{M}, \{X, \bar{X}\}, \mathcal{W} \overset{\varphi}{\pitchfork} \mathbf{S}^n)$ is a vacuum block completion.

2. $(\mathcal{M}, \{Y, \bar{Y}\}, \mathcal{W} \overset{\varphi}{\pitchfork} \mathbf{S}^n)$ is another vacuum block completion whose canonical state on $\mathcal{W}$ agrees with the canonical state of the other vacuum block completion on $\partial\mathcal{W}$.

3. The intersection between $X$ and $Y$ is transverse.

4. $X \cap Y$, $\bar{X} \cap Y$, $X \cap \bar{Y}$ and $\bar{X} \cap \bar{Y}$ are balls.

5. $\{(\partial X) \cap Y, X \setminus \partial X, (\partial X) \cap \bar{Y}\}$ is a natural partition of $X$ for the canonical state $|1_X\rangle$; the same holds if we switch $X$ and $Y$.

**Remark.** The requirement that $X \cap Y$ is a ball ($B^n$) can be understood physically as the demand that one cannot detect any information stored in the information convex set $\Sigma(X)$ by analyzing the reduced density matrix on $X \cap Y$, the piece of $X$ that is available in $Y$. Therefore, the information in $X$ is hidden from $Y$ completely and vice versa. As before, the generalization of pairing manifold to the gapped boundary context is straightforward. We omit the precise statement since it can be obtained by replacing $\mathbf{S}^n$ by $\underline{\mathbf{B}}^n$, using the boundary version of building blocks (Definition 4.6), and allowing the "balls" in condition 4 to be either $B^n$ or $\underline{B}^n$.

In Definition 5.3, the following symmetries are either manifest or implied: (1) the switch of $X \leftrightarrow \bar{X}$, (2) the switch of $Y \leftrightarrow \bar{Y}$, and (3) the switch $X \leftrightarrow Y$ combined with $\bar{X} \leftrightarrow \bar{Y}$. To see why, we observe that the intersection between $X$ and $Y$ is transverse, implying that the intersections involving their complements ($X$ with $\bar{Y}$, $\bar{X}$ with $Y$, and $\bar{X}$ with $\bar{Y}$) are transverse. In addition, if $Y$ cuts $X$ into a natural partition, then it also cuts $\bar{X}$ into a natural partition for the state $|1_X\rangle$, thanks to purity.

In fact, because $X$ and $Y$ cut each other into balls, the natural partitions that appear in a pairing must have a special property, summarized in the following lemma:

**Lemma 5.4.** *Let $\{B, C, D\}$ be a natural partition of $\Omega$, with respect to a state $\rho_\Omega \in \Sigma(\Omega)$. Suppose, $B \subset \omega_1 \subset \Omega$ and $D \subset \omega_2 \subset \Omega$, where $\omega_1$ and $\omega_2$ are balls. Then,*

1. *$\rho_\Omega$ is a minimum entropy state of $\Sigma(\Omega)$.*

2. *the same $\{B, C, D\}$ is a natural partition for $\lambda_\Omega \in \Sigma(\Omega)$ if and only if $\lambda_\Omega$ is a minimum entropy state of $\Sigma(\Omega)$.*

*Proof.* Considering an arbitrary state $\rho'$ in $\Sigma(\Omega)$, we notice that

$$
\begin{aligned}
\Delta(B, C, D)_{\rho'} &= \Delta(B, C, D)_{\rho'} - \Delta(B, C, D)_\rho \\
&= \left(S(\rho'_{BC}) - S(\rho_{BC})\right) + \left(S(\rho'_{CD}) - S(\rho_{CD})\right) \\
&= 2\left(S(\rho'_\Omega) - S(\rho_\Omega)\right).
\end{aligned} \tag{39}
$$

The first line follows from the natural partition condition $\Delta(B,C,D)_\rho = 0$. In the second line, the canceled terms are due to the fact that $S(\rho'_B) = S(\rho_B)$ and $S(\rho'_D) = S(\rho_D)$: since $B$ and $D$ are contained in ball-like subsystems of $\Omega$, any state in $\Sigma(\Omega)$ has identical reduced density matrices on them. The third line follows from the fact that any smooth deformation of a region preserves the entropy difference (isomorphism theorem), and that it is possible to deform $BC$ and $CD$ to $\Omega$ by elementary steps of extensions.

Since $\Delta(B,C,D)_{\rho'} \geq 0$ by the strong subadditivity, $S(\rho'_\Omega) - S(\rho_\Omega) \geq 0$. Therefore, $\rho_\Omega$ is a minimum entropy state of $\Sigma(\Omega)$ – the first statement holds. The second statement also follows from Eq. (39). $\qquad\square$

Determining whether a manifold is a pairing manifold may not be easy. One reason is that verifying two vacuum block completions and showing that the second completion is compatible with the first one (i.e., verifying condition 2 of Definition 5.3) can be tricky. Luckily there are alternative sufficient conditions. We describe one below.

**Lemma 5.5** (Sufficient condition for pairing, bulk)**.** *Let $\mathcal{M}$ be a connected compact manifold and $(\mathcal{M},\{X,\bar{X}\},\mathcal{W} \overset{\varphi}{\looparrowright} \mathbf{S}^n)$ be a vacuum block completion. $\mathcal{M} = Y\bar{Y}$. Suppose furthermore that the following two groups of conditions hold:*

1. *Topology conditions:*

   (a) *$X$ intersects transversely with $Y$.*
   (b) *$X \cap Y$, $\bar{X} \cap Y$, $X \cap \bar{Y}$ and $\bar{X} \cap \bar{Y}$ are balls.*
   (c) *On $\mathcal{M}$, $X \cup Y = \mathcal{W} \setminus \partial\mathcal{W}$.*
   (d) *The immersion map $\varphi$ embeds $Y$ into $\mathbf{S}^n$.*

2. *Natural partition conditions:*

   (a) *$\{(\partial X) \cap Y, X \setminus \partial X, (\partial X) \cap \bar{Y}\}$ is a natural partition of $X$ for $|1_X\rangle$.*
   (b) *$\{(\partial Y) \cap X, Y \setminus \partial Y, (\partial Y) \cap \bar{X}\}$ is a natural partition of $Y$ for $\sigma_Y$. (Note that, by assumption 1(d), $Y$ is embedded, and $\sigma_Y$ is the reduced density matrix of the reference state.)*

*Then $(\mathcal{M},\{X,\bar{X}\},\{Y,\bar{Y}\},\mathcal{W} \overset{\varphi}{\looparrowright} \mathbf{S}^n)$ gives a pairing.*

The advantage of this sufficient condition is that we only need to check one vacuum block completion. Other properties (namely the group of topology conditions and the group of natural partition requirements) are relatively easy to check. The assumption that $Y$ is embedded in the sphere makes the range of application of Lemma 5.5 more limited than Definition 5.3. We note that Lemma 5.5 has the ability to verify vacuum block completion $\{\mathcal{M},\{Y,\bar{Y}\},\mathcal{W} \overset{\varphi}{\looparrowright} \mathbf{S}^n\}$ condition for some nontrivial choices of $Y$.

We postpone the proof of Lemma 5.5 to the end of §5.3.1.

## 5.3 Consequences of pairing

This section is a collection of consequences of the definition of pairing manifold (Definition 5.3). The main consequences are the uncertainty relation (Lemma 5.9), Propositions 5.10, 5.12 and Porism 5.13.

In §5.3.1, we prepare for the proof by showing what happens when a strict subset of the conditions is satisfied. These lemmas often use some of the conditions in pairing, not symmetric with respect to $X$ and $Y$. In §5.3.2, we prove the main consequences.

### 5.3.1 Prepare for the derivations

To prepare for the derivation, we prove a sequence of lemmas, each of which uses some of the conditions in the definition of pairing. In particular, Lemma 5.7 and 5.8 are about a state $|\Psi_X(\mathcal{M})\rangle$ defined as follows:

**Definition 5.6.** We let $|\Psi_X(\mathcal{M})\rangle$ be a state on a connected compact manifold $\mathcal{M}$ with the following three properties:

1. $\mathcal{M} = X\bar{X} = Y\bar{Y}$, such that topology conditions 1(a) and 1(b) of Lemma 5.5 are satisfied.

2. $|\Psi_X(\mathcal{M})\rangle$ satisfies axioms **A0** and **A1** everywhere on $\mathcal{M}$, and

3. $\{(\partial X) \cap Y, X \setminus \partial X, (\partial X) \cap \bar{Y}\}$ is a natural partition of $X$ for $|\Psi_X(\mathcal{M})\rangle$.

The state $|1_X\rangle$ satisfying the statements in Definition 5.3 or Lemma 5.5 satisfy the conditions of Definition 5.6, but Definition 5.6 does not refer to any immersion.

In the proofs of Lemma 5.7 and 5.8, transverse intersections play an important role. Here is a reminder of a related convention. When we pick a thickened entanglement boundary $\partial\Omega = m \times \mathbb{I}$, we always choose one that is thick enough so that it can be partitioned further into thinner pieces along the interval $\mathbb{I}$ when needed. Similarly, when we choose the thickened entanglement boundaries for a transverse intersection in proofs, we choose them such that these regions can be further extended outwards. This convention applies to the partitions in Fig. 29.

**Lemma 5.7.** *The state $|\Psi_X(\mathcal{M})\rangle$ in Definition 5.6, reduced to $Y$, gives the maximum-entropy state of $\Sigma(Y)$.*

*Proof.* We relabel the natural partition of $X$ for $|\Psi_X(\mathcal{M})\rangle$ from its transverse intersection with $Y$ as $\{B = (\partial X) \cap Y, C = X \setminus \partial X, D = (\partial X) \cap \bar{Y}\}$. Choose $A$ and $C_Y \subset C$ such that $Y = ABC_Y$. The partition is schematically shown in Fig. 29(a). On the state $|\Psi_X(\mathcal{M})\rangle$ we have

$$\begin{aligned}
I(A : C_Y | B) &\leq I(A : C | B) \\
&\leq \Delta(B, C, D) \\
&= 0.
\end{aligned} \tag{40}$$

The first line and the second line follow from SSA. The last line uses the fact that $\{B, C, D\}$ is a natural partition of $X$ for the state $|\Psi_X(\mathcal{M})\rangle$.

Therefore, $I(A : C_Y | B)_{\rho_Y^{|\Psi\rangle}} = 0$, where $\rho_Y^{|\Psi\rangle} \equiv \mathrm{tr}_{\bar{Y}} |\Psi_X(\mathcal{M})\rangle\langle\Psi_X(\mathcal{M})|$. Note that $Y = ABC_Y$, where $AB$ and $BC_Y$ are balls. ($AB$ is a ball because $A \equiv Y \cap \bar{X}$ is a ball, and the transverse intersection condition implies that $A$ can extend to $AB$ smoothly.) The Markov condition implies[31] that $\rho_Y^{|\Psi\rangle}$ is the maximum-entropy state of all states consistent with the state $|\Psi\rangle$ on balls $AB$ and $BC_Y$. Then, because $\rho_Y^{|\Psi\rangle} \in \Sigma(Y)$, and every state in $\Sigma(Y)$ has the same reduced density matrices on balls $AB$ and $BC_Y$, $\rho_Y^{|\Psi\rangle}$ must be the maximum-entropy state in $\Sigma(Y)$. $\qquad\square$

We continue the study of the state $|\Psi_X(\mathcal{M})\rangle$ in Definition 5.6. We have learned a couple of facts about this state.

- Its reduced density matrix on $X$ is a minimum entropy state on $\Sigma(X)$; Lemma 5.4.

- Its reduced density matrix on $Y$ is *the* maximum entropy state on $\Sigma(Y)$; Lemma 5.7.

---

[31]The basic statement we use here is that any quantum Markov state $\lambda_{XYZ}$ ($I(X : Z|Y)_\lambda = 0$) is the maximum entropy state among all states that agrees with $\lambda$ on marginals $XY$ and $YZ$.

- Its reduced density matrix on $X \cup Y$ is a minimum entropy state on $\Sigma(X \cup Y)$. This is because the complement of $X \cup Y$ on $\mathcal{M}$ is the ball $\bar{X} \cap \bar{Y}$, where the axioms hold on this ball. Axiom **A0** applied to this ball implies the extreme point condition for the state on $X \cup Y$. An extreme point is a minimum entropy state.

  Moreover, this is sufficient to guarantee that the superselection sector on the spherical boundary of $X \cup Y$ is Abelian, because the information convex set of the ball has only one element. Therefore, the entropy of this state on $X \cup Y$ is the absolute minimum among states in $\Sigma(X \cup Y)$.

Next, we discuss a useful uniqueness property related to $|\Psi_X(\mathcal{M})\rangle$. We borrow the notation of constrained Hilbert (fusion) space. Let $\Sigma_{[\rho_X^{|\Psi\rangle}]}(X \cup Y)$ be the convex subset of $\Sigma(X \cup Y)$ with the constraint that it matches $|\Psi_X(\mathcal{M})\rangle$ on $X$.

**Lemma 5.8.** *For the state $|\Psi_X(\mathcal{M})\rangle$ in Definition 5.6, $\Sigma_{[\rho_X^{|\Psi\rangle}]}(X \cup Y)$ contains a unique state, which is the reduced density matrix of $|\Psi_X(\mathcal{M})\rangle$.*

*Proof.* First, $\Sigma_{[\rho_X^{|\Psi\rangle}]}(X \cup Y)$ is nonempty because $\rho_{X \cup Y}^{|\Psi\rangle}$, the reduced density matrix of $|\Psi_X(\mathcal{M})\rangle$, is an element. We only need to show this is the only element. Consider the regions $XA'B'D'$ in Fig. 29(b), where $B'D' = \partial \bar{X}$. Let $\lambda_{X \cup Y} \in \Sigma_{[\rho_X^{|\Psi\rangle}]}(X \cup Y)$, then it is possible to do elementary steps of extension and obtain $\lambda_{XA'B'D'}$. (This uses the fact that the intersection between $X$ and $Y$ is transverse.) The natural partition condition (Def. 5.2) implies

$$0 = \Delta(B', X, D')_\lambda \geq I(A' : X | B')_\lambda, \tag{41}$$

where the inequality is a form of SSA. Therefore, $I(A' : X | B')_\lambda = 0$. As a quantum Markov state, $\lambda_{X \cup Y}$ is uniquely determined by the marginals on $XB'$ and $A'B'$. Because $A'B' = Y \cap \bar{X}$ is a ball, $\lambda$ reduced on it gives a unique state. Furthermore, since $\lambda$ is in the information convex set, its reduced density matrix on $XB'$ is determined by that on $X$, because $X$ can smoothly extend to $XB'$. (Again, this needs the transverse intersection condition.) Thus, $\lambda_{X \cup Y}$ is uniquely determined by its reduced density matrix on $X$, i.e., by $\rho_X^{|\Psi\rangle}$. This accomplishes the proof of the lemma. $\qquad \square$

Next, we give the proof of Lemma 5.5. Recall that $|1_X\rangle$ in Lemma 5.5 is a valid choice of $|\Psi_X(\mathcal{M})\rangle$. Furthermore, we use the notation of constrained fusion spaces: $\Sigma_{[1_X]}$ is the information convex set constrained to agree with $|1_X\rangle$ on $X$, and $\Sigma_{[\sigma_Y]}$ is the information convex set constrained to agree with $\sigma_Y$ on $Y$.

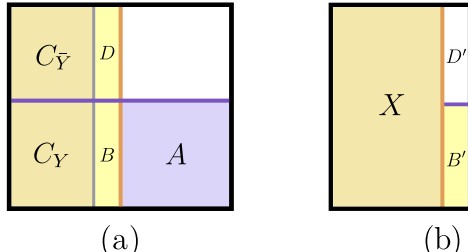
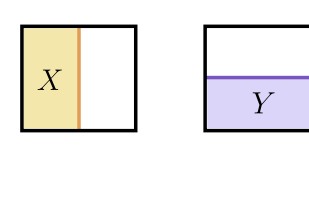

(a)          (b)

Figure 29: Schematic picture for two useful sets of subsystems on $\mathcal{M}$. (a) $BD = \partial X$, $B = Y \cap \partial X$, $Y = ABC_Y$. (b) $B'D' = \partial \bar{X}$ and $A'B' = Y \cap \bar{X}$. $X$ is on the left and $Y$ is at the bottom, and the orange and purple lines are their entanglement boundaries. The black box is not an entanglement boundary.

*Proof of Lemma 5.5.* We only need to check the second statement of Definition 5.3, i.e., the statement related to the vacuum block completion involving $Y$. Because $Y$ is embedded in $\mathbf{S}^n$, the vacuum state $\sigma_Y$ is defined.

Let $|1_X\rangle$ be the canonical state on $\mathcal{M}$ for $(\mathcal{M}, \{X, \bar{X}\}, \mathcal{W} \overset{\varphi}{\Phi} \mathbf{S}^n)$. The reduced density matrix of $|1_X\rangle$ on $Y$ must be the maximum entropy state on $\Sigma(Y)$. (This is by Lemma 5.7, noticing that $|1_X\rangle$ is a valid choice of state $|\Psi_X(\mathcal{M})\rangle$.) Because the maximum entropy state of the information convex set is a convex combination of all extreme points with nonzero probability, $|1_X\rangle$ reduces to $Y$, giving a nonzero probability of getting the reference state $\sigma_Y$.

Therefore, the set $\Sigma_{[\sigma_Y]}(\mathcal{M})$ is nonempty, where the reference state is taken to be $|1_X\rangle$. To see why this is true, we observe that the state $P^1_{\partial Y}|1_X\rangle$ is one element (not normalized) of $\Sigma_{[\sigma_Y]}(\mathcal{M})$, where $P^1_{\partial Y}$ is the projector to the vacuum sector on the embedded region $\partial Y$.

The remaining thing is to show $\bar{Y}^\diamond$ is a building block, and to show that the vacuum block completion involving $Y$ provides a canonical state that matches the other canonical state on $\partial \mathcal{W}$. The state $P^1_{\partial Y}|1_X\rangle$ satisfies the conditions of Def. 5.6 with the roles of $X$ and $Y$ reversed. Therefore, from Lemma 5.8 and the fact that $\mathcal{W}$ is the thickening of $X \cup Y$ on $\mathcal{M}$, we know $\Sigma_{[\sigma_Y]}(\mathcal{W})$ has a unique element. $\mathcal{W}$ is $Y$ and $\bar{Y}^\diamond$ glued on whole entanglement boundaries. By the associativity theorem, $\dim \mathbb{V}_{[1_{\partial \bar{Y}}]}(\bar{Y}^\diamond) = 1$. In other words, if we choose the vacuum sector on the embedded entanglement boundary $\partial \bar{Y}$ (embedded in $\mathbf{S}^n$) of $\bar{Y}^\diamond$, we get an isolated extreme point of $\Sigma(\bar{Y}^\diamond)$, and the sector on $\partial \mathcal{W}$ must be determined uniquely. This verifies that $\bar{Y}^\diamond$ is a building block, and further shows that the $\hat{1}$ on $\partial \mathcal{W}$ for the vacuum block completion $(\mathcal{M}, \{Y, \bar{Y}\}, \mathcal{W} \overset{\varphi}{\Phi} \mathbf{S}^n)$ matches the sector on $\partial \mathcal{W}$ from the other vacuum block completion $(\mathcal{M}, \{X, \bar{X}\}, \mathcal{W} \overset{\varphi}{\Phi} \mathbf{S}^n)$. This completes the proof of Lemma 5.5. $\qquad\square$

### 5.3.2 The main consequences of pairing

The consequences of pairing stated in this section are based on the full definition of pairing manifold. Thus, they are properties of pairing

$$(\mathcal{M}, \{X, \bar{X}\}, \{Y, \bar{Y}\}, \mathcal{W} \overset{\varphi}{\Phi} \mathbf{S}^n). \tag{42}$$

All these statements generalize naturally to the context with gapped boundaries.

We start with the "uncertainty relation" between the state on $X$ and the state on $Y$. When we discuss information convex sets of subregions of $\mathcal{M}$ we implicitly take either $|1_X\rangle$ or $|1_Y\rangle$ as the reference state. This covers the observation in Refs. [61,62] as special cases.

**Lemma 5.9** (Uncertainty relation)**.** *Given a pairing (42), the state $|1_X\rangle$ reduced to $Y$ gives the maximum-entropy state of $\Sigma(Y)$. The same holds if we switch $X \leftrightarrow Y$ and/or $X \leftrightarrow \bar{X}$.*

*Proof.* We only need to prove one of the four statements. The proofs of other statements are identical. The conclusion follows from Lemma 5.7, upon realizing that $|1_X\rangle$ is a valid choice of the state $|\Psi_X(\mathcal{M})\rangle$ there. This completes the proof. $\qquad\square$

**Remark.** Recall that $|1_X\rangle$ reduces to a minimum entropy state on $X$. Furthermore, any minimum entropy state on $\Sigma(X)$ has the same natural partition (Lemma 5.4). The uncertainty relation then implies that any pure state in $\Sigma(\mathcal{M})$ which reaches the minimum entropy on $X$ must reduce to the maximum entropy state of $\Sigma(Y)$.

We shall denote the maximum-entropy state of $\Sigma(X)$ as $\rho_X^\star$. A useful fact (which follows from the Structure Theorem of [4,5]) is that the probability to find sector $I \in \mathcal{C}_{\partial X}$ on $\partial X$ in the state $\rho_X^\star$ is:

$$P_I(X) = \frac{N_I(X)d_I}{\sum_{J \in \mathcal{C}_{\partial X}} N_J(X)d_J}, \quad \forall I \in \mathcal{C}_{\partial X}. \tag{43}$$

Here $d_I$ is the quantum dimension of the superselection sector $I \in \mathcal{C}_{\partial X}$, defined in Eq. (7). (Note that we can use Eq. (7) because $\partial X$ is embedded in $\mathbf{S}^n$. Recall that $X$ here participates in a pairing manifold, and it is a building block with its immersed boundary filled in, according to Def. 4.9.)

**Proposition 5.10.** *Given a pairing (42), $\Sigma(X) \cong \Sigma(\bar{X})$ and $\Sigma(Y) \cong \Sigma(\bar{Y})$. Moreover, the fusion spaces match in each sector:*

$$
\begin{aligned}
\dim \mathbb{V}_a(X) &= \dim \mathbb{V}_a(\bar{X}), \quad \forall a \in \mathcal{C}_{\partial X}, \\
\dim \mathbb{V}_\alpha(Y) &= \dim \mathbb{V}_\alpha(\bar{Y}), \quad \forall \alpha \in \mathcal{C}_{\partial Y}.
\end{aligned}
\tag{44}
$$

*Furthermore,*[32]

$$
\sum_{a \in \mathcal{C}_{\partial X}} \dim \mathbb{V}_a(X)^2 = \sum_{\alpha \in \mathcal{C}_{\partial Y}} \dim \mathbb{V}_\alpha(Y)^2 = \dim \mathbb{V}(\mathcal{M}).
\tag{45}
$$

In Eq. (44), we have adopted an obvious isomorphism between $\mathcal{C}_{\partial X}$ and $\mathcal{C}_{\partial \bar{X}}$ as follows. Considered is the isomorphism $\Sigma(\partial X) \cong \Sigma(\partial \bar{X})$ by the following path: first, extend $\partial X$ to $\partial X \partial \bar{X}$ and then restrict the region to $\partial \bar{X}$. This path always exists because $\partial X$ and $\partial \bar{X}$ lie on the opposite sides of an entanglement boundary. The same convention is adopted for matching the sector labels of $\mathcal{C}_{\partial Y}$ and $\mathcal{C}_{\partial \bar{Y}}$.

*Proof.* To derive Eq. (44), we consider the matching of probabilities of finding $a \in \mathcal{C}_{\partial X}$ on the entanglement boundaries $\partial X$ and $\partial \bar{X}$, in a particular state. To prove the first line of Eq. (44), we consider the state $|1_Y\rangle$. According to Lemma 5.9, it reduces to the maximum entropy state $\rho_X^\star$ and $\rho_{\bar{X}}^\star$. Therefore, the probability of having $a \in \mathcal{C}_{\partial X}$ on $\partial X$ for $\rho_X^\star$ and $\partial \bar{X}$ for $\rho_{\bar{X}}^\star$ must match. The rest of the analysis follows from Eq. (43). First, we take $a = 1$, $d_1 = 1$. It follows from the properties of vacuum block completion that $N_1(X) = N_1(\bar{X}) = 1$. Plugging in Eq. (43) and $P_1(X) = P_1(\bar{X})$, we find

$$
\sum_{J \in \mathcal{C}_{\partial X}} N_J(X) d_J = \sum_{J \in \mathcal{C}_{\partial X}} N_J(\bar{X}) d_J.
\tag{46}
$$

On the right-hand side, we wrote $\mathcal{C}_{\partial X}$ instead of $\mathcal{C}_{\partial \bar{X}}$ because the two sets are identical. Now take an arbitrary $a \in \mathcal{C}_{\partial X}$ and plug it into Eq. (43) again. With Eq. (46) canceling the denominator, we find $N_a(X) d_a = N_a(\bar{X}) d_a$ for any $a \in \mathcal{C}_{\partial X}$. This implies the first line of Eq. (44), since $N_a(X) \equiv \dim \mathbb{V}_a(X)$. The second line of Eq. (44) is derived similarly.

From the associativity theorem, we have

$$
\dim \mathbb{V}(\mathcal{M}) = \sum_{a \in \mathcal{C}_{\partial X}} \dim \mathbb{V}_a(X) \cdot \dim \mathbb{V}_a(\bar{X}) = \sum_{\alpha \in \mathcal{C}_{\partial Y}} \dim \mathbb{V}_\alpha(Y) \cdot \dim \mathbb{V}_\alpha(\bar{Y}).
\tag{47}
$$

Plugging in Eq. (44), we obtain Eq. (45). $\qquad \square$

The next few statements are best understood with the concept "capacity", introduced as follows.

**Definition 5.11** (Capacity). We define the capacity of an immersed region $\Omega$, as

$$
\mathcal{Q}(\Omega) = \exp\left(S_{\max}(\Omega) - S_{\min}(\Omega)\right),
\tag{48}
$$

where $S_{\max}(\Omega)$ is the entropy of the maximum entropy state in $\Sigma(\Omega)$ and $S_{\min}(\Omega)$ is the entropy of a minimum entropy state of $\Sigma(\Omega)$.

---

[32]Here, $\mathbb{V}(\mathcal{M})$ is the Hilbert space whose state space (i.e., space of density matrices) is isomorphic to $\Sigma(\mathcal{M})$.

The physical meaning of capacity is the exponential of the entropy (this is the effective number of states) that $\Omega$ can absorb without changing the local reduced density matrix. As with the information convex set $\Sigma(X)$, the capacity $\mathcal{Q}(\Omega)$ depends on the choice of the reference state.

If $\Omega$ is embedded in a ball (or more generally, immersed but has all its entanglement boundaries embedded, aligned with the context of Eq. (43)), we can write

$$\mathcal{Q}(\Omega) = \sum_{J \in \mathcal{C}_{\partial \Omega}} N_J(\Omega) d_J \,. \tag{49}$$

If $S$ is sectorizable, the capacity is the square of the "total quantum dimension for the sectorizable region $S$":

$$\mathcal{Q}(S) = \sum_{a \in \mathcal{C}_S} d_a^2 \,. \tag{50}$$

For a compact manifold $\mathcal{M}$ (i.e., without entanglement boundaries), the capacity is the Hilbert space dimension, $\mathcal{Q}(\mathcal{M}) = \dim \mathbb{V}(\mathcal{M})$.

Proposition 5.12 below says, given a pairing (42), the capacity must be the same for both $X$ and $Y$. Porism 5.13 below implies that if there are two pairings of the same $\mathcal{M}$, where one involves $X, Y$ and another involves $X', Y'$, the capacity of $X$ is the same as that of $X'$, under an extra condition.

**Proposition 5.12.** *Given a pairing* (42)*, the Hilbert space dimensions obey*

$$\sum_{a \in \mathcal{C}_{\partial X}} \dim \mathbb{V}_a(X) \cdot d_a = \sum_{\alpha \in \mathcal{C}_{\partial Y}} \dim \mathbb{V}_\alpha(Y) \cdot d_\alpha \,. \tag{51}$$

In other words, the capacity for $X$ and $Y$ match, $\mathcal{Q}(X) = \mathcal{Q}(Y)$.

*Proof.* On $\mathcal{M}$, let $V = X \cup Y$. To derive Eq. (51), we consider two different ways to compute $S_{\max}(V) - S_{\min}(V)$, which is the entropy difference between the maximum entropy state[33] ($\rho_V^\star$) and any minimum entropy state of $\Sigma(V)$. Note that there can be multiple minimum-entropy states, and two will be useful soon: (i) the reduced density matrix of $|1_X\rangle$ on $V$, which we denote as $\rho_V^{|1_X\rangle}$, and (ii) the reduced density matrix of $|1_Y\rangle$ on $V$, which we denote as $\rho_V^{|1_Y\rangle}$.

To compute the entropy difference $S(\rho_V^\star) - S(\rho_V^{|1_X\rangle})$, we consider the partition $V = XA'B'$ in Fig. 29(b). We notice that both states are quantum Markov states for this partition, $I(A' : X|B') = 0$, and they have identical reduced density matrix on $A'B'$. (For the maximum entropy state $\rho_V^\star$, we know it is the merged state as follows, $\rho_V^\star = \rho_{XB'}^\star \bowtie \sigma_{A'B'}$. Therefore,

$$
\begin{aligned}
S\left(\rho_V^\star\right) - S\left(\rho_V^{|1_X\rangle}\right) &= S\left(\rho_{XB'}^\star\right) - S\left(\rho_{XB'}^{|1_X\rangle}\right) \\
&= S\left(\rho_X^\star\right) - S\left(\rho_X^{|1_X\rangle}\right) \\
&= \ln\left(\sum_{a \in \mathcal{C}_{\partial X}} \dim \mathbb{V}_a(X) \cdot d_a\right).
\end{aligned}
\tag{52}
$$

In the first line, we used the quantum Markov chain condition to write $S_V = S_{XB'} + S_{A'B'} - S_{B'}$ for both states and cancel the terms on $A'B'$ and $B'$. In the second line, we used the fact that $X$ can be deformed to $XB'$ by elementary steps of extensions. The third line follows from formula (49) for the capacity, noticing that $\rho_X^{|1_X\rangle}$ is a minimum entropy state of $\Sigma(X)$. A parallel analysis gives $S(\rho_V^\star) - S(\rho_V^{|1_X\rangle}) = \ln(\sum_{\alpha \in \mathcal{C}_{\partial Y}} \dim \mathbb{V}_\alpha(Y) \cdot d_\alpha)$. By comparing these two expressions, we are able to verify the claim. $\qquad \square$

---

[33]The maximum entropy state is always unique, for any convex set. If there were more than one, a higher-entropy state could be made by mixing them.

**Remark.** An analog of Eq. (51) is true under weaker assumptions about $X$ and $Y$. We leave this generalization, which is relevant for the punctured $S$-matrix or the defect $S$-matrix of [63], for future work.

If a region $Y$ participating in a pairing manifold is embedded, its capacity $\mathcal{Q}(Y)$ can be related to the topological entanglement entropy (TEE) of the phase of matter, as defined in [64]. Specifically, the partition of $Y = ABC_Y$ with $C_Y = Y \cap (X \setminus \partial X)$, $B = Y \cap \partial X$, and $A = Y \setminus X$ as in Fig. 29(a) gives

$$\text{TEE} = I(A : C_Y | B)_{\sigma_Y} = \ln \mathcal{Q}(Y). \tag{53}$$

For the case of $\mathcal{M} = T^2$, this is the Levin-Wen partition of the annulus. Eq. (53) follows because

$$I(A : C_Y | B)_{\sigma_Y} = -I(A : C_Y | B)|_{\sigma_Y}^{\rho_Y^\star} = S_Y|_{\sigma_Y}^{\rho_Y^\star} = \ln \mathcal{Q}(Y), \tag{54}$$

where $\rho_Y^\star$ is the maximum entropy state in $\Sigma(Y)$. In the first step, we used the fact that $X$ cuts $Y$ into a natural partition:

$$0 = \Delta(B, C, D)_{|1_X\rangle} \stackrel{\text{SSA}}{\geq} I(A : C | B)_{|1_X\rangle} \stackrel{\text{SSA}}{\geq} I(A : C_Y | B)_{|1_X\rangle} \stackrel{\text{Lemma 5.9}}{=} I(A : C_Y | B)_{\rho_Y^\star}. \tag{55}$$

In the second step of (54) we used the fact that $AB$ and $BC_Y$ are balls and $B$ is contained in a ball.

When both $X$ and $Y$ are embedded, Proposition 5.12 implies that the TEE computed from $X$ is the same as that computed from $Y$. We shall give yet another general perspective of TEE in [38], recast into $\Delta(B, C, D)_\sigma$ for some $BCD$ partition of a ball. These partitions are analogs of a partition considered by Kitaev [65]; see around the 27th minute. Also see [66, 67] for examples with domain walls or gapped boundaries.

A consequence of the proof of Proposition 5.12 is the following:

**Porism 5.13.** If regions $X$ and $X'$ participate in pairing manifolds with the same $\mathcal{W}$, with canonical states in the same sector of $\Sigma(\partial \mathcal{W})$, then they have the same capacity,

$$\sum_{a \in \mathcal{C}_{\partial X}} d_a N_a(X) = \sum_{a' \in \mathcal{C}_{\partial X'}} d_{a'} N_{a'}(X'). \tag{56}$$

*Proof.* In the proof of Proposition 5.12, we saw that the TEE is a property of the immersion $\mathcal{W}$: it is the entropy difference between the maximum-entropy state on $V$ and any minimum-entropy state on $V$. The immersion $\mathcal{W}$ is an extension of $V$. $\qquad\square$

A non-trivial example we will give in the next section (Example 6.5) is $X =$ genus-two handlebody (or $X =$ ball minus two balls) and $X' =$ ball minus torus; each of these regions participates in a certain pairing of the pairing manifold $\#2(S^2 \times S^1)$, with the same immersion of $\mathcal{W} = \#2(S^2 \times S^1) \setminus B^3$, and therefore all have the same capacity.

# 6 Examples of pairing manifolds

Here we demonstrate that the examples in Table 1 all satisfy the requirements in the definition of pairing manifold. These include examples in different space dimensions as well as cases with gapped boundaries. It is worth noting that not every compact manifold is a pairing manifold (see §6.4), and a manifold can be a pairing manifold in more than one way (see Example 6.5).

This section is organized as follows. In §6.1, we present examples of pairing manifolds in the bulk, namely, $(\mathcal{M}, \{X, \bar{X}\}, \{Y, \bar{Y}\}, \mathcal{W} \stackrel{\varphi}{\looparrowright} \mathbf{S}^n)$. In §6.2, we present examples of pairing manifolds for systems with gapped boundaries. They are of the form $(\mathcal{M}, \{X, \bar{X}\}, \{Y, \bar{Y}\}, \mathcal{W} \stackrel{\varphi}{\looparrowright} \underline{\mathbf{B}}^n)$.

In §6.3, we verify all the natural partition requirements used in these examples. In §6.4, for pedagogical reasons, we provide various non-examples.

In our way of presenting the examples, we develop some concise diagrams which contain all the relevant information of a pairing manifold in a compact way. The rules of the diagram are explained below and we will use these rules throughout the section. Some efforts are taken to verify that the conditions for pairing manifolds are satisfied in these examples. Furthermore, we translate the consequences of pairing manifolds (described in §5.3) to these concrete examples and explain their physical meaning.

The following notation for immersed regions will be useful. In the examples below, we will form immersed unions of regions $X$ and $Y$ such that they only intersect on a ball $\bullet$; elsewhere, they are disjoint, even when they occupy the same point in the ambient space. More precisely, we define

$$X \cup_\bullet Y \equiv X \amalg Y / \sim , \tag{57}$$

where $X \amalg Y$ denotes the disjoint union, and the equivalence relation identifies[34] each point in $X \cap \bullet$ with the corresponding point in $Y \cap \bullet$. In the figures with room for ambiguity, we will color the shared region of $X$ and $Y$ using this color.

For each example, we show two kinds of diagrams:

1. The global view: This is a figure with $\mathcal{M}$ shown and an indication of how it is partitioned into $X\bar{X}$ and $Y\bar{Y}$. Also shown are the completion point(s). It is understood that the complement of the completion point(s) is $\mathcal{W}$. This diagram, however, misses the information about the immersion map $\varphi$. (The illustration of the completion point is convenient because any region immersed in $\mathcal{M}$ that does not touch the completion points must be immersed in $\mathbf{S}^n$ or $\underline{\mathbf{B}}^n$, whichever applies.)

2. The immersion view: it contains the information of the immersion $X \cup_\bullet Y \overset{\varphi}{\looparrowright} \mathbf{S}^n$ (or $\underline{\mathbf{B}}^n$). For all the examples we discuss in this section, $\mathcal{W}$ is the thickening[35] of $X \cup_\bullet Y$ and therefore the regular homotopy class of $\mathcal{W} \overset{\varphi}{\looparrowright} \mathbf{S}^n$ (or $\underline{\mathbf{B}}^n$) is implied, so does the way $X^\diamond, \bar{X}^\diamond$ and $Y^\diamond, \bar{Y}^\diamond$ are contained in $\mathcal{W}$.[36] We put a point of $\mathbf{S}^n$ at infinity, whenever this is convenient. In fact, the immersion view by itself contains all the information about the pairing manifold. For instance, it determines the topology of $\mathcal{M}$ because there is a unique topology obtainable by healing all the punctures of $\mathcal{W}$.

In the illustrations of the global view and the immersion view, in this section, we will use the following color setting. $X$ is orange and $Y$ is purple. The intersection of $X$ and $Y$ (i.e., the place we take the immersed union) is green. The place where $X$ and $Y$ "pass through" each other (without touching) is either illustrated as one region on top and covers the other or shown with transparency.

These diagrammatic representations of pairing manifolds take some practice to get used to, and we shall be pedagogical in some early examples by explicitly drawing the natural partition obtained from them. The justification of all the natural partition conditions in the remaining examples uses similar ideas, and we collect these partitions in §6.3.

---

[34]This identification can be thought of as an analog of the "plumbing" used extensively in topology [68]; see Page 129 therein.

[35]If $\mathcal{W}$ is not the thickening of $X \cup_\bullet Y$, the immersion view will, instead, indicate the immersion of $\mathcal{W}$ into the ball or sphere.

[36]In all the examples below, we are able to choose $X, Y \subset \mathcal{W}$.

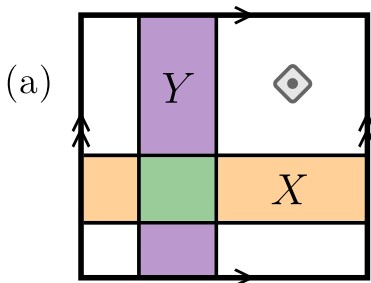
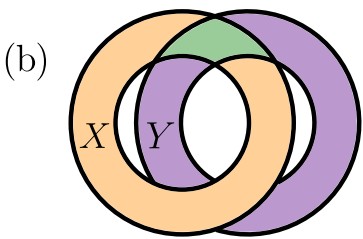

Figure 30: A pairing for the pairing manifold $\mathcal{M} = T^2$. (a) The global view. $T^2$ is the box with opposite boundaries identified. Regions $X$ and $Y$ are annuli with a transverse intersection at a ball ($B^2$) and there is a single completion point. $X$ is orange, $Y$ is purple, with the intersection $X \cap Y$ colored green. We will use this color setting throughout §6. (b) The immersion view. $X \cup_{\bullet} Y \overset{\varphi}{\pitchfork} \mathbf{S}^2$ is explicitly shown, where a point of $\mathbf{S}^2$ is taken to be the point of infinity of $\mathbb{R}^2$. ($\mathcal{W}$ is the thickening of $X \cup_{\bullet} Y$, which we omitted in the drawing for simplicity.)

## 6.1 Pairing manifold examples in the bulk

In this section, we provide examples of pairing manifolds in the 2d and 3d bulk. In these physical contexts, a connected compact manifold $\mathcal{M}$ is a pairing manifold if it allows a pairing

$$\left(\mathcal{M}, \{X, \bar{X}\}, \{Y, \bar{Y}\}, \mathcal{W} \overset{\varphi}{\pitchfork} \mathbf{S}^n\right). \tag{58}$$

The 2d and 3d cases correspond to space dimensions $n = 2$ and $n = 3$. In all the examples below, Lemma 5.5 is a more convenient list of properties to check compared with Definition 5.3.

**Example 6.1** (Torus $T^2$). The torus $\mathcal{M} = T^2$ is a pairing manifold. This is because it is possible to construct a pairing (58) for it. See Fig. 30(a) for the global view and 30(b) for the immersion view. Below, we check all the conditions of Lemma 5.5.

First, $(\mathcal{M}, \{X, \bar{X}\}, \mathcal{W} \overset{\varphi}{\pitchfork} \mathbf{S}^2)$ is a vacuum block completion. This follows from Example 4.14 in §4.2. The essential idea is recalled here for readers' convenience. From the immersion view, we can see the topology of $X$ and $\bar{X}^{\diamond}$ and verify the fact that they are building blocks. This is because $X$ is an embedded annulus, and $\bar{X}^{\diamond}$ is a 2-hole disk immersed in such a way that two of its three entanglement boundaries are embedded. Thus, $(\mathcal{M}, \{X, \bar{X}\}, \mathcal{W} \overset{\varphi}{\pitchfork} \mathbf{S}^2)$ is a vacuum block completion. Second, the conditions about the topological relations between regions (conditions 1) of Lemma 5.5 can be verified directly from Fig. 30. Lastly, we need to verify that the intersection between $X$ and $Y$ gives two natural partitions. We explain this in detail below.

First, we infer the partition for $X$ and $Y$ from the immersion view (Fig. 30(b)). These partitions, which we refer to as $X = BCD$ and $Y = B'C'D'$ are illustrated in Fig. 31. Then it is easy to explain the fact that

$$\Delta(B, C, D)_{\sigma} = 0, \quad \Delta(B', C', D')_{\sigma} = 0. \tag{59}$$

These are true because $\sigma$ is the vacuum state and the right-hand side of each equality is $4 \ln d_1 = 0$. (Here we are using the relation (8) for the quantum dimension.) This completes the checking of all the conditions in Lemma 5.5. Therefore, Fig. 30 indeed describes a pairing for the torus $T^2$.

Now we have verified that $T^2$ is a pairing manifold, we discuss a consequence: $|\mathcal{C}| = \dim \mathbb{V}(T^2)$. This agrees with the intuition from minimally entangled states [54]: for each sector on $X \subset T^2$, we have precisely one state reduced to the extreme point $\rho_X^a \in \Sigma(X)$.

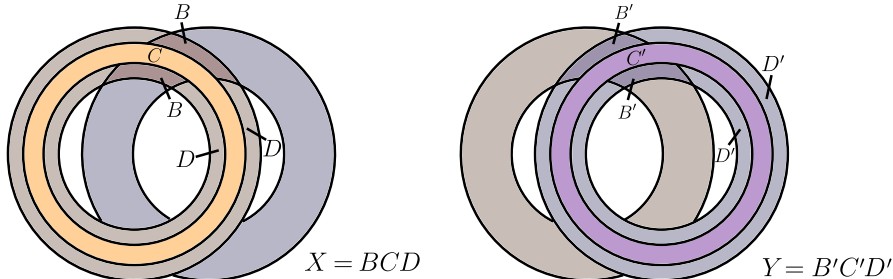

Figure 31: The natural partition of $X$ and $Y$, obtained from the transverse intersection between them. This information can be read off from the immersion view, Fig. 30(b).

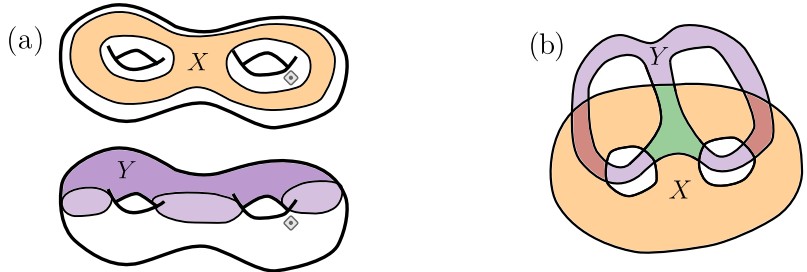

Figure 32: The genus $g$ (here $g = 2$) Riemann surface is a pairing manifold. Illustrated here is a particular pairing. (a) The global view. (b) The immersion view. Here $X$ and $Y$ are 2-hole disks.

This set of states has a clear TQFT interpretation and the state on the torus we construct is free from topological defects.

**Remark.** Our requirement that $X$ participates in a vacuum block completion excludes the possibility of adding a Dehn twist. Let $X^{\mathrm{Dehn}}$ be an annulus obtained from $X$ by adding a finite number of Dehn twists on $T^2$, and suppose that the completion point is away from $X^{\mathrm{Dehn}}$. Then $X^{\mathrm{Dehn}}$ is immersed in $\mathbf{S}^2$ in such a way that both boundaries of annulus $X^{\mathrm{Dehn}}$ are immersed instead of embedded. Then $X^{\mathrm{Dehn}}$ cannot be a building block. On the other hand, once we construct $\mathcal{W}$, immersed in $\mathbf{S}^2$ as in Fig. 30(b), we can deform $\mathcal{W}$ on $\mathbf{S}^2$ to get more interesting immersions. This, however, does not exhaust all possible immersions of the punctured torus; see Example 4.25 for a related discussion. We left the relation between such transformations and the mapping class group of $T^2$ as an open problem.

**Example 6.2** (Genus-$g$ Riemann surface, $\#_g T^2$). A genus-$g$ Riemann surface $\mathcal{M} = \#_g T^2$ is a pairing manifold. See Fig. 32 for the explicit construction of a pairing. For illustration purposes, we take $g = 2$ for concreteness. From the global view, we see that $X$ and $Y$ are 2-hole disks. They are obtained by cutting the donut either "horizontally" or "vertically". There is precisely one completion point, located in $\bar{X} \cap \bar{Y}$.

The immersion view shows how the pair of 2-hole disks $X$ and $Y$ immerse in $\mathbf{S}^2$, and in this example, they are embedded. The region $\mathcal{W}$ is the thickening of $X \cup_\bullet Y$, it is immersed in the sphere in a nontrivial way. We omit the explicit drawing of $\mathcal{W}$ for simplicity since it can be inferred from $X \cup_\bullet Y$. The fact that $(\mathcal{M}, \{X, \bar{X}\}, \mathcal{W} \overset{\varphi}{\looparrowright} \mathbf{S}^2)$ is a vacuum block completion can be checked easily from the definition. (In fact, this follows from Example 4.15 in §4.2.) Requirements about the topological relations between regions (conditions 1 of Lemma 5.5) can be verified by looking at Fig. 32. Explicitly, $X$ and $Y$ slice each other into two balls, and the intersection between $X$ and $Y$ is transverse.

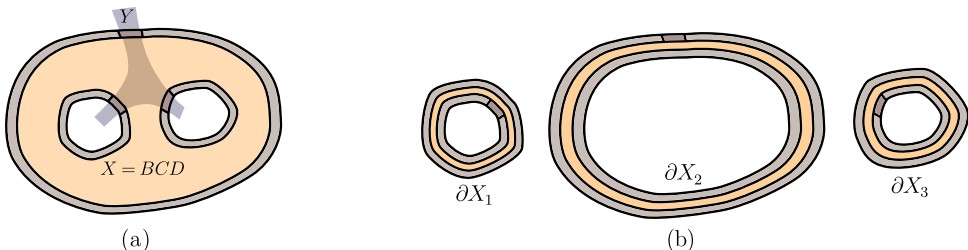

Figure 33: Natural partitions for $g = 2$. (a) $X = BCD$ according to its transverse intersection with $Y$. Only part of $Y$ is shown. (b) Using the decoupling lemma (Lemma D.1 of [6]), the problem can be simplified into a set of natural partitions on annuli, $\{\partial X_i\}_{i=1}^3$, where $\cup_{i=1}^3 \partial X_i = \partial X$.

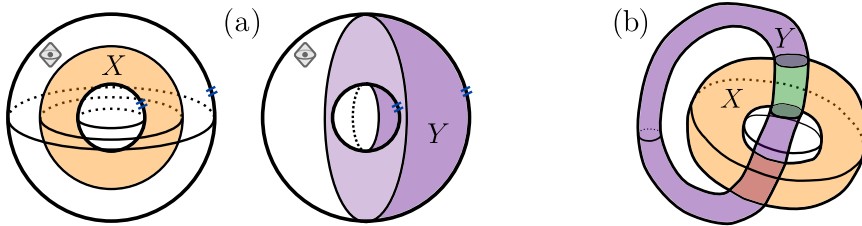

Figure 34: A way to make $\mathcal{M} = S^2 \times S^1$ a pairing manifold. (a) The global view. (b) The immersion view. $X$ is a sphere shell, $Y$ is a solid torus. $\mathcal{W}$ is the thickening of the immersed union $X \cup_\bullet Y$.

To see that the intersection between $X$ and $Y$ gives two natural partitions (conditions 2a and 2b of Lemma 5.5), we only need to consider the decomposition of $X = BCD$ according to its intersection with $Y$ and show $\Delta(B, C, D)_\sigma = 0$. This partition is illustrated in Fig. 33(a). (The decomposition of $Y$ is identical topologically.) A simplification is that $\partial X$ has three connected components, each of which is an annulus, $\partial X = \partial X_1 \partial X_2 \partial X_3$. The "decoupling lemma" (Lemma D.1 in Appendix D of [6]) converts this problem to the computation of the sum of $\Delta(\widetilde{B}_i, \widetilde{C}_i, \widetilde{D}_i)_\sigma$ for the decomposition $\partial X_i = \widetilde{B}_i \widetilde{C}_i \widetilde{D}_i$, on the connected components of $\partial X$. These partitions, illustrated in Fig. 33(b), are natural partitions for the annuli, $\forall i = 1, 2, 3$, and $\Delta(\widetilde{B}_i, \widetilde{C}_i, \widetilde{D}_i)_\sigma = 0$. One way to see this is by noticing that $d_1 = 1$ and $2 \ln d_1 = 0$. This completes the verification of sufficient conditions for pairing.

We anticipate that the detailed explanation of the two examples above introduces the readers to the idea of the global view and the immersion view, and the fact that they contain the necessary information to verify the sufficient conditions for pairing (Lemma 5.5). The steps for verifying these conditions are similar to the examples below. For this reason, in the remaining examples, we shall omit most of the consistency checks, including the somewhat nontrivial checking of the natural partition conditions. Nonetheless, the relevant details about natural partitions will be collected and explained in a separate section §6.3. In the description of the examples below, we shall shift the focus to explaining the consequence of the pairing manifold (propositions in §5.3), because these consequences are of interest.

**Example 6.3** ($S^2 \times S^1$). The 3d closed manifold $\mathcal{M} = S^2 \times S^1$ is a pairing manifold. See Fig. 34 for an explicit pairing. Here $X$ is a sphere shell and $Y$ is a solid torus. $\mathcal{W}$ is the thickening of the immersed union $X \cup_\bullet Y$. The verification of the conditions for pairing is similar to the examples above. In particular, one can verify that $X$ and $Y$ cut each other into balls, and the transverse intersection between $X$ and $Y$ gives natural partitions of $X$ and $Y$. (See §6.3 for the detailed verification of natural partitions.)

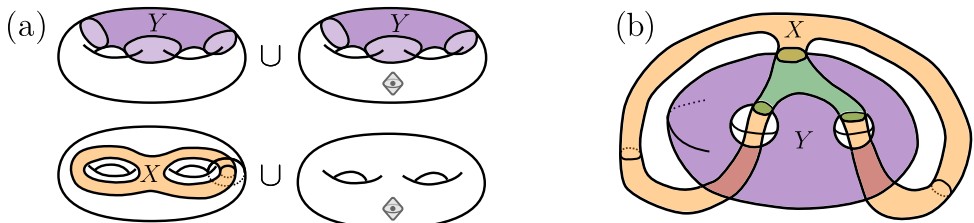

Figure 35: A pairing for the 3d manifold $\#g(S^2 \times S^1)$, where $\#g(S^2 \times S^1)$ is the connected sum of $g$ copies of $S^2 \times S^1$. (a) The global view is shown as the gluing of a pair of solid tori. (b) The immersion view. $X$ is a genus-2 handlebody, and $Y$ is a ball minus two balls. Again, $\mathcal{W}$ is the thickening of the immersed union $X \cup_\bullet Y$.

The fact that $S^2 \times S^1$ is a pairing manifold leads to a few physical consequences (propositions in §5.3). We explain them here. First, we note that $\mathcal{C}_{\text{point}} = \mathcal{C}_X$ and $\mathcal{C}_{\text{loop}} = \mathcal{C}_Y$ (this notation was introduced in [6]). By Proposition 5.10,

$$|\mathcal{C}_{\text{point}}| = |\mathcal{C}_{\text{loop}}| = \dim \mathbb{V}(S^2 \times S^1). \tag{60}$$

To see why this is true, we notice that both $X$ and $Y$ (sphere shell and solid torus) are sectorizable. Therefore, the multiplicities in (45) are either 0 or 1, so the sums reduce to the number of sectors. Remarkably, this is a derivation within the entanglement bootstrap framework that the number of the point particles and the pure flux loops must be identical in 3d. Furthermore, the number of groundstates on $S^2 \times S^1$, $\dim \mathbb{V}(S^2 \times S^1)$, equals the number of point particles (and fluxes). This agreement with the TQFT prediction provides support for the following idea: the states on $S^2 \times S^1$ which we construct from the density matrix of a ball by the torus trick has a TQFT interpretation, and the closed manifold is free from defects.

Moreover, Proposition 5.12 gives another derivation of the matching of total quantum dimension in 3d topological orders:

$$\sum_{a \in \mathcal{C}_{\text{point}}} d_a^2 = \sum_{\mu \in \mathcal{C}_{\text{flux}}} d_\mu^2, \tag{61}$$

where the left-hand side is the total quantum dimension of point excitations, and the right-hand side is the total quantum dimension of the pure flux loops. This equation is derived in [6] with a different method.

**Example 6.4** ($\#g(S^2 \times S^1)$). The 3d closed manifold $\#g(S^2 \times S^1)$ is a pairing manifold. Recall that $\#$ denotes the connected sum. See Fig. 35 for an explicit pairing, illustrated for $g = 2$. For a general $g$, $X$ is a genus-$g$ handlebody and $Y$ is a ball minus $g$ balls. The conditions for pairing can be checked explicitly. (Again, see §6.3 for the verification of the natural partitions.)

The most interesting consequence of the pairing manifold construction, in this example, is the counting of the graph excitations. According to Proposition 5.10,

$$|\mathcal{C}_g| = \sum_{a_1, \dots, a_{g+1} \in \mathcal{C}_{\text{point}}} \left( \dim \mathbb{V}_{a_1 \cdots a_g}^{a_{g+1}} \right)^2. \tag{62}$$

On the left-hand side, $\mathcal{C}_g$ denotes the list of superselection sectors of genus-$g$ graph excitations. (We note that this includes sectors where some one-cycles of the graph contain trivial excitations.) The right-hand side is the fusion space dimension of $g + 1$ point particles, which is further determined by just the data $\{\dim \mathbb{V}_{ab}^c\}$, where $a, b, c \in \mathcal{C}_{\text{point}}$. In other words, the

number of genus-$g$ graph excitations can be counted knowing just $\{\dim \mathbb{V}^c_{ab}\}$, the fusion multiplicities of the point excitations. Another consequence (via Proposition 5.12) is that the total quantum dimension of the graph excitations satisfies:

$$\sum_{\theta \in \mathcal{C}_g} d_\theta^2 = \sum_{a_1,\ldots,a_{g+1} \in \mathcal{C}_{\text{point}}} \dim \mathbb{V}^{a_{g+1}}_{a_1 \cdots a_g} d_{a_1} \cdots d_{a_g} d_{a_{g+1}} = \mathcal{D}^{2g} \,. \tag{63}$$

The second equation follows by the associativity constraints and the relation $d_a d_b = \sum_c N^c_{ab} d_c$.

**Example 6.5** (#$2(S^2 \times S^1)$ again)**.** The 3-manifold $\mathcal{M} = \#2(S^2 \times S^1)$ can be a pairing manifold in more than one way. Here we describe a pairing of $\mathcal{M}$, shown in Fig. 36, which is different from that described in Fig. 35. From the global view in Fig. 36, it can be seen that the topology of $X$ and $Y$ are identical (and so are their complements $\bar{X}$ and $\bar{Y}$ in $\mathcal{M}$). The topology of each of these regions is a solid torus with a ball removed, which is non-sectorizable. This manifold can also be described as a ball with a solid torus removed since it is $S^3$ minus a disjoint ball and solid torus.

The immersion view clearly shows how $X$ and $Y$ cut each other into balls. Their intersection is transverse, and the resulting partitions are natural. (We refer to the reader to §6.3 for the details, but point out that the decoupling idea used in Fig. 33 helps.) $\mathcal{W}$ is the thickening of the immersed union $X \cup_\bullet Y$. Its topology is $\#2(S^2 \times S^1) \setminus B^3$ (a demonstration is given in Fig. 37). Furthermore, $(\mathcal{M}, \{X, \bar{X}\}, \mathcal{W} \overset{\varphi}{\looparrowright} \mathbf{S}^3)$ gives a vacuum block completion. To see this, we observe that $X$, being an embedded region (solid torus minus a ball), is a building block, and $\bar{X}^\diamond = \mathcal{W} \setminus X$ is a solid torus minus two balls immersed in $\mathbf{S}^3$ in such a way that the torus entanglement boundary is embedded and can be filled in; moreover, one of the two spherical entanglement boundaries are embedded. This verifies that $\bar{X}^\diamond$ is a building block. This completes the verification that $(\mathcal{M}, \{X, \bar{X}\}, \mathcal{W} \overset{\varphi}{\looparrowright} \mathbf{S}^3)$ is a vacuum block completion.

It might not be obvious that this immersed region $\mathcal{W}$ in Fig. 36(b) can be smoothly deformed into the $\mathcal{W}$ considered in the other pairing shown in Fig. 35(b). Nevertheless, this is true. We illustrate the series of deformations in Fig. 37. This illustration also shows how the regions $X \subset \mathcal{W}$ deform in the process. From this information, we are able to verify that there is a non-vanishing probability of reducing $|1_X\rangle$ to the vacuum on the genus-2 handlebody contained in $\mathcal{W}$. This implies that the Abelian sector $\hat{1}_{\partial \mathcal{W}}$ must be identical for both pairings (i.e., the one considered in Fig. 36 and Fig. 35). Then, by Porism 5.13, the capacities (or "total quantum dimensions") for both pairings must match. We summarize the implications below.

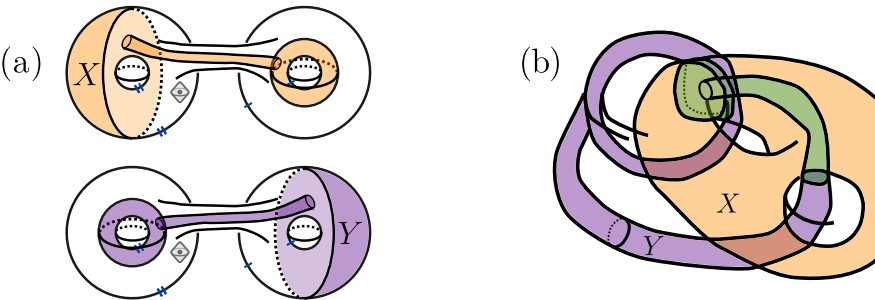

Figure 36: Another pairing for $\#2(S^2 \times S^1)$, compared with that in Fig. 35. (a) The global view. Here $\#2(S^2 \times S^1)$ is visualized as a pair of $S^2 \times S^1$ connected by a "bridge". (b) the immersion view. Both $X$ and $Y$ are homeomorphic to a solid torus minus a ball. As before, $\mathcal{W}$ is the thickening of the immersed union $X \cup_\bullet Y$.

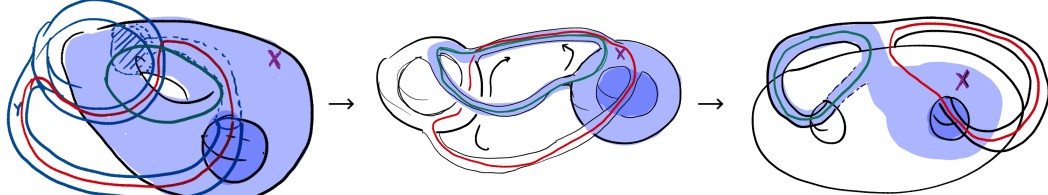

Figure 37: An explicit deformation (regular homotopy) that relates the two drawings of the immersed regions $\mathcal{W} = \#2(S^2 \times S^1) \setminus B^3$; one is that in Fig. 36(b) and another is that in Fig. 35(b). Also shown is $X$ (blue).

The matching of Hilbert space dimensions implies that

$$\sum_{a \in \mathcal{C}_{\text{point}}} \sum_{l \in \mathcal{C}_{\text{loop}}} (\dim \mathbb{V}_l^a)^2 = |\mathcal{C}_{g=2}| = \sum_{a,b,c \in \mathcal{C}_{\text{point}}} (N_{ab}^c)^2 . \tag{64}$$

Here $\mathcal{C}_{\text{loop}}$ is the set of 'shrinkable loop excitations', following the notation and terminology in [6]. $\mathcal{C}_g$ is the set of extreme points of the information convex set of the genus-$g$ handlebody.

The matching of capacities implies

$$\sum_{a \in \mathcal{C}_{\text{point}}} \sum_{l \in \mathcal{C}_{\text{loop}}} d_a d_l \dim \mathbb{V}_l^a = \mathcal{D}^4 . \tag{65}$$

In appendix D, we verify that both of these relations hold in any 3d quantum double model.

**Remark.** First, Example 6.5 is an example in $d = 3$ where neither $X$ nor $Y$ is sectorizable. Second, in Example 6.5, the extreme points of $\Sigma(X)$ can be created by operators of the form shown in Fig. 38. Interestingly, the operator is neither a string nor a membrane. Thirdly, as $\#2(S^2 \times S^1)$ has two pairings, we obtain *four* bases for its fusion space $\mathbb{V}(\#2(S^2 \times S^1))$. Below, we will recognize the unitary transformations between different bases of the fusion space of a pairing manifold as analogs of the $S$-matrix. In this example, we have *six* such matrices. Two of them are pairing matrices in the sense defined in §7; the other four are not. We leave the interpretation of the other four for future work.

## 6.2 *Pairing manifold examples with gapped boundaries

In this section, we consider a few examples of pairing manifolds with gapped boundaries. The setup corresponds to that discussed in §3.2. In this context, a connected compact manifold is a pairing manifold if it allows a pairing

$$(\mathcal{M}, \{X, \bar{X}\}, \{Y, \bar{Y}\}, \mathcal{W} \overset{\varphi}{\looparrowright} \underline{\mathbf{B}}^n) . \tag{66}$$

The 2d and 3d cases correspond to space dimensions $n = 2$ and $n = 3$. We shall find Lemma 5.5 handy in verifying the conditions required.

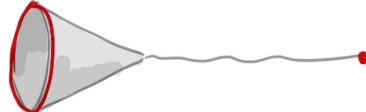

Figure 38: The form of the operator that creates a general extreme point of $\Sigma(X)$, where $X$ is a ball minus a solid torus, as appears in Example 6.5. The interesting feature is that it is neither a membrane operator nor a string operator. Rather, it is a "hybrid" type.

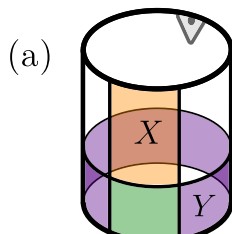
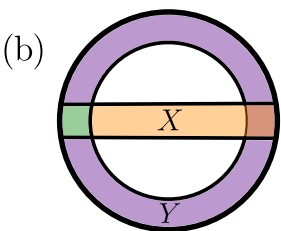

Figure 39: Cylinder with a pair of gapped boundaries as a pairing manifold. (a) The global view. (b) The immersion view. As before, the regular homotopy class of $\mathcal{W} \pitchfork \underline{\underline{\mathbf{B}}}^2$, can be inferred because $\mathcal{W}$ is the thickening of $X \cup_{\bullet} Y$.

**Example 6.6** (Cylinder with two gapped boundaries in 2d)**.** Take $\mathcal{M}$ to be a cylinder with a pair of gapped boundaries. Fig. 39 describes a way to make $\mathcal{M}$ a pairing manifold. Such an $\mathcal{M}$ should be physically interpreted as a cylinder with both of its boundaries of the same type. As the global view shows, $\mathcal{M} = X\bar{X} = Y\bar{Y}$ such that $X$ is a half-annulus, and $Y$ is an annulus that covers the entire gapped boundary. From the immersion view, we are able to see the way $X$ and $Y$ are embedded in a disk $\underline{\underline{\mathbf{B}}}^2$, and the way they make the immersed union $X \cup_{\bullet} Y$. $\mathcal{W}$ is the thickening of $X \cup_{\bullet} Y$. We are able to verify all the sufficient conditions for pairing. (The verification of the natural partition conditions will be reviewed in §6.3.)

A consequence of the fact that the cylinder with identical gapped boundaries ($S^1 \times \mathbb{I}$) is a pairing manifold is

$$|\mathcal{C}_{\text{bdy}}| = \sum_{a \in \mathcal{C}} (\dim \mathbb{V}_a(Y))^2 = \dim \mathbb{V}(S^1 \times \mathbb{I}). \tag{67}$$

Here $\mathcal{C}_{\text{bdy}}$ is the set of superselection sectors of boundary excitations. $\mathcal{C}$ is the set of anyons. $\dim \mathbb{V}_a(Y)$ are the condensation multiplicities, i.e., integers that characterize how $a$ condenses to the boundary [5, 43, 69, 70]. Another consequence is the matching of the total quantum dimension

$$\sum_{\alpha \in \mathcal{C}_{\text{bdy}}} d_\alpha^2 = \sum_{a \in \mathcal{C}} \dim \mathbb{V}_a(Y) \, d_a. \tag{68}$$

This equality is known in Ref. [5], where it is also shown that $\sum_{\alpha \in \mathcal{C}_{\text{bdy}}} d_\alpha^2 = \sqrt{\sum_{a \in \mathcal{C}} d_a^2}$.

Similarly, we can construct pairing manifolds in three dimensions with gapped boundaries. Intriguingly, not only is it possible to generate a region with more than one gapped boundary (Example 6.7), it is further possible to construct $\mathcal{M}$ with interesting boundary topology (Example 6.8).

**Example 6.7** (Sphere shell with gapped boundaries)**.** The sphere shell with a pair of gapped boundaries is a pairing manifold, and this can be seen by the construction in Fig. 40. The global view makes the following manifest: $X$ is a solid cylinder, which attaches to the gapped boundary at two disks.[37] $Y$ is a sphere shell that covers an entire gapped boundary. In other words, $Y$ has one spherical gapped boundary and one spherical entanglement boundary.

From the immersion view, we see that the two ends of $X$ attach to the same gapped boundary. Therefore, the two gapped boundaries in the global view must be "copies" of the same gapped boundary, and they must be of the same type. As before, $\mathcal{W}$ is the thickening of $X \cup_{\bullet} Y$. We are able to verify the sufficient conditions for pairing, and therefore, the construction in Fig. 40 gives a pairing of the sphere shell, viewed as a 3d manifold with a pair of spherical gapped boundaries. (See §6.3 for the details of the verification of the natural partition conditions.)

---

[37]This is a generalization of the half annulus in the context of 2d gapped boundary. It further has a generalization to the 3d domain wall, which gives an analog of the "parton sectors" described in Ref. [5].

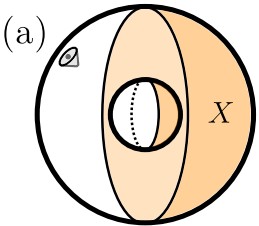 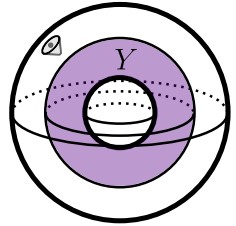 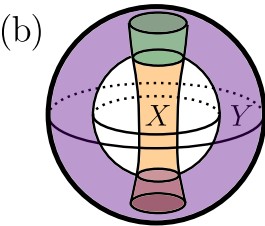

Figure 40: Sphere shell with a pair of gapped boundaries as a pairing manifold. (a) The global view. (b) The immersion view.

The physical consequence is the matching between two types of excitations and their total quantum dimensions, as follows:

$$|\mathcal{C}_{\text{bdy−flux}}| = \sum_{a\in\mathcal{C}_{\text{point}}} (\dim \mathbb{V}_a(Y))^2 . \tag{69}$$

On the left-hand side, $\mathcal{C}_{\text{bdy−flux}}$ refers to the set of superselection sectors of loop excitations lying on the boundary, which can *exist alone* on $\underline{\underline{\mathbf{B}}}^3$. They are close analogs of the pure flux loops in the 3d bulk. On the right-hand side, $\dim \mathbb{V}_a(Y)$ are the condensation multiplicities of a bulk particle $a \in \mathcal{C}_{\text{point}}$ to the gapped boundary.

Another consequence is the matching between the "total quantum dimensions":

$$\sum_{m\in\mathcal{C}_{\text{bdy−flux}}} d_m^2 = \sum_{a\in\mathcal{C}_{\text{point}}} \dim \mathbb{V}_a(Y)\, d_a . \tag{70}$$

**Example 6.8** (Solid torus with a gapped boundary)**.** The solid torus with a gapped boundary is a pairing manifold, by the construction in Fig. 41. From the global view, it can be seen that $X$ is a sphere shell cut in half by the gapped boundary and $Y$ is half of a solid torus, cut by the gapped boundary like a bagel. There is a single completion point, and it lies on the gapped boundary.

In the immersion view, everything is immersed in $\underline{\underline{\mathbf{B}}}^3$. We put a point of the spherical gapped boundary implicit at infinity. In this example, we are able to make a torus gapped boundary from pieces of a gapped boundary contained in a ball! Indeed, if we look at $X$ and $Y$ and their intersection, in the neighborhood of the gapped boundary, the topology is precisely the same as that shown in Fig. 30, the pairing for the torus $T^2$. We verified the conditions for pairing in this example.

The physical consequences are as follows. First, the matching of the Hilbert space dimension implies:

$$|\mathcal{C}_{\text{bdy−point}}| = |\mathcal{C}_{\text{bdy−arc}}| = \dim \mathbb{V}(\text{solid torus}). \tag{71}$$

This is derived by noticing that both $X$ and $Y$ are sectorizable. Here $\mathcal{C}_{\text{bdy−point}}$ is the set of superselection sectors of point excitations lying on the boundary, $\mathcal{C}_{\text{bdy−arc}}$ is the set of superselection sectors of arcs (i.e., open string) excitations ended at the boundary. Another consequence is the matching of the total quantum dimensions as follows:

$$\sum_{\alpha\in\mathcal{C}_{\text{bdy−point}}} d_\alpha^2 = \sum_{\eta\in\mathcal{C}_{\text{bdy−arc}}} d_\eta^2 . \tag{72}$$

From both Eq. (71) and (72), we see that the arc excitations must exist whenever the boundary loop excitations exist.

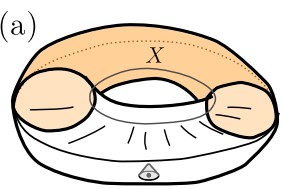
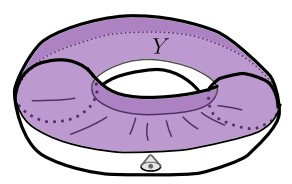
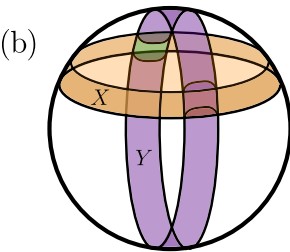

Figure 41: Solid torus with a gapped boundary in 3d as a pairing manifold. (a) The global view. (b) The immersion view. $X$ is a half-sphere shell adjacent to the gapped boundary and $Y$ is a solid torus cut in half by the gapped boundary. As usual, $\mathcal{W}$ is the thickening of the immersed union $X \cup_\bullet Y$.

Some readers may have found the analogy of the above statement with the powerful statement that in the 3d bulk: there must be nontrivial flux loops if and only if there are nontrivial point particles. (The latter is a fact explained in Example 6.3.) However, we would like to point out an important distinction. In the 3d bulk, all the fluxes (other than the vacuum) are genuine loop excitations, i.e., their support must be a loop and cannot be reduced further. The arc excitations here, however, can either be genuine arc excitations or those whose support can reduce (i.e., the arc can break). In fact, there can be models in which all the arc excitations are point excitations adjacent to the gapped boundary. Examples are topological orders attached to a trivial 3d bulk and the Walker-Wang models [71, 72]. (In both examples, $\Sigma(X)$ and $\Sigma(Y)$ are isomorphic, and they characterize the same set of excitations.) On the other hand, models with genuine arc excitations do exist, the models studied in Refs. [73, 74] are candidates.

**Remark.** An interesting generalization is genus-$g$ handlebody with a gapped boundary. This will be related to a boundary analog of the graph excitations and the counting of them.

**Example 6.9** (Riemann surfaces with domain walls). So far, we have discussed gapped boundaries; gapped boundaries are a special case of the more general notion of gapped domain walls between topological phases. The entanglement bootstrap axioms have been generalized to the context of gapped domain wall [5]. (The following examples require an extension of the notion of building blocks to this case.)

Consider a sphere whose polar caps are in topological phase $P$ and whose equatorial region is in topological phase $Q$, separated by gapped domain walls satisfying the domain wall entanglement bootstrap axioms. This is a pairing manifold where $X$ is the $n$-shaped region of [5], and $Y$ is an annulus shape that straddles the gapped domain wall. See Table 1 for an illustration.

Similarly, a torus partitioned into regions of $P$ and $Q$ separated by two gapped domain walls is a pairing manifold where $X$ and $Y$ are annuli divided in half into $P$ and $Q$ in two ways.

## 6.3 Natural partitions used in examples

In this section, we collect all the natural partitions useful in the examples in §6.1 and §6.2. The relevant partitions are summarized in Fig. 42. These are some $X = BCD$ ($Y = BCD$) associated with the transverse intersection with $Y$ ($X$) in those examples.

Before we explain why these partitions are natural partitions, we would like to mention the general strategy of showing such relations. Some of the relations directly follow from Appendix D of [6]. The useful simplification is the fact that the reference states on $\mathbf{S}^2$, $\mathbf{S}^3$, $\underline{\mathbf{B}}^2$ and $\underline{\mathbf{B}}^3$ are pure states. Let $A$ be the complement of $BCD$ on these compact manifolds. In all cases, one of the following two strategies works:



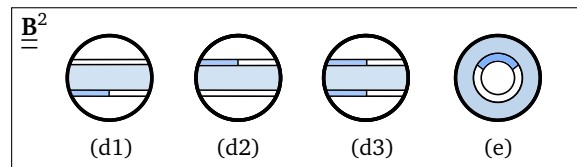

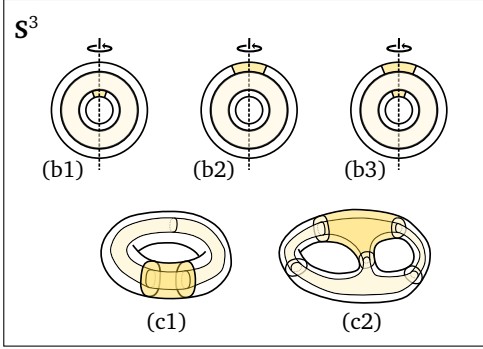

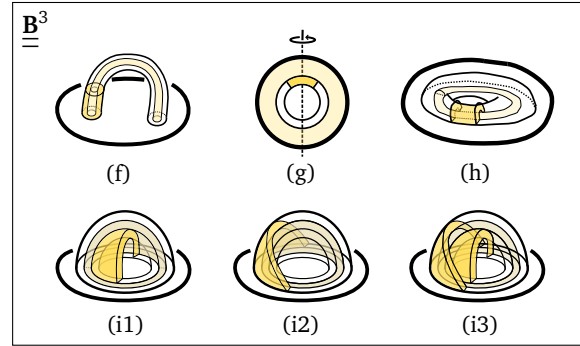

Figure 42: A collection of useful natural partitions. Each $BCD$ is embedded in $\mathbf{S}^2$, $\mathbf{S}^3$, $\underline{\underline{\mathbf{B}}}^2$ or $\underline{\underline{\mathbf{B}}}^3$. In all these cases, $C$ is the interior, and $BD$ is the thickened entanglement boundary; $C$ is light-colored, $B$ is dark-colored, and $D$ is not colored. The natural partition condition reads $\Delta(B,C,D)_\sigma = 0$.

(I) On the pure reference state $\Delta(B,C,D) = \Delta(B,A,D)$. Then we find relations between the computation of $\Delta(B,A,D)$ to the axioms on balls, and show that the value of this quantity vanishes.

(II) On the pure reference state $\Delta(B,C,D) = I(A:C|B)$. Then we use a *deformation technique* to show that $I(A:C|B) = 0$. Here the deformation technique refers to the following idea.[38] We "smoothly deform" $B$ to $AB$ by attaching balls away from $C$, in a topologically trivial way. More precisely, we construct a sequence of regions $\{A_i B\}_{i=0}^N$ such that $A_0 = \emptyset$, $A_N = A$, and

$$S_{CBA_i} - S_{BA_i} = S_{CBA_{i+1}} - S_{BA_{i+1}}, \quad \forall i = 0, \ldots, N-1. \tag{73}$$

This equation follows from strong subadditivity provided that the small ball $A_{i+i} \setminus A_i$ is attached according to the instruction above. In particular, it allows enough room to apply axiom **A1** centered at the small ball $A_{i+1} \setminus A_i$; see Fig. 43 for an illustration. Then it follows from Eq. (73) that, on the reference state

$$I(A:C|B) = I(A_N:C|B) = I(A_0:C|B) = 0. \tag{74}$$

The verification of the last equality uses the fact that $A_0 = \emptyset$. At a high level, why can this technique be useful? Although attaching these small balls cannot make topology changes to $B$ (that is, $B$ can smoothly deform into $A_i B$), the topologies of the configurations in $\{A_i\}$ need not be the same.

Below we explain why all the partitions in Fig. 42 are natural partitions for the regions in question:

- **Regions embedded in $\mathbf{S}^2$:** In Fig. 42, (a1), (a2) and (a3) are embedded in the sphere $\mathbf{S}^2$. For all of them, we apply strategy (I) and compute $\Delta(B,A,D) = 0$. For (a1) and

---

[38] This technique is known to be useful in several contexts. Notably, it is used in Lemma D.2 of [4] and in Lemma 5.7 of [6].

(a2) this boiled down to the fact that an enlarged version of **A0** and **A1** are satisfied on disks. For (a3), this reduces to the fact that the enlarged version of **A1** is satisfied on two disjoint disks. (For all three cases, we need **A0** to show that the reduced density matrix of the reference state on two disjoint disks factorizes as a tensor product.)

- **Regions embedded in $\mathbf{S}^3$:** In Fig. 42, (b1), (b2), (b3), (c1) and (c2) are embedded in the sphere $\mathbf{S}^3$. The logic for showing $\Delta(B, C, D) = 0$ for (b1), (b2) and (b3) are identical to the 2d analogs above. Namely, these follow straightforwardly from strategy (I), and therefore, we omit the details. The verification of (c1) and (c2) uses strategy (II). Namely, we use the deformation technique, which involves a sequence of deformations, starting with $B$ and ending with $AB$, and in the intermediate steps, we attach balls in locations away from $C$. This allows us to show $I(A : C|B) = 0$. Note that this strategy generalizes to genus-$g$ handlebodies. (A related observation is that for the case $X$ is a genus-$g$ handlebody, both $B$ and $AB$ are genus $g$ handlebodies.)

- **Regions embedded in $\underline{\mathbf{B}}^2$:** In Fig. 42, (d1), (d2), (d3) and (e) are embedded in the ball $\underline{\mathbf{B}}^2$, where $\underline{\mathbf{B}}^2$ is the compact disk with an entire gapped boundary. Strategy (I) is useful for solving all these cases. For cases (d1), (d2), and (d3), we need to verify $\Delta(B, A, D)_\sigma = 0$, and this is reduced to the fact that boundary versions of **A0** and **A1** are satisfied. (We also need the fact that the reference state on two disjoint $\underline{B}^2$ has zero mutual information, a property follows from the boundary version of **A0**.) To verify (e), we notice that the $\Delta(B, A, D)_\sigma = 0$ for this case is precisely the bulk version of **A1**.

- **Regions embedded in $\underline{\mathbf{B}}^3$:** In Fig. 42, (f), (g), (h), (i1), (i2) and (i3) are embedded in ball $\underline{\mathbf{B}}^3$. It is sufficient to say that the verifications for (g), (i1), (i2), and (i3) follow directly from strategy (I). The verifications for (f) and (h) follow directly from strategy (II).

Finally, we remark that the verification of natural partitions discussed in this section are partitions of an embedded region of a ball (sphere) of the following forms: some choice of $B, C, D$ such that $\Delta(B, C, D)_\sigma = 0$, or some choice of $A, B, C$ such that $I(A : C|B)_\sigma = 0$, for a reference state $\sigma$ on the ball (sphere). Pairing manifolds are further related to TEEs [64, 75], which are partitions of subsystems such that $I(A : C|B)_\sigma$ or $\Delta(B, C, D)_\sigma$ is *nonzero*. This connection was briefly discussed in a paragraph above Porism 5.13.

## 6.4 Non-examples of pairing manifolds

In this section, we discuss a few non-examples. Why non-examples? The goal is twofold. The first goal is to provide examples of $\mathcal{M}$ which are not pairing manifolds and show that some choices of regions $X$ and $Y$ cannot appear in any pairing. Besides the pedagogical motivations,

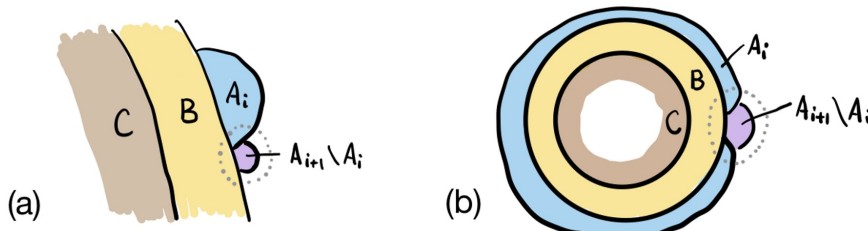

Figure 43: A schematic illustration of a step in strategy (II). $A_{i+i} \setminus A_i$ is always contained in a ball away from $C$. Note that the topology of $A_{i+1}$ and $A_i$ can either be (a) the same or (b) different.

these non-examples are also useful for attempts to generate more examples of pairing manifolds. In some examples (in §6.4.1), a topological obstruction explains why something does not work.

The second goal is to show that when some of the conditions of pairing manifolds are violated or relaxed, some of the consequences can fail. This justifies the need for various conditions stated in the definition of the pairing manifold. (Such examples are given in §6.4.2.) Furthermore, we consider the presentation of non-examples as an opportunity to show that some of the ideas and techniques we developed in the study of pairing manifolds can be generalized to other contexts. For instance, we give some interesting constructions of states on closed manifolds, even though these broader constructions do not possess all the properties of pairing manifolds.

### 6.4.1 Topology types allowing no pairing

We provide examples of connected compact manifolds $\mathcal{M}$ which cannot be pairing manifolds no matter how we attempt to partition them. In other words, there is no way to find a pairing for it.

**Non-Example 6.10** (*Manifolds with at least 3 gapped boundaries)**. Let $\mathcal{M}$ be a compact connected manifold with at least 3 gapped boundaries, in any space dimension $d \geq 2$. Then $\mathcal{M}$ does not allow any pairing, and therefore it cannot be a pairing manifold. (A corollary is that a pair of pants with three gapped boundaries does not allow any pairing; see Fig. 44 for a partition that does not work.)

To see why this is true, suppose there is a pairing such that $\mathcal{M} = X\bar{X} = Y\bar{Y}$. Because $X$ and $Y$ slice each other into two balls (so do the complements), $\mathcal{M}$ must be a union of 4 balls. Here a ball is either $B^n$ or $\underline{B}^n$. Therefore, each ball touches at most one gapped boundary, and it only covers part of it. Therefore, four such balls cannot cover three or more entire gapped boundaries. Since $\mathcal{M}$ has three or more (entire) gapped boundaries, it cannot be a pairing manifold.

The next example shows that not every pair of regions $X$ and $Y$ can be combined into a pairing manifold.

**Non-Example 6.11** ($X$ and $Y$ are not compatible)**. In this example, we ask if a given pair $X$ and $Y$ may appear in a pairing $(\mathcal{M}, \{X, \bar{X}\}, \{Y, \bar{Y}\}, \mathcal{W} \overset{\varphi}{\looparrowright} \mathbf{S}^n)$. An illustrative example is the $X$ and $Y$ shown in Fig. 45, where $X$ is an annulus and $Y$ is a 2-hole disk. It is clear that the "misguided" attempt shown in Fig. 45(b) cannot be the way to combine $X$ and $Y$. This is because $X$ did not cut $Y$ into two balls. Furthermore, the attempt to combine $X$ and $Y$ as Fig. 45(c) cannot work either. There, although $X$ and $Y$ cut each other into two balls, the intersection is not transverse. Is it true that there is no way for the $X$ and $Y$ shown in Fig. 45(a) to appear in any pairing? The answer is yes.

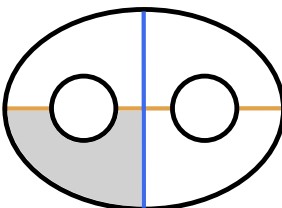

Figure 44: A pair of pants allows no pairing, as is implied by non-Example 6.10. For instance, the partition shown here does not work because the shaded region is not a disk (neither $B^2$ nor $\underline{B}^2$).

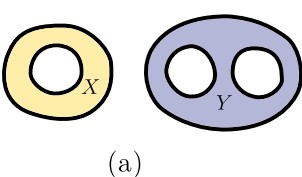
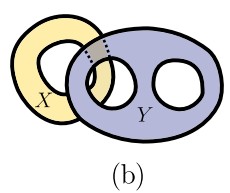
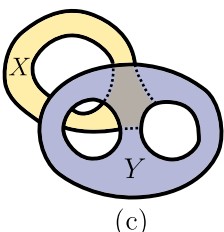

(a)         (b)         (c)

Figure 45: Example of regions $X$ and $Y$, which cannot appear in any pairing together. (a) The topology of $X$ and $Y$. (b) and (c) are two "misguided" ways to combine $X$ and $Y$.

A general strategy is to verify the failure of the consequences of pairing (§5.3). Suppose that $X$ and $Y$ participate in a pairing, then for any reference state, we must have (Proposition 5.10)

$$\sum_{a \in \mathcal{C}_{\partial X}} N_a(X)^2 = \sum_{b \in \mathcal{C}_{\partial Y}} N_b(Y)^2. \tag{75}$$

This equation generally does not hold for the $X$ and $Y$ in Fig. 45. For instance, for the reference state of the toric code, the left-hand side of Eq. (75) gives 4, whereas the right-hand side gives 16. The existence of such a reference state shows that $X$ and $Y$ cannot appear together in any pairing.

**Non-Example 6.12** ($T^3$, open question). The 3-torus can be constructed as $X\bar{X} = Y\bar{Y} = Z\bar{Z}$ where $X, \bar{X}, Y, \bar{Y}, Z, \bar{Z}$ are torus shells ($T^2 \times B^1$). If we take the $T^3$ as the 3d region identified by translations symmetry of a cubic lattice, they are standard partitions parallel to the $xy$, $yz$, and $zx$-planes. It is clear that $X$ and $Y$ (similarly, $Y$ and $Z$, $Z$ and $X$) cannot appear in any pairing together,[39] because they cut each other into solid tori, instead of balls. However, we do not know general proof that there is no pairing for $T^3$. Below is a possible obstruction. Suppose there is no way to divide $T^3$ into 4 balls ($B^3$), then one can conclude that $T^3$ does not allow any pairing.

### 6.4.2  Topologies are valid, quantum state conditions fail

Each non-example below has a set of regions and immersion maps $\mathcal{M}$, $\{X, \bar{X}\}$, $\{Y, \bar{Y}\}$ and $\mathcal{W} \overset{\varphi}{\looparrowright} \mathcal{N}$, and a quantum state on $\mathcal{M}$. The non-examples fail to fulfill at least one of the requirements of a pairing (Definition 5.3, or its boundary generalizations). In particular, the manifolds $\mathcal{M}$, $X$, and $Y$ satisfy the topological requirements of the definition, but some of the more interesting quantum state conditions fail.

**Non-Example 6.13** ($T^2$ with an anyon). We have shown that $T^2$ has a pairing (Example 6.1). Here we still consider $\mathcal{M} = T^2$, and the same immersion $\mathcal{W} \overset{\varphi}{\looparrowright} \mathbf{S}^2$ as Fig. 30(b), but consider a different state on $T^2$. We then discuss why this reference state is not what we can construct from a pairing.

Take the Ising anyon model considered in Example 4.26 with the set of anyon labels given by $\mathcal{C} = \{1, \sigma, \psi\}$. We consider the state $|\varphi^\psi\rangle$ constructed in Example 4.26. It turns out that this state reduces on $X$ and $\bar{X}^\diamond$ to extreme points. In particular, $|\varphi^\psi\rangle$ reduces to the extreme point of $\Sigma(X)$ that carries the non-Abelian anyon label $\sigma$. For the partition $X = BCD$ in Fig. 31 (which would be a natural partition if we consider the state $|1_X\rangle$), $\Delta(B, C, D) = 4 \ln d_\sigma = \ln 4$, not 0. This is true for any state in the fusion space[40] $\mathbb{V}_{\langle \psi \rangle}(T^2)$, since $\dim \mathbb{V}_{\langle \psi \rangle}(T^2) = 1$ for the

---

[39]On the other hand, it is possible to relate $T^3$ to a pairing manifold by dimensional reduction.

[40]Here, we use $\mathbb{V}_{\langle \psi \rangle}(T^2)$ to denote the Hilbert space associated with the torus $T^2$ given the reference state $|\varphi^\psi\rangle$. Note that, $\mathbb{V}_{\langle \psi \rangle}(T^2) = \mathbb{V}_\psi(T^2 \setminus B^2)$.

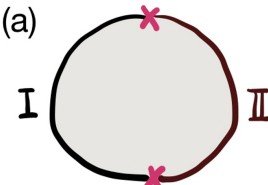
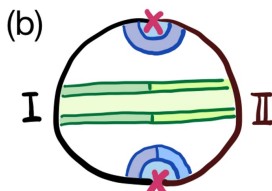

Figure 46: (a) A disk with two gapped boundary types, labeled as I and II. Boundary defects are points at the junction of the two boundaries. We shall call such a disk a "defect disk". (b) Axiom **A0** holds on the defects, whereas **A1** receives nontrivial correction at the defects. A ribbon connecting (half-annulus) connecting the two boundary types. There is no vacuum on the ribbon under the assumption that I and II are different boundary types.

chosen state $|\varphi^\psi\rangle$. So, the natural partition condition in Definition 5.3 is violated. Interestingly enough, the reference state on $T^2$ still satisfies the axioms everywhere; this is because $\psi$ is Abelian.

At the same time, the state $|\varphi^\psi\rangle$ reduces to the extreme point of $\Sigma(Y)$ labeled by the non-Abelian particle $\sigma$ on $Y$. Thus, both $X$ and $Y$ are at some extreme points. These extreme points are neither the minimum entropy states nor the maximum entropy states. This should be contrasted with the prediction in Lemma 5.9. However, the uncertainty relation, in a broader sense (not stated) still holds. Furthermore, in this example, $\sum_{a \in \mathcal{C}_{\partial X}} (\dim \mathbb{V}_a)^2 > \dim \mathbb{V}_{\langle \psi \rangle}(T^2)$, where the left-hand side is 3 and the right-hand side is 1.

In this non-example, the following relations still hold:

$$|\mathcal{C}_X| = |\mathcal{C}_Y|, \qquad \sum_{a \in \mathcal{C}_X} d_a^2 = \sum_{b \in \mathcal{C}_Y} d_b^2. \tag{76}$$

**Non-Example 6.14** ($T^2$ with a defect)**.** Here is another way to violate the pairing conditions on $T^2$. This is closely related to Example 4.27 in §4.3, which takes $\mathcal{N}$ to be an annulus through which a defect line passes, and constructs $T^2$ with some completion in $\mathcal{N}$. (This already violates the immersion in $\mathbf{S}^2$.) Consider partitions of $T^2$ into annuli, $T^2 = X\bar{X} = Y\bar{Y}$. Here $X$ and $Y$ are embedded in $\mathcal{N}$, $X$ is $\mathcal{V}_1$ of Fig. 24(a), and $Y$ is regular homotopic to $\mathcal{N}$. Thus the defect passes through $Y$. If the defect is nontrivial, $Y$ does not have any Abelian superselection sector at all. This violates the natural partition requirement.

The violation of conditions of pairing in this non-example (stronger than the previous one) leads to a violation of the consequence: $|\mathcal{C}_X| \neq |\mathcal{C}_Y|$.

**Defect disk:** The next two non-examples take a state on a disk with two types of gapped boundaries as the "reference state". See Fig. 46. We use labels I and II to label the two *different*[41] boundary types. We assume that boundary **A0** is satisfied everywhere, but boundary **A1** is violated at the points where the boundary condition changes. This is enough to guarantee $\dim \mathbb{V}(\bigcirc) = 1$.

We shall see that Example 6.15 makes more defect points, while Example 6.16 removes the defects and makes more boundaries of different types.

**Non-Example 6.15** (\*A "square" with two boundary types)**.** Let $\mathcal{M}$ be a square with 4 boundaries, where the types are I-II-I-II, as shown in Fig. 47(a). Suppose we have a defect disk with a reference state on it. Let $\mathcal{W}$ be the immersed region in the defect disk, as shown in Fig. 47(b).

---

[41]We say two boundaries are different if a ribbon connecting the two boundaries does not have a vacuum; see Fig. 46(b).

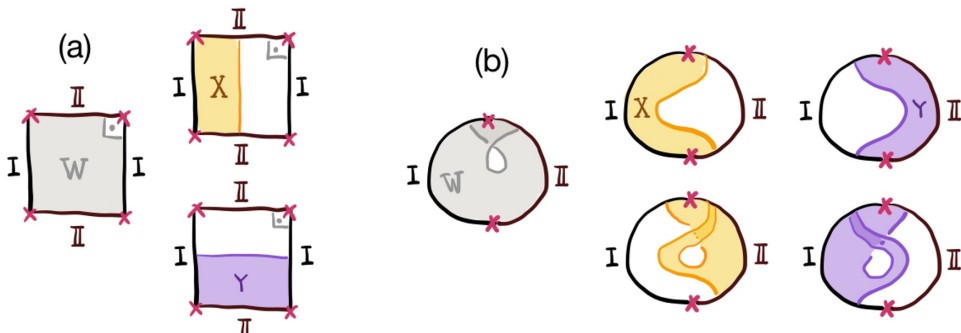

Figure 47: Make a square by immersion in a disk will two types of gapped boundaries. (a) The global view. (b) The immersion view.

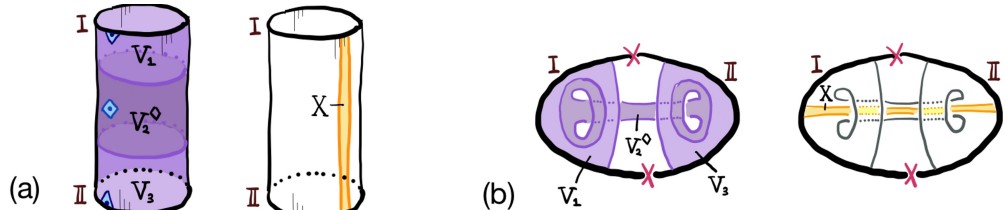

Figure 48: Starting from a "defect disk" we are able to construct a state on a cylinder with two different gapped boundary types. This state satisfies the axioms everywhere. In non-Example 6.16, we explain why this state cannot be obtained from any pairing that involves the region $X$ indicated. (a) A global view. There are three completion points. (b) The immersion view.

The observation we make here is that the manifold $\mathcal{M}$ can be "completed" with the knowledge of the defect disk.

This construction is a close analog to the pairing manifold construction: we can make $\mathcal{M}$ in two different ways, by first making $\mathcal{W}$ immersed in the defect disk, and "healing" the puncture. The two ways can be summarized by saying $\mathcal{M} = X\bar{X} = Y\bar{Y}$, which involves regions cutting each other into two balls.

However, this does not qualify as a pairing of $\mathcal{M}$. This is because there are defects lying in $X \cap Y$; they violate the boundary version of axiom **A1**, and thus the "disk" here is neither $B^2$ nor $\underline{B}^2$. Related to this, the sector on $\partial\mathcal{W}$ cannot be an Abelian sector since $\partial\mathcal{W}$ is a ribbon connecting two boundary types.

An interesting feature of this construction is that, although the original defect disk had a unique ground state, $\mathcal{M}$ can have a nontrivial multiplicity (for example, for the toric code with $I = e$ and $II = m$, there are two ground states [43]).

What consequences of pairing manifold does this non-example violate? We leave this for interested readers.

**Non-Example 6.16** ($^\star$Cylinder with two different gapped boundaries)**.** As we discussed in Example 6.6, a cylinder with two gapped boundaries allows a pairing. In that case, the two gapped boundaries are of the same type. Here, starting with a defect disk with two gapped boundary types, I and II, we construct a cylinder with two different gapped boundaries of type I and II, free from defects.

In fact, we can construct the state from building blocks. As shown in Fig. 48, there are three building blocks. There are three completion points; two are in the bulk and one is on the gapped boundary.

In what sense this is a nonexample? No state in $\mathbb{V}(\text{cylinder})$ allows a pairing with $X$ indicated in Fig. 48. The reason is that the ribbon $X$ connects two different boundary types, and therefore, it does not have a vacuum. The natural partition requirement is violated.

**Remark.** Consider the $\mathcal{M}$, $X$ and $Y$ in non-Example 6.16. From the logic we developed in deriving the consequences (§5.3) of the pairing manifold, it is not hard to verify that if there is an Abelian sector in $\Sigma(X)$, then $\Sigma(Y) \cong \Sigma(\bar{Y})$. More generally, it is possible that $\Sigma(Y) \cong \Sigma(\bar{Y})$ but $\Sigma(X)$ does not have a vacuum sector.

# 7 Pairing matrix and remote detectability

For each pairing

$$(\mathcal{M}, \{X, \bar{X}\}, \{Y, \bar{Y}\}, \mathcal{W} \overset{\varphi}{\looparrowright} \mathbf{S}^n) \quad \text{or} \quad (\mathcal{M}, \{X, \bar{X}\}, \{Y, \bar{Y}\}, \mathcal{W} \overset{\varphi}{\looparrowright} \underline{\mathbf{B}}^n), \tag{77}$$

we can uniquely define a *pairing matrix*. This is a generalization of the topological $S$ matrix in the anyon theory. However, it is a different generalization compared with the 3d $S$ matrix [22] which is associated with the mapping class group of the three-dimensional torus. In contrast, a pairing matrix often does not associate with the mapping class group of any manifold.

The pairing matrix pairs two types of excitations (or excitation clusters), those detected by $\Sigma(X)$ and those detected by $\Sigma(Y)$. We shall see that the pairing matrix, denoted as $S$, is always unitary. The interpretation of this property is that the pair of excitations remotely detect each other. The full explanation of the relation to remote detectability will appear in [38, 76], where we relate the pairing matrix to overlaps of *open* string or membrane (or other forms of) operators that create the topological excitations. These operators act within a ball and overlap nontrivially with each other. In the context of 2d gapped phases, this reduces to a braiding definition of $S$-matrix appeared in [16]. We give a preview of the remaining steps in §7.2.

As we shall explain in Definition 7.1, the most general pairing matrix is a square matrix and is unitary. Loosely speaking, it is the following matrix,

$$S_{(a,i,j)(\alpha,I,J)} = \langle a_X^{(i,j)} | \alpha_Y^{(I,J)} \rangle / [\text{unfixed gauge}], \tag{78}$$

with some requirements on "fixing the gauge". As we shall discuss, there will be three types of pairing matrices depending on if $X$ and $Y$ are sectorizable. Type-I pairing matrix has both $X$ and $Y$ sectorizable, and there is no unfixed gauge. For the other two types, the unfixed gauge exists and is of physical relevance. We will say more about this in [38].

## 7.1 Pairing matrix

Below we explain the notation in some detail to prepare for the accurate discussion that follows. The Hilbert space $\mathbb{V}_{\mathcal{M}}$ has two bases, for a given pairing (77). The first basis is

$$|a_X^{(i,j)}\rangle, \quad a \in \mathcal{C}_{\partial X}, \quad i,j = 1,\ldots,\dim\mathbb{V}_a(X). \tag{79}$$

Physically, $a$ represents the sector that appears on the sectorizable region $\partial X$. The first label $i$, in the superscript, labels a state on $\mathbb{V}_a(X)$, and the second label $j$ labels a state on $\mathbb{V}_a(\bar{X})$. Recall that $\dim\mathbb{V}_a(X) = \dim\mathbb{V}_a(\bar{X})$. There is a vacuum sector, $1 \in \mathcal{C}_{\partial X}$ for which $\dim\mathbb{V}_1(X) = 1$. This is because $X^{\diamond/\triangle}$ is a building block for which $\partial X$ is embedded in a sphere or a ball. For the vacuum sector, we often write $1 = (1,1,1)$ for short. The second basis is

$$|\alpha_Y^{(I,J)}\rangle, \quad \alpha \in \mathcal{C}_{\partial Y}, \quad I,J = 1,\ldots,\dim\mathbb{V}_\alpha(Y), \tag{80}$$

where $\alpha$ is the sector that appear on $\partial Y$, the $I$ and $J$ in the superscript labels the states on $\mathbb{V}_\alpha(Y)$ and $\mathbb{V}_\alpha(\bar{Y})$, respectively. Similarly, there is a vacuum $1 \in \mathcal{C}_{\partial Y}$ for which $\dim \mathbb{V}_1(Y) = 1$.

Because these two are orthonormal bases, it is natural to consider the matrix of inner products,

$$M_{(a,i,j),(\alpha,I,J)} = \langle a_X^{(i,j)} | \alpha_Y^{(I,J)} \rangle, \tag{81}$$

which must be a square and unitary. While the matrix of inner products can be defined for any two bases, the pairing matrix is more special.

One thing that makes the pairing matrix special is the fact that $X$ and $Y$ cut each other into balls. The uncertainty relation says that the state $|1_X\rangle$ (that is $|1_X^{(1,1)}\rangle$) reduces to a minimum entropy state on $\Sigma(X)$ and at the same time reduces to the maximum entropy state on $\Sigma(Y)$ (Lemma 5.9). This implies that there is a way to "fix the gauge" such that we can write:

$$|1_X\rangle = \sum_{\alpha \in \mathcal{C}_{\partial Y}} \sum_{I=1}^{\dim \mathbb{V}_\alpha(Y)} \frac{\sqrt{d_\alpha}}{\mathcal{D}_{\text{pair}}} |\alpha_Y^{(I,I)}\rangle, \tag{82}$$

$$|1_Y\rangle = \sum_{a \in \mathcal{C}_{\partial X}} \sum_{i=1}^{\dim \mathbb{V}_a(X)} \frac{\sqrt{d_a}}{\mathcal{D}_{\text{pair}}} |a_X^{(i,i)}\rangle, \tag{83}$$

where $\mathcal{D}_{\text{pair}} = \sqrt{\mathcal{Q}(X)} = \sqrt{\sum_{a \in \mathcal{C}_{\partial X}} \dim \mathbb{V}_a(X) d_a}$. Here $\mathcal{Q}(X)$ is the capacity of $X$. (Recall that $\mathcal{Q}(X) = \mathcal{Q}(Y)$, by Proposition 5.12. $\mathcal{D}_{\text{pair}}$ is determined by the pairing.)

Why do Eqs. (82) and (83) hold, for some gauge choice? To answer it, we recall a simple problem on a tensor product Hilbert space. Suppose $|\phi\rangle$ is a maximally entangled state in a 2-qudit system $AB$, where $\dim \mathcal{H}_A = \dim \mathcal{H}_B = K$. $\mathcal{H}_{AB} = \mathcal{H}_A \otimes \mathcal{H}_B$. Then it is possible to find a unitary matrix $U_A$ such that

$$U_A \otimes 1_B |\phi\rangle = \frac{1}{\sqrt{K}} \sum_{i=1}^{K} |i\rangle_A \otimes |i\rangle_B, \tag{84}$$

where $\{|i\rangle_A\}$ and $\{|i\rangle_B\}$ appeared in the right-hand side are two orthonormal bases. Back to the original problem. The states in the two bases (79) and (80) are not in a tensor product Hilbert space in the sense that $(i, j)$ do not associate with two $\dim \mathbb{V}_a$ dimensional Hilbert spaces in a tensor product Hilbert space. Nonetheless, we can still do a unitary rotation on the fusion space $\mathbb{V}_a(X)$ by some unitary operator supported within $X$. This explains the gauge choice (82) and (83).

Formally, a pairing matrix is defined by the following gauge choice.

**Definition 7.1** (Pairing matrix). For a given pairing (77), the pairing matrix is the unitary square matrix

$$S_{(a,i,j),(\alpha,I,J)} = \langle a_X^{(i,j)} | \alpha_Y^{(I,J)} \rangle, \tag{85}$$

written in a gauge choice of the bases $\{|a_X^{(i,j)}\rangle\}$ and $\{|\alpha_Y^{(I,J)}\rangle\}$ such that Eqs. (82) and (83) hold.

This definition implies that the "first row" and the "first column" are:

$$S_{(1,1,1)(\alpha,I,J)} = \frac{\sqrt{d_\alpha}}{\mathcal{D}_{\text{pair}}} \delta_{I,J}, \quad S_{(a,i,j)(1,1,1)} = \frac{\sqrt{d_a}}{\mathcal{D}_{\text{pair}}} \delta_{i,j}. \tag{86}$$

Does this fix the gauge completely? If not, what is the residual gauge freedom? We shall answer this question for each of the three types of pairing matrices.

**Three types of pairing matrices:** Pairing matrices fall into three types based on whether $X$ and $Y$ are sectorizable or not. (If only one of them is sectorizable, we shall use the convention

that $X$ is sectorizable.) We have summarized examples of pairings with all three types in Table 1 and §6.

**Type-I:** both $X$ and $Y$ are sectorizable, in which case the pairing matrix is a matrix with entries $S_{a\alpha}$, where $a$ runs over all sectors in $\mathcal{C}_{\partial X}$ such that $\dim \mathbb{V}_a(X) = 1$ and $\alpha$ runs over all sectors in $\mathcal{C}_{\partial Y}$ such that $\dim \mathbb{V}_\alpha(Y) = 1$. This is because, $\dim \mathbb{V}_a(X)$ and $\dim \mathbb{V}_\alpha(Y)$ are either 0 or 1.

With a suitable change of notation, the pairing matrix becomes

$$S_{a\alpha}, \quad \text{with} \quad a \in \mathcal{C}_X, \quad \alpha \in \mathcal{C}_Y. \tag{87}$$

This implies that $|\mathcal{C}_X| = |\mathcal{C}_Y|$, i.e., the numbers of superselection sectors of sectorizable regions $X$ and $Y$ match. The first row and column then becomes

$$S_{1\alpha} = \frac{d_\alpha}{\mathcal{D}_{\text{pair}}}, \quad S_{a1} = \frac{d_a}{\mathcal{D}_{\text{pair}}}. \tag{88}$$

The absence of the square root on the numerator is because of the embedding of $a \in \mathcal{C}_X$ into $\mathcal{C}_{\partial X}$ (by a partition trace on $X \setminus \partial X$), which squares the quantum dimension. In this case, there is no residual gauge symmetry, this is because all we can do is to do phase rotation on each vector $\{|a_X\rangle\}$ and $\{|\alpha_Y\rangle\}$. Eq. (88) completely determines the relative phases.

On general grounds, a type-I pairing matrix describes the overlap between two sets of minimally entangled states (MES) [54] on sectorizable regions. In the most familiar case where the pairing manifold is $T^2$, discussed in Example 6.1, these two MESs are with respect to the two cuts of $T^2$ parallel to the horizontal and the vertical direction. These MESs[42] are precisely what appeared in [54] in their prescription for extracting the $S$-matrix from the set of torus ground states (see also [55, 57, 58]). Therefore, we expect that the pairing matrix for $T^2$ is identical to the $S$ matrix of the anyon theory. Moreover, the matrix that relates the point particle and the flux loops, in 3d, considered in [23] can be understood as the type-I pairing matrix associated with Example 6.3.

**Type-II:** $X$ is sectorizable and $Y$ is not. In this case, the pairing matrix reads

$$S_{a(\alpha,I,J)}, \quad \text{with} \quad a \in \mathcal{C}_X, \quad \alpha \in \mathcal{C}_{\partial Y}. \tag{89}$$

There is an apparently-unavoidable residual gauge symmetry when $\dim \mathbb{V}_\alpha(Y) \geq 2$, which rotates both indices $I$ and $J$. (This is for the same reason that for the state in Eq. (84), any unitary on $A$ can be combined with some unitary on $B$, which leaves the state invariant.) The first row of the matrix is proportional to $\delta_{I,J}$, and so the phases of the states $|\alpha, I, J\rangle$ with $I \neq J$ cannot be fixed by this method. In particular, the half-linking matrix of [24] is an example of a type-II pairing matrix, where the possible $I, J$ comes from the condensation multiplicities. In our perspective, this is the pairing matrix associated with Example 6.6.

**Type-III:** Neither $X$ nor $Y$ is sectorizable. Here we need pairing in its most general form that appeared in Definition 7.1. While we have not been able to identify related examples in previous literature, a concrete topological context where type-III matrix appears is the partition of $(S^2 \times S^1) \# (S^2 \times S^1)$ discussed in Example 6.5.

Pairing matrices have the following curious features other than the relation to braiding non-degeneracy.

---

[42]Ref. [54] focuses on the case of abelian topological order, where all extreme points have the same entropy, which is, therefore, a global minimum; more generally, the entropy of different MES will differ and in general are only local minima of the entanglement entropy. The improved prescription in [55] addresses problems of [54] that our construction does not share: those of identifying an MES associated with the vacuum, and of prescribing an order for the bases of MES states associated with each direction of the torus. We do not have these problems because of the existence of the canonical states $|1_X\rangle$ and $|1_Y\rangle$, and because in cases where $X$ and $Y$ are the same (such as the annulus), both our bases are defined starting from the same set of extreme points.

First, for the case of the 3d quantum double model a finite group $G$, the pairing matrix for $S^2 \times S^1$ is the character table, properly normalized. The reader may believe this statement on general grounds; we will give an explicit demonstration using minimal diagrams in [38]. Other examples will be given as well in [38]. Our general demonstration of the unitarity of the pairing matrix gives an independent "proof" (not the most direct one) of the orthogonality theorem for characters of finite groups.

Second, we shall present a pair of Verlinde-like formulas for each type of pairing matrix. As we shall discuss in [38], a convenient way is to think in terms of the algebras of flexible operators associated with the excitation clusters detected by $X$ and $Y$, respectively. The special features of type-II and III pairing matrices lead to the noncommutativity of these operator algebras.

## 7.2 Relation to remote detectability

In this brief subsection, we give a sketch of the steps relating the pairing matrix $S$ to a process by which the excitations involved remotely detect each other. The full argument will appear in a sequel to this paper.

The idea is a generalization of the argument in [16] for the case of $\mathcal{M} = T^2$. The idea there is to use unitary open-string operators to create and transport anyon-anti-anyon pairs. The existence of these operators can be guaranteed by the axioms on the reference state, and the phase produced by braiding the pair of anyons around each other can be shown to participate in a Verlinde formula involving the fusion coefficients for those anyons.

The first step to generalizing this construction to more general excitation types (in general dimension, and including gapped boundaries) is the notion of *flexible operator algebra*. This is an algebra of operators acting on a region $\Omega$ with the property that when acting on the reference state, their support can be deformed topologically within $\Omega$. These are the generalizations of *closed* string operators in the case of anyons. These operators are not necessarily unitary. In [38], we will show a precise sense in which the flexible operator algebra contains information about fusion rules and braiding properties. This algebraic view is dual to the information convex set. In the special case of a reference state on a pairing manifold $\mathcal{M} = X\bar{X} = Y\bar{Y}$, we can construct two bases of flexible operators $\{W_Y^{(a,i,j)}\}$ and $\{W_X^{(\alpha,I,J)}\}$ such that the pairing matrix can be represented as

$$S_{(a,i,j)(\alpha,I,J)} = \langle 1_X | W_Y^{(a,i,j)\dagger} W_X^{(\alpha,I,J)} | 1_Y \rangle. \tag{90}$$

This expression requires operations on topologically nontrivial subsets $X, Y \subset \mathcal{M}$. (For the case $\mathcal{M} = T^2$, $X$ and $Y$ will be two non-contractible annuli.)

The final step is to cut the flexible operators in half and "fold" the resulting *open-ended operators* in a certain way [76] so that the right-hand side of Eq. (90) becomes an expectation of open-ended operators supported within a ball. Their action on the vacuum of the ball has an explicit interpretation in terms of the creation and adiabatic transport of topological excitations. Explicitly, $W^{(a,i,j)} \overset{X}{\sim} \sqrt{d_a} \left( U_L^{ai} \right)^\dagger U_R^{aj}$, where the equivalence relation $\overset{X}{\sim}$ means that the two sides agree when acting on elements of the information convex set of $X$. Using this relation, we can directly relate the pairing matrix to the phases acquired by the associated topological excitations under braiding.

## 8 Discussion

In this paper, we have effectively constructed an entanglement bootstrap reference state on closed manifolds, starting from a state on the ball. We have shown explicitly that this can be

done for arbitrary orientable 2-manifolds, and for various examples in higher dimensions. The crucial technique is to immerse (i.e., locally embed) a closed manifold with a puncture in the ball and to construct quantum states on the immersed region. It is natural to ask: for which closed manifolds does this method work?

**Differential topology questions:** The underlying technique of our construction makes it clear that the notion of *immersion*, i.e., local embedding, plays an important role. Furthermore, this allows us to borrow tools from existing topology literature, differential topology especially.

Our notion of immersion is not precisely the same as the notion from differential topology because we work with a discrete lattice instead of smooth manifolds. Nevertheless, because we work with a coarse-grained lattice that is large enough (but finite), we expect that in this regime, differential topology should effectively apply. In other words, an immersion in the sense of differential topology should imply an immersion in our sense. The finiteness of lattice in our setup rules out pathologies like the Alexander Horned Sphere, which has an infinite amount of structure in a finite region. This differentiates our setup from the "topological category" in math. So we believe that results in the smooth category should apply to our needs. It would be nice to make this more precise. As those distinctions are for a small set of exotic cases, one may expect them to be irrelevant physically. On the other hand, more careful readers may wonder if there is another mathematical structure that precisely captures our setup; a candidate is "coarse structure" [77], which provides a context to answer "what happens on the large scale".[43]

The result on immersion of differential manifolds most relevant to us is that of Hirsch and Smale (see *e.g.* [49], chapter 6) showing that a punctured $d$-manifold $\mathcal{W}$ may be immersed in the $d$-ball if and only if its tangent bundle $T\mathcal{W}$ is trivial. The only-if is clear since the immersion provides a trivialization of the tangent bundle. Since the tangent bundle of any oriented 3-manifold is trivializable, any oriented 3-manifold (minus a ball) can be immersed in the ball.

In the very recent paper [32], the case of 4-manifolds is shown to have the most favorable possible conclusion for our purposes: the *only* obstruction to immersing a punctured connected 4-manifold is the Stiefel-Whitney classes of its tangent bundle. This means that any spin 4-manifold (for which $w_1(T\mathcal{W})$ and $w_2(T\mathcal{W})$ vanish) can be immersed in the 4-dimensional ball. Another fact offered by the same reference is that any punctured connected orientable 4-manifold can be immersed in $\mathbb{CP}^2$. These results suggest what manifolds we can hope to construct by our approach in 4d, starting with a reference state on either a ball or a $\mathbb{CP}^2$.

What precisely are the secrets hidden in the ground states on manifolds with nontrivial characteristic classes, *e.g.* $\mathbb{RP}^1$ and $\mathbb{CP}^2$, (or obstruction for putting ground states on them) is not clear to us. One interesting point is that the vanishing of the characteristic classes of $T\mathcal{M}$ is *also* precisely the condition for $\mathcal{M}$ to be null-bordant, that is, for $\mathcal{M}$ to be the entire boundary of a $(d+1)$-manifold. In the Atiyah-Segal axiomatic approach to TQFT [1], a manifold $\mathcal{M}$ must be null-bordant in order to construct a vacuum on $\mathcal{M}$.

It is also worth noting that the characteristic classes of the tangent bundle of $\mathcal{M}$, are also believed to be precisely the data characterizing the response to placing an invertible phase with a finite internal symmetry group and vanishing thermal Hall response on $\mathcal{M}$ [78]. Interestingly, here $\mathcal{M}$ is a spacetime, whereas in our case $\mathcal{M}$ is a space manifold. (Relatedly, comparing to a TQFT, which produces a number for any $(d+1)$-manifold, our approach so far only allows us to study spacetime manifolds of the form $\mathcal{M}_d \times \mathcal{M}_1$.) It is an interesting question if there is any connection between the relevance of characteristic classes in the two seemingly different problems. An optimistic possibility[44] is that this is the *only* data that we miss because of this obstruction to Kirby's torus trick.

---

[43]We thank Daniel Ranard for suggesting this possibility.

[44]Suggested by Jake McNamara.

Kitaev's unpublished work on the classification of invertible phases [50, 79] involves an apparently different construction of ground states on closed manifolds from a ground state on a ball. Intriguingly, there is an identical restriction on which manifolds can be constructed; see [50] at around 1 hour and 35 minutes. In particular, the condition of a manifold having a stable normal frame implies that the characteristic classes vanish.

**Braiding nondegeneracy and beyond:** In two dimensions, the braiding matrix is called $S$ because of its relation to the eponymous generator of $\mathsf{SL}(2, \mathbb{Z})$, the modular group of $T^2$. The braiding matrices in 3d relating graphs and collections of particles that we have studied in this paper are related to connected sums of $S^2 \times S^1$ and lack such a connection. The orientation-preserving mapping class group of $S^2 \times S^1$ is generated by two order-two elements. One of them reverses the orientation of each factor. The other generalizes the Dehn twist: as we go around the circle, we do a $2\pi$ rotation of the $S^2$. We expect that the latter generator is related to the statistics of the excitations; note that, unlike the Dehn twist of $T^2$, this element is of order two, reflecting the fact that the particle excitations in 3+1d can only be bosons or fermions.

We anticipate that a full understanding of the braiding matrix for the Hopf excitations [6, 80] will use $T^3$ as the analog of the pairing manifold and that the modular transformations of $T^3$ [22, 81, 82] will play a role. However, as we mentioned in non-Example 6.12, $T^3$ seems not to be a pairing manifold by the definition in this paper: there are three apparent ways to divide $T^3 = (S^1)^3$ in half by cutting along each of the three circles; however, these regions $X, Y, Z$ intersect in pairs in solid tori rather than balls. The excitations detected by $\Sigma(X)$ are paired with some excitations detected by $\Sigma(Y)$ and some excitations detected by $\Sigma(Z)$. So there is a sort of triality rather than a duality.

The concept of pairing manifold involves two partitions $\mathcal{M} = X\bar{X} = X\bar{Y}$, where $X$ and $Y$ cut each other into balls. This is a somewhat intricate condition designed in order to prove the nontrivial statement of braiding nondegeneracy. However, it might be possible to relax some of the assumptions to cover other braiding-related phenomena. For instance, is there a sense that a "thickened Klein bottle" in 3d can be a choice of $X$ in a pairing manifold or its generalization? Note that the thickened Klein bottle can immerse (but cannot embed) in a three-dimensional ball.

Our derivation of braiding non-degeneracy generalized to systems with gapped boundaries. The pairing matrix in that setup pairs excitations on the gapped boundaries and bulk excitations that condense on the boundary. One may hope to generalize this idea to higher-dimensional defects of TQFTs in higher dimensions. There are interesting defects [83, 84] that have trivial braiding with all particle excitations and do not modify the information convex set of a linked solid torus. One example[45] of string-like defects (Cheshire charge loops) appear in $\mathbb{Z}_2 \times \mathbb{Z}_2$ gauge theory, where the loop is defined in the spacetime path integral by the insertion $W(C) = e^{\mathbf{i} \int_C a_1 \cup a_2}$. It corresponds to inserting an SPT (specifically, a cluster state) along the worldsheet of the loop. Based on the analog to the condensation to a gapped boundary (e.g., Example 6.6 and 6.7), we suspect that the solid torus that thickens the string should contain some signature.

**Universal data from a state on a ball:** The construction of states on closed manifolds from a reference state on a ball suggests that the ball may contain all the information of a topologically ordered system (or an emergent TQFT).

The construction of closed manifolds in this work relies on the understanding of the topology of immersed regions as well as the structure of information convex sets on such regions. The diverse topology comes from the fact that immersed regions are topologically more diverse than embedded regions.

---

[45]We thank Shu-Heng Shao and Jake McNamara for mentioning this example to us.

There are two other rich properties offered by immersion that deserve further study. First, immersion also provides inequivalent ways to immerse a manifold of the same topology; see, e.g., Fig. 5(b) and Example 4.25 for illustrations. (A related fact in 3d differential topology is that surfaces can immerse in multiple ways [42].) One may wonder if inequivalent immersions of a region give isomorphic information convex sets. This is an open question in 2d, even for the context illustrated precisely in Fig. 5(b). Second, the presence of immersion also implies that regions can deform through a sequence of immersed configurations. Each class of deformation that maps a region back to itself gives rise to (potentially nontrivial) automorphisms of the information convex set associated with the region. It is an interesting question what information these automorphisms characterize.

As is known from previous studies, information convex sets do not characterize all the data of a gapped phase by itself. One object we do not expect them to characterize in 2+1d is the chiral central charge, which is, nonetheless, characterized in a different way [20] by the density matrix on a ball. (See Ref. [21] for a related proposal for systems with $U(1)$ symmetry.) An open question is how to extract the higher dimensional analogs of chiral central charge (which exists in $4n + 2$ space dimensions based on TQFT arguments) and higher dimensional analogs of Hall response (which exists in even space dimensions). Even for invertible phases, which necessarily have a trivial information convex set, there are more questions to be answered. For instance, there is an invertible phase in 4+1d, which has $\mathbb{Z}_2$ classification [78, 85–89]. A non-additive characterization may detect this classification.

The perspective offered above, namely that an arbitrary 3-manifold (with entanglement boundaries) has a role to play in the entanglement bootstrap, begs the question of whether this huge multiplicity of data is all independent or whether it is determined from some finite subset. In two dimensions, the associativity theorem implies that the data of the annulus and two-hole disk suffice to determine the fusion multiplicities for any other region. In three dimensions, we do not yet know an analogous statement. There are multiple ways to partition a 3-manifold. One may ask if the prime decomposition of the general 3-manifold allows for the existence of such minimal data. While we do not know the answer to this question, we notice that the "sequential completion" discussed in §4.3 provides a way to obtain connected sums of 3-manifolds (as with the boundary analog). Three manifolds can also be split into two handlebodies, a procedure known as Heegaard splitting. A third method is to obtain 3-manifolds by gluing along torus boundaries, leading to the JSJ decomposition. It is an interesting question if we can learn anything from the consistency between different ways of obtaining a 3-manifold.

# Acknowledgments

We are grateful to Anuj Apte, Dan Freed, Tarun Grover, Peter Huston, Isaac Kim, Michael Levin, Xiang Li, Ho Tat Lam, Ting-Chun Lin, Daniel Ranard, Jake McNamara, Julio Parra-Martinez, David Penneys, Shu-Heng Shao and Hao-Yu Sun for helpful discussions and comments, to Lauren Miyashiro for the bundt cake picture, and to [90] for teaching us the word 'porism'.

**Funding information** This work was supported in part by funds provided by the U.S. Department of Energy (D.O.E.) under the cooperative research agreement DE-SC0009919, by the University of California Laboratory Fees Research Program, grant LFR-20-653926, and by the Simons collaboration on Ultra-Quantum Matter, which is a grant from the Simons Foundation (652264, JM).

# A Notation glossary

| notation | meaning | first appears in |
|---|---|---|
| $S^d$ | a $d$-dimensional sphere | §1 |
| $\mathbf{S}^d$ | a $d$-dimensional sphere with a reference state | §1 |
| $B^d$ | a $d$-dimensional ball | §1 |
| $\mathbf{B}^d$ | a $d$-dimensional ball with a reference state | §1 |
| $\underline{B}^d$ | a $d$-dimensional half-ball adjacent to a gapped boundary | |
| $\underline{\mathbf{B}}^d$ | a $d$-dimensional half-ball adjacent to a gapped boundary with a reference state | |
| $\underline{\underline{\mathbf{B}}}^d$ | a $d$-dimensional ball with an entire gapped boundary with a reference state | |
| $\mathcal{B}_g$ | a ball minus $g$ balls | §1 |
| $A\#B$ | connected sum of manifolds $A$ and $B$ | §1.1 |
| entanglement boundary | a boundary of a region that is not part of the gapped boundaries | §2 |
| $\partial\Omega$ | thickened entanglement boundary of the region $\Omega$ | §2.1 |
| $\mathrm{ext}(\Sigma(\Omega))$ | the set of extreme points of the convex set $\Sigma(\Omega)$ | Def. 2.1 |
| $\Sigma_{[\kappa_A]}(\Omega)$ | convex subset of $\Sigma(\Omega)$ of density matrices that restrict to $\rho_A^\kappa$ on $A$ | Def. 2.1 |
| $\Sigma_{I[\kappa_A]}(\Omega)$ | convex subset of $\Sigma_I(\Omega)$ of density matrices that restrict to $\rho_A^\kappa$ on $A$ | Def. 2.1 |
| $\mathbb{V}_{I[\kappa_A]}(\Omega)$ | constrained fusion space | Def. 2.1 |
| $X \overset{\mathcal{N}}{\sim} X'$ | regions $X$ and $X'$ can be smoothly deformed into each other as immersed regions of $\mathcal{N}$ | §3 |
| $E(h)$ | the centralizer subgroup of an element $h\in G$ | Prop.3.2 |
| $\mathcal{W} \looparrowright \mathcal{X}$ | immersion of $\mathcal{W}$ into $\mathcal{X}$ | §4 |
| compact manifold | a manifold with no entanglement boundaries | §4 |
| closed manifold | a compact manifold with no gapped boundaries | §4 |
| $\#gT^2$ | genus-$g$ Riemann surface | Fig. 25 |
| $\mathcal{Q}(X)$ | $e^{S_X\vert_{\rho^{\min}}^{\rho^{\max}}}$ where $\rho^{\max}$ is the maximum-entropy state in $\Sigma(X)$ and $\rho^{\min}$ is any minimum-entropy state in $\Sigma(X)$ | Def. 5.11 |

| notation | meaning | first appears in |
|---|---|---|
| $S_\Omega\vert_{\rho_2}^{\rho_1}$ | $S(\rho_\Omega^1)-S(\rho_\Omega^2)$ | (54) |
| $A\cup_{\bullet} B$ | the immersed union of two regions $A$ and $B$ | (57) |
| $G_g$ | genus-$g$ handlebody | §C |
| $E(C)$ | the centralizer of a representative of the conjugacy class $C$ | §C.2 |

# B Consistency relations with quantum dimensions

In entanglement bootstrap, two kinds of consistency relations are often derived and then used; they are summarized in Table 2. The first kind is derived from merging *whole components* of entanglement boundaries. The consistency relations derived this way are associativity of fusion multiplicities, analogous of $N_{abc}^d = \sum_i N_{ab}^i N_{ic}^d$. Consistency relations of the second kind involve both fusion multiplicities and quantum dimensions, and they are derived by merging *part of* the entanglement boundaries of the given regions; see Fig. 49 for an illustration. One example is

Table 2: Two kinds of merging processes and the consistency relations they derive.

|  | merging | consistency relation |
|---|---|---|
| 1st kind | entire entanglement boundaries | with only fusion multiplicities $N^c_{ab}$, $N^c_{[ab]}$ |
| 2nd kind | parts of entanglement boundaries | with fusion multiplicities $N^c_{ab}$, $N^c_{[ab]}$ and quantum dimensions $d_a$ |

$d_a d_b = \sum_c N^c_{ab} d_c$. While the first kind may look more familiar to TQFT audiences, consistency relations of the 2nd kind are arguably more fundamental in entanglement bootstrap: starting from the axioms **A0** and **A1**, we need the second kind of merging to derive the isomorphism theorem, the Hilbert space theorem, before we can talk about fusion spaces.

In [6], a general Associativity Theorem was derived, with which we read off the desired associativity relations without repeating the proof in each context. This simplifies the task of finding consistency relations of the first kind.

The goal of this appendix is to provide a few convenient results for the second kind of consistency relation: the kind that requires merging part of the entanglement boundaries. As indicated in Table 2, the fusion multiplicities include those from constrained fusion spaces (Definition 2.1). We start with the general setup (Definition B.1).

**Definition B.1** (merging setup, as illustrated in Fig. 49). Consider a region $ABCD$ embedded in a ball.

1. $ABC$ and $BCD$ are connected sectorizable regions of the form $\mathcal{M} \times \mathbb{I}$, where $\mathcal{M}$ is a connected manifold.

2. $\partial(ABC) = \Omega_1 \cup \Omega_2$, where $\Omega_1$ and $\Omega_2$ are separated, $\Omega_2 \subset A$ may be empty

3. $\partial(BCD) = \Omega_3 \cup \Omega_4$, where $\Omega_3$ and $\Omega_4$ are separated, $\Omega_4 \subset D$ may be empty

4. $\partial(ABCD) = \Omega_2 \cup \Omega_4 \cup \Omega_5$, where $\Omega_5$ is the newly formed entanglement boundary, obtained by merging part of $\Omega_1$ and $\Omega_3$. ($\Omega_2$, $\Omega_4$ and $\Omega_5$ are separated from each other.)

5. We assume that $I(A : C|B)_{\rho^a} = 0$, for any $a \in \mathcal{C}_{ABC}$ and $I(B : D|C)_{\rho^d} = 0$ for any $d \in \mathcal{C}_{BCD}$. In addition, we assume that, in our setup, $\rho^a_{ABC}$ and $\rho^d_{BCD}$ can be merged in a way required by the merging theorem as long as they match on $BC$, that is $\mathrm{Tr}_A \rho^a_{ABC} = \mathrm{Tr}_D \rho^d_{BCD}$.

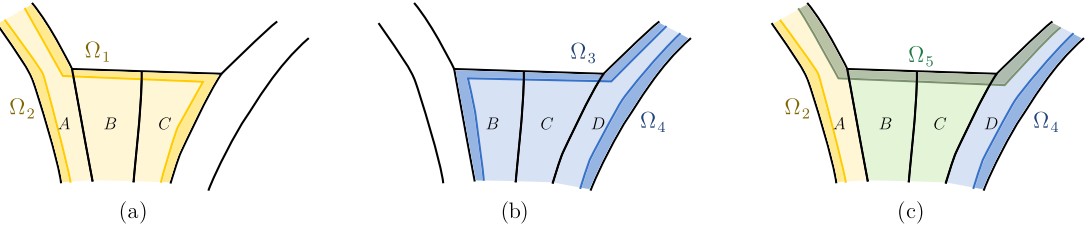

Figure 49: Schematic depiction of the merging setup of this appendix (Definition B.1). (a) The thickened entanglement boundaries of $ABC$ are $\Omega_1$ and $\Omega_2$, where $\Omega_2$ may be empty. (b) The thickened entanglement boundaries of $BCD$ are $\Omega_3$ and $\Omega_4$, where $\Omega_4$ may be empty. (c) After merging, $\Omega_5$ appears as a newly formed entanglement boundary.

This reduced density matrix on $BC$, $(\mathrm{Tr}_A \rho^a_{ABC})$ must then be an extreme point of $\Sigma(BC)$ because $ABC$ is sectorizable (Proposition 2.23 in [6]).

6. Let $a \in \mathcal{C}_{ABC}$, $d \in \mathcal{C}_{BCD}$, $b \in \mathcal{C}_{\partial(BC)}$ and $e \in \mathcal{C}_{\Omega_5}$. We say $a, d, b$ ($a$ and $d$) are *compatible* if $\Sigma_{[adb]}(ABCD)$ ($\Sigma_{[ad]}(ABCD)$) is nonempty. Here and below, we use $[abd]$ ($[ad]$) as the short-hand notation of $[a_{ABC} b_{\partial BC} d_{BCD}]$ ($[a_{ABC} d_{BCD}]$). The quantum dimensions $d_a, d_b, d_d, d_e$ are defined from the entropy difference Eq. (7), noting that the reference state ($\sigma$) exists on the respective sectorizable regions. Let $N^e_{[ad]}$ be the dimension of the constrained fusion space associated with $\Sigma^e_{[ad]}(ABCD)$. (See Lemma 2.2 to recall the notations.)

7. We define $\widehat{d}_a$ and $\widehat{d}_d$ as

$$
\widehat{d}_a = \begin{cases} d_a^2, & \Omega_2 = \emptyset, \\ d_a, & \Omega_2 \neq \emptyset, \end{cases} \quad \text{and} \quad \widehat{d}_d = \begin{cases} d_d^2, & \Omega_4 = \emptyset, \\ d_d, & \Omega_4 \neq \emptyset. \end{cases} \tag{B.1}
$$

They allow simple physical interpretations. $\widehat{d}_a$ and $\widehat{d}_d$ are the quantum dimensions of the superselection sectors on regions $\Omega_1$ and $\Omega_3$, obtained by reducing $\rho^a_{ABC}$ and $\rho^d_{BCD}$ respectively.

**Proposition B.2.** *Under the setup in Definition B.1, when $a, b, d$ are compatible*

$$
\frac{\widehat{d}_a \widehat{d}_d}{d_b} = e^{-I(A:D|BC)_\sigma} \sum_e N^e_{[ad]} d_e, \tag{B.2}
$$

*and moreover, the probability of finding $e \in \mathcal{C}_{\Omega_5}$ in the merged state of $\rho^a_{ABC}$ and $\rho^d_{BCD}$ is*

$$
P_{ad \to e} = \frac{N^e_{[ad]} d_e}{\sum_e N^e_{[ad]} d_e}. \tag{B.3}
$$

Most of the consistency relations in §6.2, 6.3, and 6.4 of [6] are special cases of Proposition B.2. In particular, the consistency relations about knot multiplicities in §6.3 and 6.4 therein requires a non-vacuum state on $BC$. The consistency relations in §6.4 of [6] (for the torus minus a torus knot) also follow from this theorem. An example including a nonvanishing conditional mutual information factor $I(A:D|BC)_\sigma$ is Eq. (6.4) of [6]. More broadly, Proposition B.2 can be used to derive analogous consistency relations for torus links!

In §C.1 we will apply Proposition B.2 to more examples, including the genus-two handlebody. In §C.2, we explicitly verify these consistency conditions for the case of the $S_3$ quantum double model in 3d.

It is possible that all connected sectorizable regions are of the form $\mathcal{M} \times \mathbb{I}$, but we do not know. In the latter case, it is interesting to generalize Proposition B.2. Furthermore, there can be an analogous statement without the assumption that $ABC$ or $BCD$ are sectorizable.

To prepare for the proof of Proposition B.2 we present one lemma B.3 and its corollary.

**Lemma B.3.** *Under the setup in Definition B.1, and suppose $a \in \mathcal{C}_{ABC}$ and $d \in \mathcal{C}_{BCD}$ are compatible. If $e \in \mathcal{C}_{\Omega_5}$ and $\rho^{ade}_{ABCD} \in \mathrm{ext}(\Sigma^e_{[ad]}(ABCD))$, then*

$$
S_{ABCD}|_\sigma^{\rho^{ade}} = \ln\left(\frac{d_a^2}{\widehat{d}_a}\right) + \ln\left(\frac{d_d^2}{\widehat{d}_d}\right) + \ln d_e. \tag{B.4}
$$

*In particular, every extreme point of $\Sigma^e_{[ad]}(ABCD)$ has the same von Neumann entropy.*

*Proof.* Both $\sigma_{ABCD}$ and $\rho_{ABCD}^{ade}$ are extreme points of $\Sigma(ABCD)$. Therefore, the contributions of entropy difference $S_{ABCD}|_{\sigma}^{\rho^{ade}}$ come from the three thickened entanglement boundaries, $\Omega_2$, $\Omega_4$ and $\Omega_5$:

$$S_{ABCD}|_{\sigma}^{\rho^{ade}} = \frac{1}{2}(S_{\Omega_2} + S_{\Omega_4} + S_{\Omega_5})|_{\sigma}^{\rho^{ade}}$$

$$= \ln\left(\frac{d_a^2}{\widehat{d}_a}\right) + \ln\left(\frac{d_d^2}{\widehat{d}_d}\right) + \ln d_e . \tag{B.5}$$

The second line follows from the physical meanings of $\widehat{d}_a$ and $\widehat{d}_d$. Explicitly,

$$\frac{1}{2}S_{\Omega_2}|_{\sigma}^{\rho^a} = S_{ABC}|_{\sigma}^{\rho^a} - \frac{1}{2}S_{\Omega_2}|_{\sigma}^{\rho^a}$$

$$= \ln d_a^2 - \ln\widehat{d}_a . \tag{B.6}$$

A similar derivation shows $\frac{1}{2}S_{\Omega_4}|_{\sigma}^{\rho^d} = \ln d_d^2 - \ln\widehat{d}_d$. $\qquad\square$

Lemma B.3 implies that the state with maximal entropy in $\Sigma_{[ad]}^e(ABCD)$ is a convex combination of mutually orthogonal extreme points in $\Sigma_{[ad]}^e(ABCD)$, as:

$$\rho_{ABCD}^{(ade)_{\max}} = \frac{1}{N_{[ad]}^e} \sum_{i=1}^{N_{[ad]}^e} \rho_{ABCD}^{(ade)_i} , \tag{B.7}$$

where $\{\rho_{ABCD}^{(ade)_i}\}_{i=1}^{N_{ad}^e}$ are mutually orthogonal extreme points of $\Sigma_{[ad]}^e(ABCD)$ associated with an orthonormal basis of $\mathbb{V}_{[ad]}^e(ABCD)$.[46] The state $\rho_{ABCD}^{(ade)_{\max}}$ has entropy

$$S(\rho_{ABCD}^{(ade)_{\max}}) = S(\sigma_{ABCD}) + \ln\left(N_{[ad]}^e \frac{d_a^2 d_d^2}{\widehat{d}_a\widehat{d}_d} d_e\right) , \tag{B.8}$$

where $\sigma_{ABCD}$ is the reference state. Now we are ready to prove Proposition B.2.

*Proof of Proposition B.2.* Suppose $a \in \mathcal{C}_{ABC}$ and $d \in \mathcal{C}_{BCD}$ are compatible. Merge $\rho_{ABC}^a$ with $\rho_{BCD}^d$ to obtain the merged state $\tau_{ABCD}^{ad}$.

We are going to compute $S_{ABCD}|_{\sigma}^{\tau^{ad}} \equiv S(\tau_{ABCD}^{ad}) - S(\sigma_{ABCD})$ in two different ways. On one hand, $S_{ABCD}|_{\sigma}^{\tau^{ad}}$ can be calculated from the entropies of the density matrices of the subregions, knowing $I(A:D|BC)_{\sigma}$. On the other hand, the merged state $\tau_{ABCD}^{ad}$ is the state with maximal entropy in $\Sigma_{[ad]}(ABCD)$ and we use the structure theorem of $\Sigma_{[ad]}(ABCD)$ to express the entropy difference.

1. By definition of conditional mutual information,

$$S_{ABCD}|_{\sigma}^{\tau^{ad}} = (S_{ABC} + S_{BCD} - S_{BC} - I(A:D|BC))|_{\sigma}^{\tau^{ad}} . \tag{B.9}$$

By the merging lemma [4], $I(A:D|BC)_{\tau^{ad}} = 0$ for an arbitrary pair of compatible $a$ and $d$. Since $BC$ is embedded in $ABC$, from Proposition 2.23 in [6], $\mathrm{Tr}_A \rho_{ABC}^a$ is an extreme point of $\Sigma(BC)$ (and also $\Sigma(\partial(BC))$.) Let $b \in \mathcal{C}_{\partial(BC)}$ be the sector label associated with this state. Using the definition of quantum dimension $d_I \equiv \exp(\frac{S(\rho^I)-S(\sigma)}{2})$ we get:

$$S_{ABCD}|_{\sigma}^{\tau^{ad}} = \ln\frac{d_a^2 d_d^2}{d_b} + I(A:D|BC)_{\sigma} . \tag{B.10}$$

The reason we have $d_b$ instead of $d_b^2$ in the denominator is $S_{\partial(BC)}|_{\sigma}^{\tau^{ad}} = 2S_{BC}|_{\sigma}^{\tau^{ad}}$.

---

[46]Note that the fusion space $\mathbb{V}_{[ad]}^e(ABCD)$ is well-defined because $a, d, e$ fully determine the superselection sector at $\partial(ABCD)$.

2. On the other hand, $\tau_{ABCD}^{ad}$ is the maximum-entropy state of $\Sigma_{[ad]}(ABCD)$, following the "standard" maximization procedure [4, 91] of entanglement bootstrap, we have

$$
\begin{aligned}
S_{ABCD}|_{\sigma}^{\tau^{ad}} &= \max_{\{P_{ad\to e}\}} S\left(\sum_e P_{ad\to e}\, \rho_{ABCD}^{(ade)_{\max}}\right) - S(\sigma_{ABCD}) \\
&= \max_{\{P_{ad\to e}\}} \sum_e P_{ad\to e}\left(S_{ABCD}|_{\sigma}^{\rho^{(ade)_{\max}}} - \ln P_{ad\to e}\right) \\
&= \max_{\{P_{ad\to e}\}} \sum_e P_{ad\to e}\left(\ln\left(N_{[ad]}^e \frac{d_a^2}{\widehat{d}_a}\frac{d_d^2}{\widehat{d}_d}d_e\right) - \ln P_{ad\to e}\right) \\
&= \ln\left(\sum_e N_{[ad]}^e \frac{d_a^2}{\widehat{d}_a}\frac{d_d^2}{\widehat{d}_d}d_e\right) - \max_{\{P_{ad\to e}\}} D(Q_{ad\to e}||P_{ad\to e}) \\
&= \ln\left(\sum_e N_{[ad]}^e \frac{d_a^2}{\widehat{d}_a}\frac{d_d^2}{\widehat{d}_d}d_e\right).
\end{aligned}
\tag{B.11}
$$

Here $\{P_{ad\to e}\}$ is a probability distribution, i.e., $P_{ad\to e} \geq 0$ and $\sum_e P_{ad\to e} = 1$. The second equality is due to the orthogonality of different extreme points in $\Sigma_{[ad]}^e(ABCD)$. The third line follows from Eq. (B.8). In the fourth line, $Q_{ad\to e} = \frac{N_{[ad]}^e d_e}{\sum_e N_{[ad]}^e d_e}$ (which is a probability distribution), and

$$
D(\{p_i\}||\{q_i\}) \equiv \sum_i p_i \ln(p_i/q_i),
\tag{B.12}
$$

is the classical relative entropy with two probability distributions $\{p_i\}$ and $\{q_i\}$. The last line is obtained by the fact that $D(\{p_i\}||\{q_i\}) \geq 0$ for any $\{p_i\}$ and $\{q_i\}$, where "=" holds if and only if $\{p_i\} = \{q_i\}$. This last step also implies that the optimal probability distribution $\{P_{ad\to e}\}$ is

$$
P_{ad\to e} = \frac{N_{[ad]}^e d_e}{\sum_e N_{[ad]}^e d_e}.
\tag{B.13}
$$

An alternative way to find Eq. (B.13) uses the Lagrange multiplier. To get the maximum of $S_{ABCD}|_{\sigma}^{\tau^{ad}}$ with the constraint that $\sum_e P_{ad\to e} = 1$, we let

$$
h(P_{ad\to e}) \equiv \sum_e P_{ad\to e}\left(\ln\left(N_{[ad]}^e \frac{d_a^2}{\widehat{d}_a}\frac{d_d^2}{\widehat{d}_d}d_e\right) - \ln P_{ad\to e}\right) + \lambda\left(\sum_e P_{ad\to e} - 1\right),
$$

be a function of $P_{ad\to e}$. When this function is at maximum, its first derivative is zero. That is,

$$
\frac{\partial h}{\partial P_{ad\to e}} = \ln(N_{[ad]}^e \frac{d_a^2}{\widehat{d}_a}\frac{d_d^2}{\widehat{d}_d}d_e) - \ln P_{ad\to e} - 1 + \lambda = 0.
\tag{B.14}
$$

This equation gives $P_{ad\to e} \propto N_{[ad]}^e d_e$. After normalizing, we arrive at Eq. (B.13).

Comparing Eq. (B.10) and Eq. (B.11), we obtain Eq. (B.2). $\qquad\square$

# C  Information convex set for handlebodies

A genus-$g$ handlebody, which we denote as $G_g$, is a boundary connected sum of solid tori; its entanglement boundary is a genus-$g$ Riemann surface. As we explained in §3.1.2, its information convex set detects excitations supported on *graphs* living in the complement of the handlebody, $S^3 \setminus G_g$. We referred to these excitations as *graph excitations*.

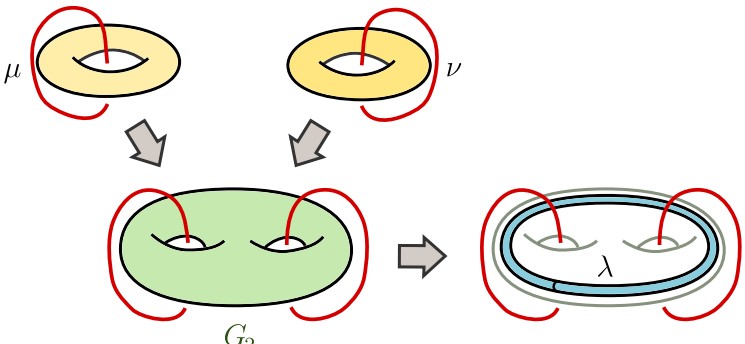

Figure 50: An illustration fusion of two flux loops, viewed in the information convex sets. The three solid tori and the genus-2 handlebody $G_2$ are the regions we consider information convex sets. Flux excitations are the red loops. (This view is complementary to Fig. 9 in the main text.)

In this appendix, we discuss some properties of graph excitations. In §C.1, we write down general relations that arise by thinking of a graph as resulting from the fusion of loops. In §C.2, we count graph excitation types for 3-dimensional quantum double models. We also discuss a different fusion process of loops called Borromean fusion, which was considered earlier in [6].

## C.1 Relations between quantum dimensions and fusion probabilities of graph excitations

Graph excitations can be created by the fusion of flux loops; see Fig. 9 for the case of genus two. If we label the pure flux excitations of the two constituent loops as $\mu, \nu \in \mathcal{C}_{\text{flux}}$, as indicated in Fig. 50, we can ask about the probability $P_{(\mu \times \nu \to \lambda)}$ of obtaining the sector $\lambda \in \mathcal{C}_{\text{flux}}$ near the "outer boundary" of the genus-2 handlebody.

As we shall discuss, $P_{(\mu \times \nu \to \lambda)}$ can be computed using the information convex set of genus-2 handlebody. (However, unlike the fusion probabilities of point particles, it is unknown to us if $P_{(\mu \times \nu \to \lambda)}$ are determined by a set of integers.) Let the genus-2 handlebody be $G_2$. Since $G_2$ is a sectorizable region, $\Sigma(G_2)$ is a simplex. We can label the set of superselection sectors as

$$(\mu, \nu, \lambda)_k \in \mathcal{C}_{G_2}, \tag{C.1}$$

where $\mu, \nu$ and $\lambda$ are the sectors we can detect after doing a partial trace to reduce $G_2$ to the three solid tori indicated in Fig. 50. $k$ labels a discrete set of extra degrees of freedom, if necessary. (An example of its necessity is given at the end of Appendix C.) The following facts follow:

1. $(1, 1, 1)_1 = 1$.

2. Among $(\mu, 1, \lambda)_k$, there is only one possible choice, denoted as $(\mu, 1, \mu)$.

3. Among $(1, \nu, \lambda)_k$, there is only one possible choice, denoted as $(1, \nu, \nu)$.

4. Among $(\mu, \nu, 1)_k$, there is only one possible choice, denoted as $(\mu, \bar{\mu}, 1)$. The derivation is shown in Fig. 51.

We can define the quantum dimension of an extreme point of $\Sigma(G_2)$ as

$$d_{(\mu, \nu, \lambda)_k} \equiv \exp\left( \frac{S\left(\rho_{G_2}^{(\mu, \nu, \lambda)_k}\right) - S\left(\sigma_{G_2}\right)}{2} \right). \tag{C.2}$$

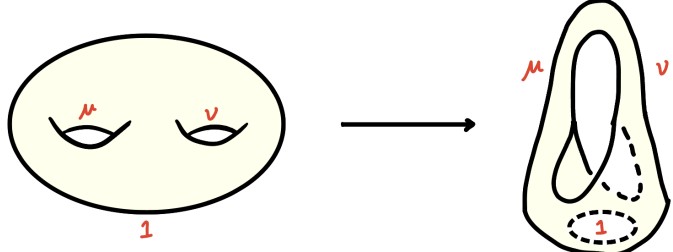

Figure 51: The explanation of $\nu = \bar{\mu}$ for $(\mu, \nu, 1)_k$. After a smooth deformation of a genus-2 handlebody, we fill in the hole labeled by 1 with a disk (by merging). The result is a solid torus. This relates $\mu$ and $\nu$ as $\nu = \bar{\mu}$.

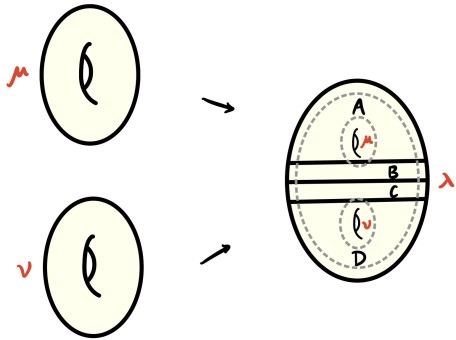

Figure 52: Merging two solid tori into a genus-2 handlebody. $\mu, \nu \in \mathcal{C}_{\text{flux}}$ label the sectors for the two small inner solid tori. $\lambda \in \mathcal{C}_{\text{flux}}$ labels the sector for the "outermost" solid torus, indicated in the figure.

The quantum dimensions satisfy the following:

$$d_1 = 1, \quad d_{(\mu, \bar{\mu}, 1)} = d_{(\mu, 1, \mu)} = d_{(1, \mu, \mu)} = d_\mu. \tag{C.3}$$

It is possible to choose a convention of labeling $k$ such that

$$d_{(\mu, \nu, \lambda)_k} = d_{(\nu, \mu, \lambda)_k} = d_{(\bar{\nu}, \bar{\mu}, \bar{\lambda})_k}. \tag{C.4}$$

We shall stick to this convention.

**Proposition C.1.** *In the general genus-2 handlebody setup, we have the matching of quantum dimensions:*

$$d_\mu^2 d_\nu^2 = \sum_{\lambda, k} d_{(\mu, \nu, \lambda)_k}^2, \quad \forall \mu, \nu \in \mathcal{C}_{\text{flux}}. \tag{C.5}$$

*The probability distribution $\{P_{(\mu \times \nu \to \lambda)}\}$ satisfies:*

$$P_{(\mu \times \nu \to \lambda)} = \frac{\sum_k d_{(\mu, \nu, \lambda)_k}^2}{d_\mu^2 d_\nu^2}. \tag{C.6}$$

**Corollary C.1.1.** *A few special elements of $\{P_{(\mu \times \nu \to \lambda)}\}$ are:*

$$P_{(\mu \times 1 \to \nu)} = \delta_{\mu, \nu}, \quad P_{(\mu \times \nu \to \lambda)} = P_{(\nu \times \mu \to \lambda)}, \quad P_{(\mu \times \nu \to 1)} = \frac{1}{d_\mu^2} \delta_{\nu, \bar{\mu}}. \tag{C.7}$$

*Proof.* Consider the merging process in Fig. 52: $\rho^{\mu}_{ABC} \bowtie \rho^{\nu}_{BCD} = \tau^{\mu\nu}_{ABCD}$. Note that this merging setup is one example of the general setup in Definition B.1. The relevant data in the setup are substituted by

$$a \to \mu, \quad d \to \nu, \quad \widehat{d}_a \to d^2_\mu, \quad \widehat{d}_d \to d^2_\nu, \quad d_b \to 1, \quad d_e \to d^2_{(\mu,\nu,\lambda)_k},$$

$$I(A:D|BC)_\sigma \to 0, \quad N^e_{[ad]} \to N^{(\mu,\nu,\lambda)_k}_{[\mu\nu]} = \{0,1\},$$

$$P_{ad\to e} \to P_{(\mu\nu\to(\mu,\nu,\lambda)_k)}, \quad \sum_e \to \sum_{\substack{\lambda,k \text{ s.t.} \\ (\mu,\nu,\lambda)_k \text{ exists}}} . \tag{C.8}$$

Plugging these in Eq. (B.2) and Eq. (B.3) we get:

$$d^2_\mu d^2_\nu = \sum_{\lambda,k} d^2_{(\mu,\nu,\lambda)_k}, \tag{C.9}$$

and

$$P_{(\mu\nu\to(\mu,\nu,\lambda)_k)} = \frac{d^2_{(\mu,\nu,\lambda)_k}}{\sum_{\lambda,k} d^2_{(\mu,\nu,\lambda)_k}} = \frac{d^2_{(\mu,\nu,\lambda)_k}}{d^2_\mu d^2_\nu}. \tag{C.10}$$

Note that $P_{(\mu\times\nu\to\lambda)}$ is different from $P_{(\mu\nu\to(\mu,\nu,\lambda)_k)}$, because $\lambda$ is not a sector label of $\partial(G_2)$. To compute the former, we to do a sum of $k$, as:

$$P_{(\mu\times\nu\to\lambda)} = \sum_k P_{(\mu\nu\to(\mu,\nu,\lambda)_k)} = \sum_k \frac{d^2_{(\mu,\nu,\lambda)_k}}{d^2_\mu d^2_\nu}. \tag{C.11}$$

This completes the proof. $\square$

## C.2 Quantum double examples

The graph excitations are not just pure fluxes stuck together. This is illustrated by the case of the quantum double, where the label set on extreme points of the genus-two handlebody $\{C_{g,h}\}$ can be larger than $\mathcal{C}^2_{\text{flux}}$.

The minimal diagram for genus-$g$ handlebody (see Fig. 53) requires only $g$ boundary links in a bouquet (meaning that they all begin and end at the unique boundary vertex $v$). Note that these links are in one-to-one correspondence with generators of the fundamental group with base point $v$. There are no faces required, so these group elements are all independent, and therefore

$$C_{(g_1,\dots,g_n)} \equiv \{(g_1,\dots,g_n)\}/\sim, \tag{C.12}$$

where the equivalence relation is $(g_1,\dots,g_n) \simeq (tg_1\bar{t},\dots,tg_n\bar{t})$. In the case of genus-two, the extreme point labelled by $(g,h)$ can be written as

$$\rho_{(g,h)} = \frac{1}{d^2}\left(|g,h\rangle\langle g,h| + |tg\bar{t},th\bar{t}\rangle\langle tg\bar{t},th\bar{t}| + ((d^2-2) \text{ more terms})\right). \tag{C.13}$$

**Example C.2** (Information convex set of genus-two handlebody for $S_3$ quantum double)**.** The list of genus-two graph excitations and their quantum dimensions are given in Table 3.

We now give the proof of the formula for the number of genus-$g$ graph excitations for the quantum double model with any gauge group, which we restate here:

Table 3: Representative $(g, h)$ for each extreme point of $\Sigma(G_2)$ for the $S_3$ quantum double, and the associated (squared) quantum dimension. This number is the number of representatives of the equivalence class. For example, the density matrix on the minimal diagram for the sector $(r, r)$ can be written $\rho_{(r,r)} = \frac{1}{2}|r, r\rangle\langle r, r| + \frac{1}{2}|r^2, r^2\rangle\langle r^2, r^2|$. The fact that there are two extreme points that have the labels $(\mu, \nu) = (C_r, C_r)$ on the two one-cycles of the handlebody (namely $(g, h) = (r, r)$ and $(g, h) = (r, r^2)$) is an example of the need for the label $\lambda$ in Eq. (C.1). Similarly, there are two sectors labelled $(C_s, C_s)$, namely $(s, s)$ and $(s, sr)$.

| $(g, h)$ | $(1,1)$ | $(1,r)$ | $(r,1)$ | $(s,1)$ | $(1,s)$ | $(r,r)$ | $(r,r^2)$ | $(r,s)$ | $(s,r)$ | $(s,s)$ | $(s,sr)$ |
|---|---|---|---|---|---|---|---|---|---|---|---|
| $d^2_{(g,h)}$ | 1 | 2 | 2 | 3 | 3 | 2 | 2 | 6 | 6 | 3 | 6 |

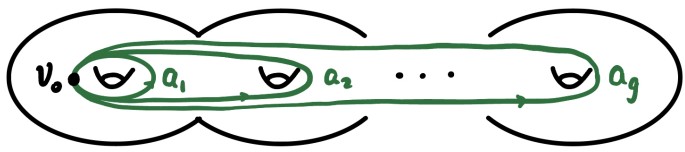

Figure 53: The minimal diagram for genus-$g$ handlebody.

**Proposition 3.2** (Graph sectors for 3d quantum double). *For 3d quantum double with finite group G, the set $(\mathcal{C}_g)$ of superselection sectors of graph excitations characterized by the information convex set of genus-g handlebody has*

$$|\mathcal{C}_g| = \frac{1}{|G|} \sum_{h \in G} |E(h)|^g \,, \tag{14}$$

*elements, where $E(h)$ is the centralizer group of $h : E(h) \equiv \{k \in G | kh = hk\}$. $|G|$ denotes the order of finite group G. In particular when $G = S_3$, $|\mathcal{C}_g| = 6^{g-1} + 3^{g-1} + 2^{g-1}$.*

To prove Proposition 3.2, firstly we need a simple lemma in group theory.

**Lemma C.3** (Burnside's lemma, or Cauchy–Frobenius lemma). *Let G be a finite group that acts on a set X. For each $h \in G$, denote $X^h$ as the set of elements in X that are invariant under h, i.e. $X^h \equiv \{x \in X | h \cdot x = x\}$. The orbit of element $x \in X$ is the set of elements in X that x can be moved by elements of G. Then the number of orbits in X under the action of G is equal to*

$$\frac{1}{|G|} \sum_{h \in G} |X^h| \,. \tag{C.14}$$

*Proof.* The proof can be found in textbooks on abstract algebra, for example [92]. □

*Proof of Proposition 3.2.* The minimal diagram [6, 45] for genus-$g$ handlebody consists of 1 vertex $v_0$, $g$ links $\{a_1, a_2, \ldots, a_g\}$ and zero plaquette.

The vertex term acts on the links by

$$A^h_{v_0} |a_1, \ldots, a_g\rangle = |ha_1 h^{-1}, \ldots, ha_g h^{-1}\rangle \,. \tag{C.15}$$

Therefore the number of elements in $\mathcal{C}_g$ is equal to the number of orbits of set $\{a_1, \ldots, a_g\}$ under the action of $G$ (Here the group action of $G$ is left multiplication by $h \in G$ and right multiplication by $h^{-1} \in G$). The invariant set $\{a_1, \ldots, a_g\}^h$ for $h \in G$ is a set of group elements that commute with $h$, i.e. $\{a_1, \ldots, a_g\}^h = \{a_1, \ldots, a_g | a_1 h = ha_1, \ldots, a_g h = ha_g\}$. Hence, its size is $|E(h)|^g$.

From Lemma C.3, the number of orbits of $\{a_1, \ldots, a_g\}$ under $G$ action is

$$\frac{1}{|G|} \sum_{h \in G} |E(h)|^g. \tag{C.16}$$

$\square$

Proposition 5.10 (in particular Eq. (45)) implies that the number of extreme points of the information convex set of the genus-$g$ handlebody is related to the fusion dimensions of the ball minus $g$ balls by the following relation:

$$|\mathcal{C}_g| = \sum_{a_1 \cdots a_{g+1}} \left( N^1_{a_1 \cdots a_{g+1}} \right)^2. \tag{C.17}$$

For the case of the 3d quantum double model, Prop. 3.2 then implies

**Proposition C.4.**

$$\frac{1}{|G|} \sum_{h \in G} |E(h)|^g = \sum_{a_1 \cdots a_{g+1}} \left( N^1_{a_1 \cdots a_{g+1}} \right)^2. \tag{C.18}$$

We can give a direct proof of this relation using character orthogonality.

*Proof.* Since $N^1_{a_1 \cdots a_{g+1}}$ is real, we can write

$$\sum_{a_1 \cdots a_{g+1}} \left( N^1_{a_1 \cdots a_{g+1}} \right)^2 = \sum_{a_1 \cdots a_{g+1}} \left| \frac{1}{|G|} \sum_{g \in G} \chi_{a_1}(g) \cdots \chi_{a_{g+1}}(g) \right|^2$$
$$= \frac{1}{|G|^2} \sum_{g,g'} \prod_{i=1}^{g+1} \left( \sum_{a_i} \chi_{a_i}(g) \chi^\star_{a_i}(g') \right). \tag{C.19}$$

Character orthogonality says

$$\sum_a \chi_a(C) \chi_a(C')^\star = \delta_{CC'} \frac{|G|}{n_C}, \tag{C.20}$$

where $C$ and $C'$ are conjugacy classes of $G$, and $n_C$ is the number of elements in the conjugacy class $C$. This does not completely fix $g$ and $g'$:

$$\sum_{a_i} \chi_{a_i}(g) \chi^\star_{a_i}(g') = \delta_{g' \in C_g} \frac{|G|}{n_g}. \tag{C.21}$$

So

$$\sum_{a_1 \cdots a_{g+1}} \left( N^1_{a_1 \cdots a_{g+1}} \right)^2 = \frac{1}{|G|^2} \sum_{g g'} \delta_{g' \in C_g} \left( \frac{|G|}{n_g} \right)^{g+1}$$
$$= \frac{1}{|G|^2} \sum_g n_g \left( \frac{|G|}{n_g} \right)^{g+1} = \frac{1}{|G|^2} \sum_C n_C^2 \left( \frac{|G|}{n_C} \right)^{g+1} \tag{C.22}$$
$$= \sum_C \left( \frac{|G|}{n_C} \right)^{g-1} = \sum_C |E(C)|^{g-1}.$$

$\square$

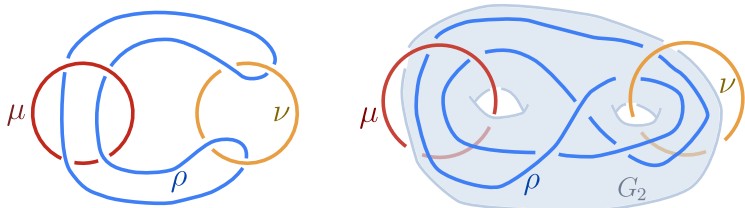

Figure 54: Left: Borromean rings of flux. Right: The Borromean fusion process can be described via the merging of two solid tori into a genus-two handlebody, $G_2$ (in light blue). The outcome of the process is measured by the solid torus labelled $\rho$, which is contained in $G_2$. In terms of the two edges of the minimal diagram for the genus-two solid (with holonomies $g$ and $h$), the curve labeled $\rho$ has holonomy $ghg^{-1}h^{-1} = [g,h]$.

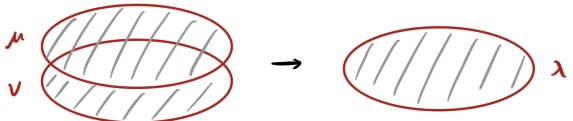

Figure 55: Fusion of two fluxes on top of each other. $\mu, \nu \in \mathcal{C}_{\text{flux}}$, and the result $\lambda$ is another flux. Each excitation is created by a membrane operator supported on a disk.

**Graphs from fusion of loops.** Here we discuss two fusion processes of pure fluxes. One is the fusion process depicted in Fig. 55, and the other is the Borromean fusion defined in [6]. In each case, we can extract the probabilities for each possible outcome by the following method. Consider the merging of two solid tori in the respective extreme points $\mu$ and $\nu$ as in Fig. 52. The resulting state on $G_2$ is the maximum entropy state consistent with the marginals $\mu$ and $\nu$ on $ABC$ and $BCD$. From the definition of the quantum dimension in terms of the entropy, this state has the form

$$\rho^\star_{\mu\nu} = \sum_{C_{(g,h)}, C_g = \mu, C_h = \nu} \frac{d^2_{(g,h)}}{\mathcal{D}^2} \rho_{(g,h)} \, . \tag{C.23}$$

Define $\lambda$ as the state of the solid torus indicated by the dashed outer curve in Fig. 52. In terms of the minimal diagram, the associated group element is just $gh$. Define $\rho$ as the state of another solid torus, the one indicated in Fig. 54. In terms of the minimal diagram, the associated group element is the group commutator $ghg^{-1}h^{-1} \equiv [g,h]$. In each case, the value of $\lambda$ and $\rho$ is specified by the representative $(g,h)$. The probability of a given fusion outcome can be read off from the values of $\lambda$ and $\rho$ in each of the summands in (C.23) and their respective quantum dimensions.

**Example C.5** (Two kinds of fusion of pure fluxes for $S_3$ quantum double). In Table 4, we show the outcomes of these two fusion processes for the case with $G = S_3$.

From this table, we can verify the relation Eq. (C.5). In the sectors involving a 1 on one of the two cycles, the relation holds trivially. In the sector $(C_r, C_r)$, the label $k$ takes two values and the RHS of (C.5) is $2 + 2 = 4 = d_r^4$. In the sector $(C_r, C_s)$, there is a unique sector whose dimension is $d_r^2 d_s^2$. In the sector $(C_s, C_s)$ there are two sectors and the RHS gives $3 + 6 = d_s^4$.

More generally, it is not a mystery that equation (C.5) is satisfied in any quantum double model: the $|\mu||\nu| = d_\mu^2 d_\nu^2$ states labelled $|(g,h)\rangle\langle(g,h)|$ with $g \in \mu$ and $h \in \nu$ form a collection of orbits under the equivalence relation in Eq. (C.12). Each orbit is labeled with some $\lambda$ specified by $\lambda = C_{gh}$. (For given $\mu, \nu$, if there is more than one orbit with the same $\lambda$, then

Table 4: Here, we specify the states $(\mu, \nu)$ of the subsystems along the two cycles of the genus two solid, $G_2$. Then we construct the maximum entropy state on $G_2$ consistent with this data. In that maximum entropy state, we can ask about the state on the various solid tori embedded in $G_2$ depicted in Figs. 52 and 54.

| $(\mu, \nu)$ | $C_{(g,h)}$ | $\lambda = C_{gh}$ | $\rho = C_{[g,h]}$ | $d^2_{(g,h)}$ | $p$ |
|---|---|---|---|---|---|
| $(C_1, C_1)$ | $C_{(1,1)}$ | $C_1$ | $C_1$ | 1 | 1 |
| $(C_r, C_1)$ | $C_{(r,1)}$ | $C_r$ | $C_1$ | 2 | 1 |
| $(C_s, C_1)$ | $C_{(s,1)}$ | $C_s$ | $C_1$ | 3 | 1 |
| $(C_r, C_r)$ | $C_{(r,r)}$ | $C_r$ | $C_1$ | 2 | 1/2 |
| $(C_r, C_r)$ | $C_{(r,r^2)}$ | $C_1$ | $C_1$ | 2 | 1/2 |
| $(C_r, C_s)$ | $C_{(r,s)}$ | $C_{sr}$ | $C_s$ | 6 | 1 |
| $(C_s, C_s)$ | $C_{(s,s)}$ | $C_1$ | $C_1$ | 3 | 1/3 |
| $(C_s, C_s)$ | $C_{(s,sr)}$ | $C_r$ | $C_r$ | 6 | 2/3 |

the label $k$ in (C.1) is required.) In any case, the number of elements in the orbit labelled $(\mu, \nu, \lambda)_k$ is $d^2_{(\mu,\nu,\lambda)_k}$, and therefore

$$\sum_{k,\lambda} d^2_{(\mu,\nu,\lambda)_k} = d^2_\mu d^2_\nu. \tag{C.24}$$

We should give an example of the need for the label $k$ in (C.1). In a quantum double model with gauge group $G$, this arises when there exist $g, h \in G$ such that $C_{gh} = C_{gth\bar{t}}$ (so that $\mu, \nu, \lambda$ are the same for $(g, h)$ and for $(g, th\bar{t})$, but $C_{(g,h)} \neq C_{(g,th\bar{t})}$, so they have different orbits. An example where this occurs is $G = A_5$, the group of even permutations on five elements, with $(g, h) = ((123), (123))$, in cycle notation.

# D  Quantum double illustration of pairing manifold relations

As a consequence of the fact that $\#2(S^2 \times S^1)$ is a pairing manifold in two ways, we found a number of relations between genus-two graph excitations and shrinkable loops in §6.1. Here, for illustration purposes, we verify some of these relations in a class of examples.

For the example of the quantum double model with $G = S_3$, we can directly calculate the LHS of (64) from Table 7 of [6]. Since the entries are all 1 or 0, the LHS is just the number of ones in the table:

$$\sum_{\ell \in \mathcal{C}_{\text{loop}}} \sum_{a \in \mathcal{C}_{\text{point}}} (N_\ell^a)^2 \overset{S_3}{=} 11. \tag{D.1}$$

This agrees with the number of genus-two graph excitations for this model, verifying (64).

More generally, we can verify (64) for an arbitrary quantum double model with gauge group $G$, using (C.13) of [6] and its complex conjugate. Repeated use of character orthogo-

nality gives

$$
\begin{aligned}
\sum_{\ell \in \mathcal{C}_{\mathrm{loop}}} \sum_{a \in \mathcal{C}_{\mathrm{point}}} (N_\ell^a)^2 &= \sum_{\ell=(C_b,R_\ell)} \frac{1}{|E_b|^2} \sum_{a \in \mathcal{C}_{\mathrm{point}}} \sum_{g,g' \in E_b} \chi_{R_\ell}(g) \chi_{R_a}^\star(g) \chi_{R_\ell}(g')^\star \chi_{R_a}(g') \\
&= \sum_{C_b} \frac{1}{|E_b|^2} \sum_{g,g' \in E_b} \underbrace{\sum_{R_\ell \in (E_b)_{\mathrm{ir}}} \chi_{R_\ell}(g) \chi_{R_\ell}(g')^\star}_{=\sum_{h \in E_b} \delta_{g',kg\bar{k}}} \underbrace{\sum_{a \in (G)_{\mathrm{ir}}} \chi_{R_a}^\star(g) \chi_{R_a}(g')} \\
&= \sum_{C_b} \frac{1}{|E_b|} \sum_{g \in E_b} \underbrace{\sum_{a \in (G)_{\mathrm{ir}}} \chi_{R_a}^\star(g) \chi_{R_a}(g)}_{=|E_g|} \\
&= \sum_{C_b} \frac{1}{|E_b|} \sum_{g \in E_b} |E_g| \\
&= \mathcal{C}_{\mathrm{genus}\ 2} \,.
\end{aligned}
\tag{D.2}
$$

To see the last relation, note that for any $f(b,g)$,

$$
\begin{aligned}
\sum_{b \in G} \sum_{g \in E_b} f(b,g) &= \sum_{b \in G} \sum_{g \in G} \delta_{g \in E_b} f(b,g) \\
&= \sum_{b \in G} \sum_{g \in G} \delta_{b \in E_g} f(b,g) \\
&= \sum_{g \in G} \sum_{g \in E_g} f(b,g),
\end{aligned}
\tag{D.3}
$$

since $g \in E_b \Leftrightarrow b \in E_g \Leftrightarrow gb = bg$. Therefore

$$
\begin{aligned}
\sum_{C_b} \frac{1}{|E_b|} \sum_{g \in E_b} |E_g| &= \frac{1}{|G|} \sum_{b \in G} \sum_{g \in E_b} |E_g| \\
&= \frac{1}{|G|} \sum_{g \in G} \sum_{b \in E_g} |E_g| = \frac{1}{|G|} \sum_{g \in G} |E_g|^2 \\
&= \frac{1}{|G|} \sum_{C_g} n(C_g) |E_g|^2 = \sum_{C_g} |E_g| = \mathcal{C}_2 \,.
\end{aligned}
\tag{D.4}
$$

Similarly, we can verify the relation for the capacities, (65), for an arbitrary quantum double model. The ingredients are:

$$
d_a = \dim R_a = \chi_a(1), \quad d_{\ell=(C_b,R_\ell)} = n(C_b) \dim R_\ell = n(C_b) \chi_{R_\ell}(1),
\tag{D.5}
$$

and (C.13) of [6] again. This gives

$$
\begin{aligned}
\sum_{\ell \in \mathcal{C}_{\mathrm{loop}}} \sum_{a \in \mathcal{C}_{\mathrm{point}}} d_a d_\ell N_\ell^a &= \sum_{\ell=(C_b,R_\ell)} \frac{1}{|E_b|} \sum_{a \in \mathcal{C}_{\mathrm{point}}} \sum_{g \in E_b} \chi_{R_\ell}(g) \chi_{R_a}^\star(g) \chi_{R_\ell}(1)^\star \chi_{R_a}(1) n(C_b) \\
&= \sum_{\ell=(C_b,R_\ell)} \frac{n(C_b)}{|E_b|} \sum_{g \in E_b} \chi_{R_\ell}(g) \chi_{R_\ell}(1)^\star \underbrace{\sum_{a \in G_{\mathrm{ir}}} \chi_{R_a}^\star(g) \chi_{R_a}(1)}_{=|G|\delta_{g,1}} \\
&= |G| \sum_{C_b} \frac{n(C_b)}{|E_b|} \underbrace{\sum_{R_\ell \in (E_b)_{\mathrm{ir}}} \chi_{R_\ell}(1) \chi_{R_\ell}(1)^\star}_{=\sum_{R_\ell \in (E_b)_{\mathrm{ir}}} \dim(R_\ell)^2 = |E_b|} \\
&= \sum_{C_b} n(C_b)^2 = \mathcal{D}^4 \,.
\end{aligned}
\tag{D.6}
$$

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
