# Peer review of "Remote detectability from entanglement bootstrap I: Kirby's torus trick"

_SciPost Physics, doi:SciPost Phys. 18, 126 (2025)_

## Round 1 · Referee Report · Anonymous (Referee 1) · 2024-1-5

Strengths

1- The paper gives an entanglement-based understanding of the idea of remote detectability of excitations in quantum phases exhibiting topological order, in 2 and 3 spatial dimensions.

2- The paper provides a systematic construction of quantum states on space manifolds with nontrivial topologies starting from a reference state on a topologically trivial space. The case of 2 spatial dimensions is dealt with conclusively while a lot of progress is made towards the case of 3 spatial dimensions. This lends concrete evidence to the longstanding lore in topological order literature that says that all topological data (including e.g. modular S,T matrices that encode anyon braiding in 2+1d) should be recoverable from a single ground state wave function, or density matrix, defined on a sphere.

3- The paper introduces the notion of "pairing matrices" which are closely related to modular S and T matrices of 2+1d topological orders. In 3+1d, these are morally similar to the mapping class group representations of 3-manifolds discussed in the literature, although there are key differences which are highlighted in the paper.

4- Figures are thoughtfully included throughout the paper, which greatly aid in the understanding of many of the topological manipulations.

Weaknesses

1- The authors have made clear efforts to make their paper largely self-contained. However, in some parts, the text is still not very welcoming to readers who are not already familiar with their past work on entanglement bootstrap. Some examples are identified in the Report below, along with some questions that may aid in clarifying some conceptual jumps.

2- A significant chunk of the paper [section 3.2 (pp 30-34), definition 4.6 (pg 44), example 4.8 (pg 46), sections 4.2.3-4.2.4 (pp 51-54), section 6.2 (pp 85-89), non-examples 6.10 (pg 93) and 6.16 (pp 97-98)] discusses systems with gapped boundaries. However, it seems that almost all of the discussion in these sections are straightforward generalizations of the case with no gapped boundaries. The overall presentation would be clearer if these cases could instead be put in an appendix that is devoted to all issues associated with gapped boundaries.

3- Even though the authors argue that the unitarity of the pairing matrix implies remote detectability, this connection is not made clear. In particular, they write in section 7 (pp 98-99), "[...] the pairing matrix, denoted as $S$, is always unitary, which shows that the pair of excitations remotely detect each other". It is not clear in this description what remote process achieves such detactability. Further clarification on this point is crucial to provide clarity to one of the most important results (as the authors themselves note) obtained in the paper.

Report

The primary technique developed in this paper is a recipe to construct ground states on topologically nontrivial spatial manifolds (in 2 and 3 dimensions) starting from a single reference state on a topologically trivial region, either a ball or a sphere, by making use of immersions (i.e. local embeddings). The paper also discusses the case where the system has gapped boundaries in which case the reference state is defined on a ball with a gapped boundary. They provide a recipe for stitching together different "blocks" that makes use of various theorems proven earlier by (some or all of) the authors. A special class of topologically nontrivial manifolds, "pairing manifolds", are presented. Two sets of basis states can be constructed on each pairing manifold, and (roughly) the overlap between these two bases defines the "pairing matrix" for the topological order on that manifold. This matrix is unitary, which the authors argue indicates remote detectability of topological excitations that are dual to certain "building blocks" that construct those manifolds.

The paper builds on the authors' program of entanglement bootstrap, which is of great conceptual value, as noted in Strength #2. However, as noted in Weakness #1, there are some conceptual gaps at various places that we would like to request clarification on:
1.1) On pp 18-19, the authors discuss fusion multiplicity and fusion spaces. Could they describe what these are? These concepts are not explained anywhere prior to this in the paper and they make many appearances throughout the rest of the paper.
1.2.a) In the proof of Lemma 2.2 (statement 1), the authors state "The compactness follows from that of $\Sigma(\Omega)$." Is this because these subsets of $\Sigma(\Omega)$ are closed? If so, it's not obvious why that is the case. Could the authors please clarify?
1.2.b) In the proof of Lemma 2.2 (statement 2), it is not obvious why "The extreme point label $\kappa$ on $A$ induces a label $\Phi(\kappa)$ of the extreme points of $\Sigma(\partial A)$ " and why "any element of $\Sigma_{I[\kappa_A]}(\Omega)$ is obtained by merging some element of $\Sigma_{I \Phi(\kappa)}( \tilde E)$ with the same state $\rho^\kappa_A$": could the authors please clarify these points further?
1.3) The authors refer to "topological defects" in footnote 23 (page 36) as well as under Example 4.27. Could they clarify what exactly these defects are? Are they symmetry defects, or string/membrane operators, or something else entirely?
1.4) On page 45, the last sentence of the first bullet point states "The remaining boundary must also carry the vacuum, according to the structure theorem of the information convex set of embedded k-hole disks."-- does the "structure theorem" refer to one of the two structure theorems described in section 2 (pp 18-19)?
1.5) In the proof of Proposition 4.10, how does one see that the $\hat \sigma_{\mathcal V_i}$ states "have the quantum Markov state structure"? Also, why is the "merged state on $\mathcal W$ [is] an extreme point" of $\Sigma _{\hat 1}(\Omega)$?
1.6) In the proof of lemma 5.7, the the reasoning behind the statements "This implies that $\rho_Y^{|Ψ〉}$ is the maximum-entropy state of all states consistent with the state $|Ψ〉$ on balls $AB$ and $BC_Y$." and "[...] $\rho_Y^{|Ψ〉}$ must be the maximum-entropy state in $\Sigma(Y)$" are not clear. Could the authors please elaborate on these?
1.7) Under the last bullet point on page 67, the statements "This is sufficient to guarantee that the superselection sector on the spherical boundary of $X \cup Y$ is Abelian. The minimization of entropy follows." Could the authors please elaborate on their reasoning behind these? It does not seem entirely obvious.
1.8) In the proof of Lemma 5.8, how exactly does the natural partition condition for the state $|Ψ_X (\mathcal M)〉$ imply eq (5.4)?
1.9) In the remark preceding Proposition 5.10, the authors note that "we can use Eq. (2.6) because $\partial X$ is embedded in $\mathbf{S}^n$". Do we assume that all the connected components of $\partial X$ are embedded? This certainly is not always true for building blocks, according to Definition 4.5. Could the authors please clarify what they mean in the quoted sentence? A similar statement is also made in the discussion preceding eq (7.4); does that have the same explanation?
1.10) Below eq (6.4), the authors state that the multiplicities in eq (5.8) are either 0 or 1 since X and Y are sectorizable. Could they please elaborate on why this is the case?

There are a few other points of confusion which we would request the authors' comments on:
2.1) In footnote 9, the authors state that the axioms A0 and A1 may not be exact for chiral phases. In what way do they fail? Is it related to correlation lengths diverging?
2.2) What are the labels $a_L$ and $a_R$ in eq (2.5)? Do they take values in $C_{\partial \Omega_L}$ and $C_{\partial \Omega_R}$?
2.3) In the Remark above Lemma 2.2, the notation $\Omega$ seems to be replaced by $\mathcal M$. Is that intended? Since the authors refer to the subregion $A$, perhaps they could clarify how $\mathcal M$ is related to $A$ and $\Omega$.
2.4) On page 24, first paragraph of section 3, the authors state "For non-sectorizable regions, information about fusion spaces can be detected; in the case of a nontrivial fusion space, the information is quantum.": could the authors please explain what they mean by this? In particular, what exactly does the information being "quantum" mean in this context.
2.5) In the description of the 3d $S_3$ quantum double (Example 3.1), the quantum dimension of the pure flux loops are stated to be $1, \sqrt{2}, \sqrt{3}$. This seems a bit confusing since the quantum dimensions of the counterparts of these excitations in 2d are $1, 2, 3$, respectively (equal to the sizes of the corresponding conjugacy classes). Is this difference due to some difference in definitions?
2.6) In statement 2 of Lemma 4.4, what does $\widetilde{\mathcal{W}}_+ = \widetilde{\mathcal{W}} $ mean? How are they equal if one is obtained from the other by thickening of the boundary.
2.7) What does $\sigma_Y$ mean in condition 2(b) of Lemma 5.5? Is this the reduced density matrix on Y obtained from the reference state $\sigma$?
2.8) What does the notation $\text{dim} \mathbb{V}(\mathcal M)$, used e.g. in eq (5.8) mean? Does it refer to the Hilbert space of ground states on the manifold $\mathcal M$?
2.9) The notation in eq (6.4) is somewhat unclear. Do $\mathcal C_{\text{point}}$ and $\mathcal C_{\text{loop}}$ refer to $\mathcal C_{X}, \mathcal C_{Y}$ or $\mathcal C_{\partial X}, \mathcal C_{\partial Y}$?

I believe the results obtained in the paper are interesting. The paper is suitable for publication in SciPost Physics, provided the authors address questions above and the requested changes below.

Requested changes

Major changes: 1- I suggest the authors move the discussions associated with gapped boundaries to an appendix. Section 6.3 may also be better suited to an appendix. I request the authors to consider that possibility. 2- Please address the requests for clarification in the Report above. 3- Please elaborate on the relation between the unitarity of pairing matrix and remote detectability (more details provided in Weakness #3).

Minor changes: 1- Some additional figures would be really helpful to better understand the second bullet point under the second example of bulk building blocks described under Example 4.7. In particular, an illustration of the following would be very illuminating: "[...] obtain the immersed region in question by merging a state on an immersed disk to the vacuum of the k disjoint embedded annuli". 2- The sentence above Proposition 4.12 is redundant and should be removed. 3- Please use a darker green font for the phrase "this color" in the discussion following eq (6.1). The current one makes the words rather hard to read on a white background. 4- In the last sentence of the Remark above Example 6.2, "the relation to" is redundant.

---

## Round 2 · Referee Report · Anonymous (Referee 1) · 2024-10-31

Report

The updated manuscript adequately clarifies the relationship of the setup of entanglement bootstrap to continuum topology. I believe that the discussion as presented is logically consistent, and includes necessary caveats and disclaimers wherever certain reasonable assumptions are made without rigorous proof.

Recommendation

Publish (surpasses expectations and criteria for this Journal; among top 10%)

---

## Round 2 · Referee Report · Anonymous (Referee 2) · 2025-3-17

Strengths

1 the paper addresses an important issue in the study of topological phases of matter
2 connects topology and quantum information

Weaknesses

1 Not mathematically rigorous
2 Some arguments are very short and cryptic and often refer to prior work by the authors which is also not rigorous
3 There are no non-trivial examples of states satisfying the axioms (at least not in this paper)

Report

Since the topic and the approach are interesting, I can recommend the paper for publication in SciPost. It may invite further studies of the "entanglement bootstrap" which will clarify the issues mentioned above.

Recommendation

Publish (meets expectations and criteria for this Journal)

---

## Round 2 · Author Response

In this new version, we included an edited version of the draft. We believe this version addresses the comments in referee report 2 (by referee 3). The reply to the referee report is updated separately. We strongly believe that our manuscript now meets the requirement for publication in SciPost Physics.

---

## Round 2 · List of Changes

1. We added a footnote in the introduction (on page 5). We mentioned explicitly that part of the data carried by the large ball (associated with the reference state) is a cell complex. The topology can thus be defined using the cell complex.

  2. We edited footnotes 8, 9, and 12 to make the meaning more clear. In particular, in footnote 9, we commented that continuum topology is not needed for our purpose. The purpose of the ``extra thickness'' is explained in footnote 12, and we explained why this is well under control. This should remove the confusion that worries the referee.

  3. We added a Remark on page 40, after the statement of the completion trick. This remark addresses the definition of topology from the cell decomposition.

---

## Editorial Decision

published